# The Tube-Web Spiders of the Genus *Ariadna* (Araneae: Segestriidae) from South Australia and Victoria [†]

Jessica R. Marsh [1,2,3,*], Mark I. Stevens [2,4], Tessa Bradford [2,4] and Volker W. Framenau [1,5,6]

1   Harry Butler Institute, Murdoch University, Murdoch, WA 6150, Australia
2   Earth and Biological Sciences, South Australian Museum, Adelaide, SA 5001, Australia
3   Invertebrates Australia, Ltd., Osborne Park, WA 6017, Australia
4   School of Biological Sciences, University of Adelaide, Adelaide, SA 5005, Australia
5   Department of Terrestrial Zoology, Western Australia Museum, Locked Bag 49,
    Welshpool DC, WA 6986, Australia
6   Centre for Taxonomy & Morphology, Leibnitz Institute for the Analysis of Biodiversity Change (LIB),
    Zoological Museum Hamburg, Martin-Luther-King-Platz 3, 20146 Hamburg, Germany
*   Correspondence: jess.marsh@murdoch.edu.au
†   urn:lsid:zoobank.org:pub:4DA6C8E7-F62F-4522-B064-B4B33EDC924C.

**Abstract:** The tube-web spider genus *Ariadna* Audouin, 1826 has been revised for South Australia and Victoria, revealing a remarkable diversity, particularly centred in the arid north of South Australia. We describe 23 species as new, ten of which are supported by molecular data, where these were available. We recognise two species groups for some of the species based on a combination of genitalic morphology, macrosetae patterns and somatic characters: the *clavata* species group, which includes *Ariadna clavata* Marsh, Baehr, Glatz & Framenau, 2018 and *A. spinosa* **sp. nov.** from South Australia, and *A. otwayensis* **sp. nov.** and *A. sinuosa* **sp. nov.** from Victoria, and the *formosa* species group, including *A. formosa* **sp. nov.** and *A. umbra* **sp. nov.** from South Australia, and *A. tria* **sp. nov.** from Victoria. Seventeen new species could not be placed into these two species groups: *A. arenacea* **sp. nov.**, *A. bellatula* **sp. nov.**, *A. curvata* **sp. nov.**, *A. deserta* **sp. nov.**, *A. diucrura* **sp. nov.**, *A. flavescens* **sp. nov.**, *A. inflata* **sp. nov.**, *A. insula* **sp. nov.**, *A. pollex* **sp. nov.**, *A. propria* **sp. nov.**, *A. rutila* **sp. nov.**, *Ariadna simplex* **sp. nov.**, *A. subplana* **sp. nov.**, *A. una* **sp. nov.**, *A. ungua* **sp. nov.**, *A. valida* **sp. nov.** and *A. woinarskii* **sp. nov**. We provide updated diagnoses and distributional data for *A. clavata* and *A. tangara* Marsh, Baehr, Glatz & Framenau, 2018; however, the holotype of *A. burchelli* (Hogg, 1900) from Victoria could not be located for this project.

**Keywords:** Australia; DNA barcodes; morphology; new species; Synspermiata; taxonomy

## 1. Introduction

Tube-web spiders in the family Segestriidae Simon, 1893 [1] construct tube webs with a characteristic circular opening, often with spurs of silk radiating away from the retreat [2,3]. The tube webs are typically around 5 cm long, with a 5–10 mm diameter opening. Tube webs are constructed in suitable crevices in the bark of trees and in logs, within borer holes in tree trunks and branches, in cracks of rocks, and in burrows in the ground [4,5]. Segestriidae have a characteristic body form, with six eyes and, dissimilar to other spiders, the third leg positioned forward when resting (for example [2,4]). The genitalia are simplified: the female genitalia are haplogyne, lacking sclerotised external copulatory structures, and the male pedipalp consists of an inflated, globular bulb and a simple embolus (for example [6]).

Segestriidae have a worldwide distribution in five genera, two of which occur in Australia: *Ariadna* Audouin, 1826 [7] (currently 24 described species) and *Gippsicola* Hogg, 1900 [8] (three species). *Gippsicola* has been recorded in New South Wales, Queensland, South Australia, Victoria and Western Australia [9]. *Gippsicola raleighi* Hogg, 1900 [8] is the

only species of *Gippsicola* recorded from South Australia and Victoria. *Ariadna* occurs in all Australian states and territories; however, no species from the Northern Territory have been described (J. Marsh, unpubl. data).

Prior to this study, two species of *Ariadna* were described from South Australia— *A. tangara* Marsh, Baehr, Glatz & Framenau, 2018 [4] and *A. clavata* Marsh, Baehr, Glatz & Framenau, 2018 [4]—and one from Victoria—*A. burchelli* (Hogg, 1900) [8].

South Australia is situated in the central southern portion of Australia, with a total land area of 984,321 km$^2$ [10]. The southern parts of South Australia have a largely Mediterranean climate, characterised by dry hot summers and wet mild winters, whereas the rest of the state is semi-arid to arid; the average annual rainfall for the state thus varies considerably, ranging from a maximum of 1200 mm along the southern coast to a minimum of 150 mm in the north of the state [11]. South Australia has a total of 17 terrestrial (IBRA) bioregions (Australia's bioregions (IBRA) – DCCEEW, accessed on 18 June 2022). Victoria is situated in south-eastern Australia and borders South Australia to the west. It has a total land area of 227,444 km$^2$ and a climate that ranges from temperate in the southern coastal and central regions to the dry, warm semi-arid region in the northwest of the state, and the alpine snowfields of the Victorian Alps in the northeast. The average rainfall varies considerably in the state, with the highest rainfall in the Otway Ranges and Gippsland in Southern Victoria, with an annual average rainfall exceeding 2500 mm, to less than 300 mm in the northwest [12]. There are 28 IBRA bioregions identified within Victoria [13].

This paper constitutes a revision of *Ariadna* for South Australia and Victoria, describing 23 new species and redescribing two. Species hypotheses are based on morphology and supported by molecular data where possible.

## 2. Methods

### 2.1. Molecular Analyses

All mtDNA COI were DNA barcoded using single-molecule real-time sequencing (SMRT) [14] in the PacBio Sequel platform (Pacific Biosciences, Menlo Park, CA, USA) at the Canadian Centre for DNA Barcoding (CCDB), University of Guelph, ON, Canada (see [15] for details). For the mitochondrial small ribosomal subunit, (12S) gene DNA was extracted from a single leg using either the Gentra Puregene DNA Purification protocol or the Mag-Max CORE kit on a KingFisher Duo Prime purification system (Thermo Fisher Scientific, Australia), according to the manufacturer's protocol for fresh tissues. PCR amplification of the 12S gene was carried out using primers 12sbi (5′-AAGAGCGACGGGCGATGTGT-3′) and 12sai (5′-AAACTAGGATTAGATACCCTATTAT-3′) [16]. PCRs were carried out in a 25 µL reaction volume consisting of nuclease-free water, 1 X Immolase PCR buffer, 1.5 mM MgCl$_2$, 0.8 mM dNTP mix, 0.05 mg/mL BSA, 0.24 µM primer, 0.5 u Immolase DNA polymerase (Bioline; NSW, Australia) and approximately 0.1–2 ng of extracted DNA. The cycling conditions were 94 °C for 5 min, followed by 40 cycles of 94 °C for 45 s/46 °C for 45 s/72 °C for 60 s, with a final extension at 72 °C for 10 min. Product purification and Big Dye Terminator sequencing were conducted by The Australian Genome Research Facility (AGRF). In addition to the new sequences here, we included other *Ariadna* and outgroups from BOLD (http://www.boldsystems.org/, accessed on 30 May 2022) and GenBank (https://www.ncbi.nlm.nih.gov/, accessed on 30 May 2022). The resulting sequence files were aligned in Sequencher v5.1 (http://www.genecodes.com) to produce a 594 bp (COI) and a 291 bp (12S) alignment. The nucleotide sequence divergence was calculated using uncorrected p-distances as implemented in MEGA 11.0.13 [17]. Substitution models and maximum likelihood (ML) trees were generated using the IQ-TREE web server using default settings [18], with 1000 ML bootstraps. The resulting trees were visualised and modified using FigTree v1.4.4 (http://tree.bio.ed.ac.uk/software/figtree/, accessed on 30 May 2022) and Adobe Illustrator (Adobe Systems, Inc., San Jose, CA, USA). We use mitochondrial (mt) DNA to explore species boundaries within the genus *Ariadna*, which has been successfully used for this purpose in numerous invertebrate groups (for example [19–22]), including segestriid spiders [15,23].

### 2.2. Morphological Analyses

Morphological descriptions, macrosetae notation and imaging follow [4]; descriptions of the embolus follow [2]. Methods for the dissection of female genitalia follow [24] and [3]. Due to the damaging nature of dissecting the female genitalia, where another specimen was available from the same collecting locality as the holotype and where there was no doubt of conspecificity, this specimen was dissected to maintain the structure of the paratype female. Leg measurements are given as the total length, followed by the length of each segment of that leg.

Examination of the specimens was conducted using Zeiss Stemi 305 and Nikon SMZ18 stereomicroscopes. Images were taken using a Leica DFC 500 camera attached to a Leica MZ16A microscope. Images were taken in 10–20 focal planes and combined into a single image using AutoMontage Pro Version 5.2. Line drawings were traced from photographic images using Vectinator Graphic Design Version 4.10.3. Measurements were taken digitally using AutoMontage Pro Version 5.2. Measurements of the pedipalp tibia length were taken along the dorsal surface, and those of the pedipalp tibia width were taken at the widest point along the tibia. All measurements are given in millimetres.

The primary types of Australian species were examined during 2019–2022 as part of ongoing research on this family, with the exception of *A. natalis* Pocock, 1900 [25]. *Ariadna dysderina* L. Koch, 1873 [26] holotype (ZMH A0000801), *A. segmentata* Simon, 1893 [27] holotype (MNHN AR16175), *A. major* Hickman, 1929 [28] holotype (QVMAG QVM13:7361) and *A. muscosa* Hickman, 1929 [28] syntypes male and female (both QVMAG QVM13:7339) were only examined as images. A type of *A. burchelli* (Hogg, 1900) [8] could not be located at the Natural History Museum, London, or the Museum Victoria, Melbourne, the likely collections where it should be housed (J. Beccaloni, S. Hinkley, pers. Comm. To JRM).

We did not describe putative species of which only females are known, as they often have no definitive diagnostic morphological features. Due to sexual dimorphism, it is often not possible to match males to females based on morphology; therefore, a conservative approach was adopted, and male and female specimens were treated as conspecific only when both the molecular and morphology data indicated this to be the case. Following this, historically described species based only on the female could not be diagnosed from NSW, WA and QLD until a full revision for those states has been completed.

The taxonomic section lists species by species group (where species could be allocated to groups), and then alphabetically by location.

### 2.3. Species Delineation

Species delineation was conducted using a combination of morphological data and mitochondrial sequence data (tree topology and sequence divergence).

### 2.4. Abbreviations

ALE, anterior lateral eyes;
AME, anterior median eyes;
ap, apex;
bas, basal;
d, dorsal;
DL, dorsal lobe;
dp, dorsoprolateral;
dr, dorsoretrolateral;
ITC inferior tarsal claw;
p, prolateral;
PA, prolateral apophysis;
PAE, prolateral apical extension of the cymbium;
PLE, posterior lateral eyes;
PME, posterior median eyes;
r, retrolateral;

RA, retrolateral apophysis;
RAE, retrolateral apical extension of the cymbium;
STC, superior tarsal claw;
v, ventral;
VL, ventral lobe;
vp, ventroprolateral;
vr, ventroretrolateral.

*2.5. Collections*

AM, Australian Museum;
MV, Museum Victoria;
NHM London, Natural History Museum, London;
SAMA, South Australian Museum, Adelaide;
TMAG, Tasmanian Museum and Art Gallery;
QM, Queensland Museum;
QVMAG, Queen Victoria Museum and Art Gallery.

*2.6. Collection of Fresh Material*

All fresh material for this project was collected under the following Scientific Research Permits form the Government of South Australia, Department of Water and Environment: U26935 and U26373.

**3. Results**

We here describe 23 new species of *Ariadna* and provide updated distributions and diagnoses for two described species. We place seven species into two species groups based on genitalic morphology, macrosetae patterns and somatic characters: the *clavata* species group and the *formosa* species group. We were not able to allocate the remaining 18 species to a species group. Seven of the species treated in this manuscript are known from both sexes, the remainder only from males. The species treated here are distributed across South Australia but are only known from limited locations in Victoria.

*3.1. Molecular Species Delineation*

For our new species described here (see below), the molecular results supported the morphological results for most species (Figures 1 and 2) and allowed us to link males and females of *A. valida* sp. nov. and *A. woinarskii* sp. nov. (Table 1 and Figure 2).

The intraspecies sequence divergence for COI clearly differentiated *A. woinarskii* sp. nov. from all other species by at least 15.5% (Table S1). The intraspecies sequence divergence for 12S was greatest within *A. woinarskii* sp. nov. with a mean of 2.7%, within *A. valida* sp. nov. with a mean of 4.8% within and within *A. tangara* with a mean of 7.8% (Table S2).

The interspecific mtDNA 12S divergences between *A. subplana* sp. nov. and *A. una* sp. nov. and *A. subplana* sp. nov. and *A. bellatula* sp. nov. were 8.6% and 8.2%, respectively. These divergence values were relatively lower than observed in the COI (Figure 1 and Table S1) and were the lowest between all other pairwise species comparisons for our new species (Table S2). However, morphological analyses for these species revealed interspecific differences in the shape of the cymbium and embolus of the male pedipalp and in the size and relative position of the apophyses of metatarsus I (see below) and are consistent with species assessments based on the molecular and morphological analyses.

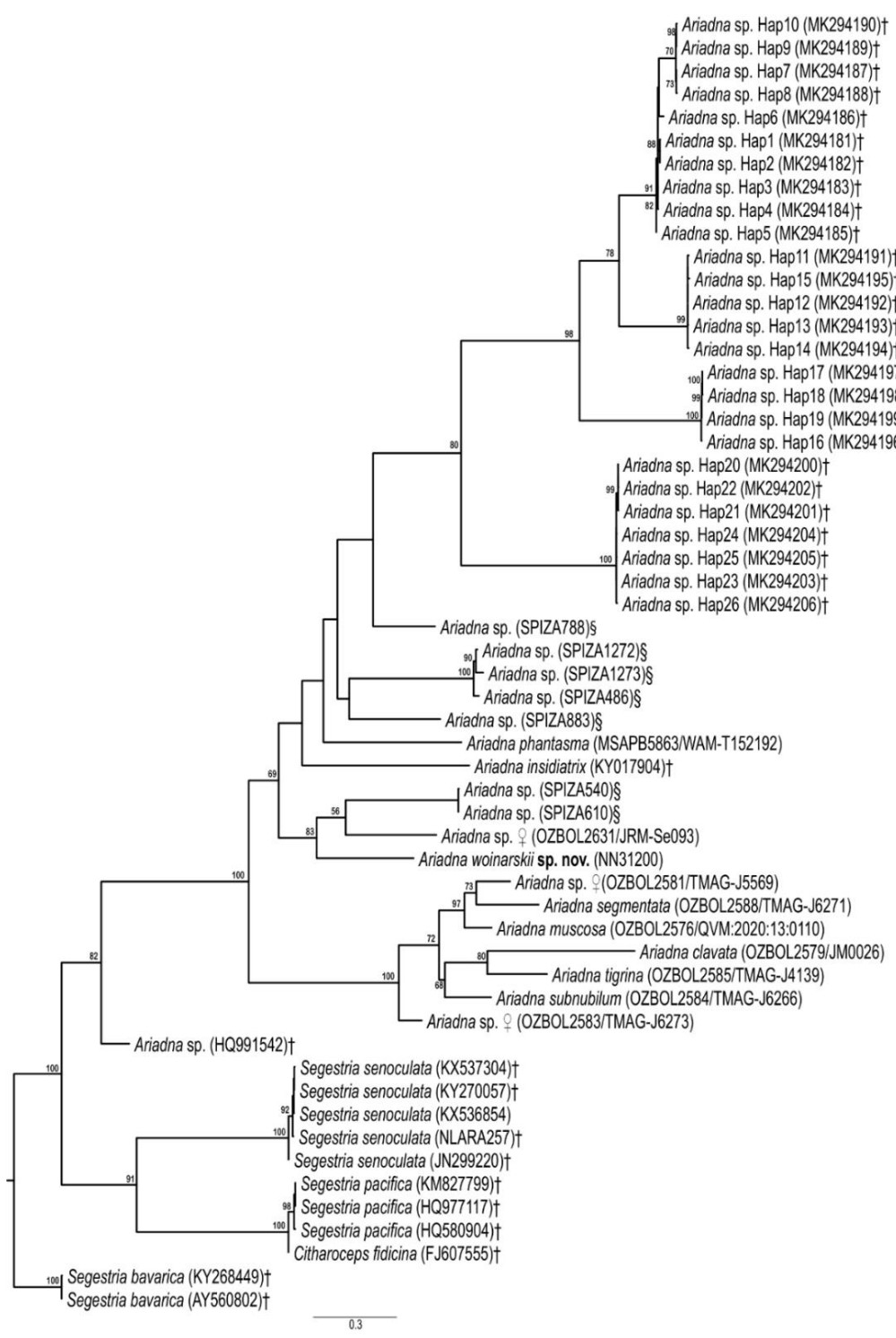

**Figure 1.** Maximum likelihood tree of the mitochondrial DNA COI gene of all 45 *Ariadna* sequences used and 11 outgroup taxa. Included are 21 *Ariadna* species, including the newly described *A. woinarskii* sp. nov. Bootstrap support values greater than 50% are shown at each node. Details for all new sequences are available in Table 1, and undescribed species with only females are indicated (♀). GenBank accessions (†) and BOLD process IDs (§) are indicated. The *Citharoceps fidicina* sequence obtained from GenBank is likely a misidentification.

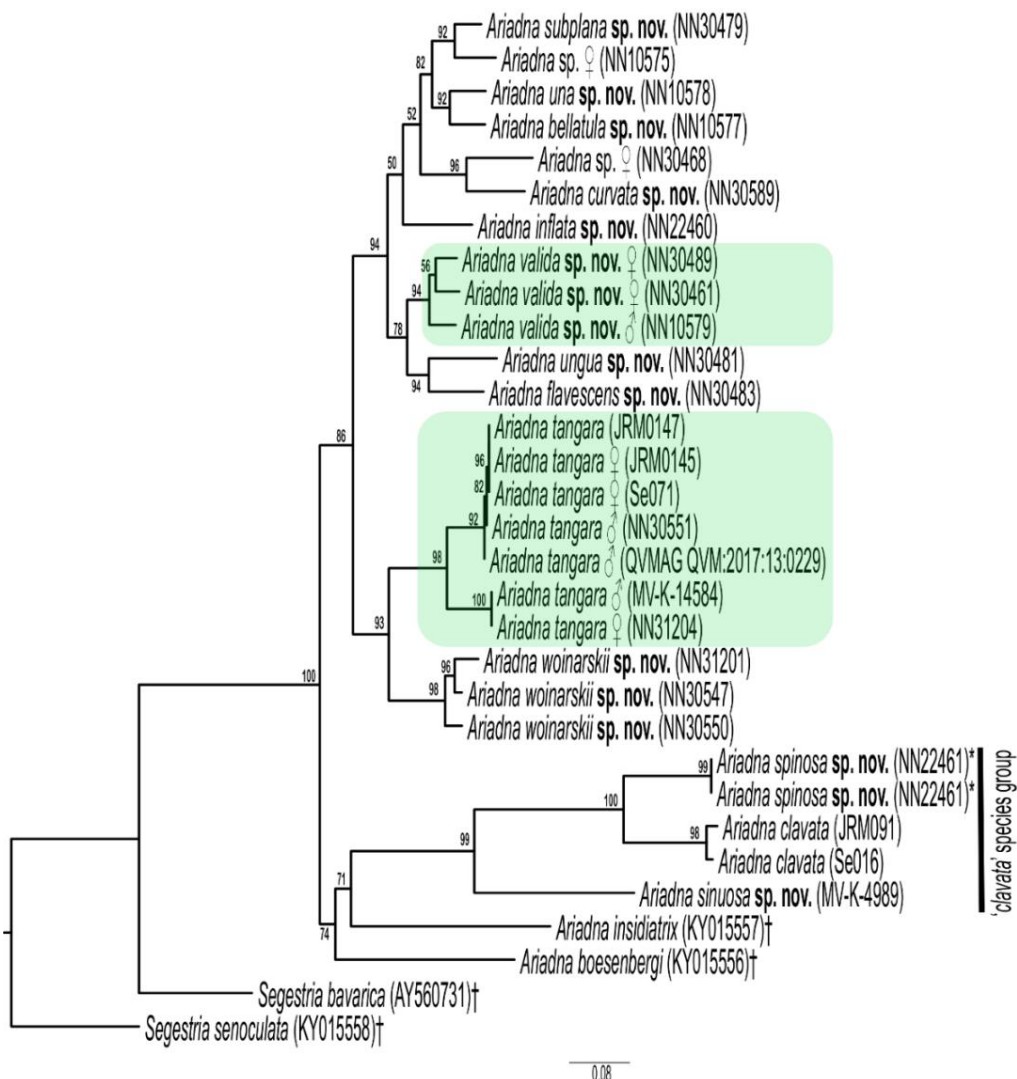

**Figure 2.** Maximum likelihood tree of the mitochondrial 12S gene of all 27 *Ariadna* sequences used and two outgroup taxa. Included are 17 *Ariadna* species, including eleven species newly described here. Bootstrap support values greater than 50% are shown at each node. Details for all new sequences are available in Table 1. Undescribed species from our study are indicated where only a female is known (♀), and we indicate where we have both males (♂) and females (♀). GenBank accessions are indicated (†), and we also indicate (green highlight) where we have both males and females of a species. * Indicates a specimen sequenced twice independently.

Members of the *clavata* species group formed a distinct clade, with strong morphological and molecular support to separate *A. spinosa* sp. nov. and *A. sinuosa* sp. nov. from *A. clavata* (Figure 2). *A. woinarskii* sp. nov. and *A. tangara* formed two well-supported clades (Figure 2). *A. tangara* was also split into two distinct clades (Figure 2), a South Australian and Tasmanian clade and a separate Victorian clade, with a mean divergence of 7.9% (12S). This is greater than any of the intraspecies divergences recorded in this study (maximum of 5.5%) and was comparable to the interspecies divergences between other species pairs (Table S2) and will be examined in greater detail in future works.

**Table 1.** Specimens sequenced in the current study, for all relevant and new species, with BOLD process IDs for COI and GenBank accession codes for 12S. Species treated in this study are presented in the order they appear in this manuscript.

| Taxon | Museum and Registration Number | Locality | Latitude | Longitude | GenBank Code COI | 12S |
|---|---|---|---|---|---|---|
| *Ariadna clavata* | JM0026 | SA, Kangaroo Island | 35.7871° S | 137.767° E | OZBOL2579 | |
| *Ariadna clavata* | JRM0091 | SA, Kangaroo Island | 35.87924° S | 135.57320° E | | OP516334 |
| *Ariadna clavata* | JRM Se016 | SA, Parndana Conservation Park | 35.725940° S | 137.316301° E | | OP516335 |
| *Ariadna spinosa* | SAMA NN22461 | SA, Apoinga | 33.93277° S | 138.95388° E | | OP516340 |
| *Ariadna sinuosa* | MV K-4989 | VIC, Central Highlands | 37.67861° S | 145.73889° E | | OP516339 |
| *Ariadna bellatula* | SAMA NN10577 | SA, Mundy Dam | 26.6733° S | 133.01667° E | | OP516333 |
| *Ariadna curvata* | SAMA NN30589 | SA, Arkaroola | 30.27778° S | 139.34694° E | | OP516336 |
| *Ariadna flavescens* | SAMA NN30483 | SA, Arckaringa Creek | 26.68778° S | 134.83972° E | | OP516337 |
| *Ariadna inflata* | SAMA NN22460 | SA, Marion Bay | 35.15944° S | 137.08777° E | | OP516338 |
| *Ariadna subplana* | SAMA NN30479 | SA, Stony Desert | 29.13° S | 135.16° E | | OP516343 |
| *Ariadna tangara* | MV K-14584 | VIC, Abbortsford | 37.80° S | 145.00° E | | OP516350 |
| *Ariadna tangara* | SAMA NN31204 | VIC, Newstead | 37.0873° S | 144.08017° E | | OP516346 |
| *Ariadna tangara* | JRM0147 | SA, Kangaroo Island | 35.787181° S | 137.766994° E | | OP516345 |
| *Ariadna tangara* | JRM0145 | SA, Kangaroo Island | 35.787181° S | 137.766994° E | | OP516344 |
| *Ariadna tangara* | JRM Se071 | SA, Kangaroo Island | 35.94° S | 136.78° E | | OP516348 |
| *Ariadna tangara* | SAMA NN30551 | SA, Coromandel Valley | 35.03° S | 138.63° E | | OP516349 |
| *Ariadna tangara* | QVMAG QVM:2017:13:0229 | TAS, Kings Meadow | 41.4673° S | 147.1542° E | | OP516347 |
| *Ariadna una* | SAMA NN10578 | SA, Illintjitja | 26.27722° S | 130.17278° E | | OP516351 |
| *Ariadna ungua* | SAMA NN30481 | SA, Lake Dam | 30.91° S | 139.82° E | | OP516352 |
| *Ariadna valida* | SAMA NN10579 | SA, Illintjitja | 26.26° S | 130.39° E | | OP516353 |
| *Ariadna valida* | SAMA NN30489 | SA, Mamungari Conservation Park | 28.57° S | 130.69° E | | OP516355 |
| *Ariadna valida* | SAMA NN30461 | SA, Mamungari Conservation Park | 28.55° S | 129.04° E | | OP516354 |
| *Ariadna woinarskii* | SAMA NN31200 | SA, Kangaroo Island | 35.6274° S | 137.236° E | OZBOL2629 | |
| *Ariadna woinarskii* | SAMA NN31201 | SA, Kangaroo Island | 35.6274° S | 137.236° E | | OP516358 |
| *Ariadna woinarskii* | SAMA NN30547 | SA, Lincoln National Park | 34.80° S | 135.94° E | | OP516356 |
| *Ariadna woinarskii* | SAMA NN30550 | SA, Head of Bight | 31.46° S | 131.12° E | | OP516357 |
| *Ariadna* sp. | SAMA NN10575 | SA, Mitchell Nob | 26.18° S | 131.88° E | | OP516341 |
| *Ariadna* sp. | SAMA NN30468 | SA, Oodnadatta | 26.93° S | 133.71° E | | OP516342 |
| *Ariadna* sp. | JRM Se093 | SA, Flinders Ranges | 31.45° S | 138.76° E | OZBOL2631 | |
| *Ariadna muscosa* | QVMAG QVM:2020:13:0110 | TAS, Launceston | 41.44° S | 147.11° E | OZBOL2576 | |
| *Ariadna phantasma* | WAM-T152192 | WA, Lake Cowan west | 32.185° S | 121.727° E | MSAPB5863 | |
| *Ariadna segmentata* | TMAG J6271 | TAS, Bruny National Park | 43.40° S | 147.28° E | OZBOL2588 | |
| *Ariadna subnubila* | TMAG J6266 | TAS, McPartlan Pass | 42.83796° S | 146.24072° E | OZBOL2584 | |
| *Ariadna tigrina* | TMAG J4139 | TAS, Flat Rock | 42.613° S | 147.257° E | OZBOL2585 | |
| *Ariadna* sp. | TMAG J5569 | TAS, Flinders Island | 40.057° S | 148.087° E | OZBOL2581 | |
| *Ariadna* sp. | TMAG J6273 | TAS, South West National Park | 43.193° S | 146.265° E | OZBOL2583 | |

### 3.2. Morphological Characters for Species Delineation

In males, the key morphological characters for species delineation were the presence of apophyses on metatarsus I, their shape and relative position, the presence of PAE or RAE on the pedipalp cymbium, and the shape of the embolus (Figure 3). We recorded the following character states for an apophysis presence: (i) no apophyses; (ii) two apophyses, with the retrolateral apophysis smaller than the prolateral apophysis; (iii) two apophyses, both reduced and indistinct; and (iv) two apophyses, both distinctly raised and subequal in size. Of the species examined in this study, three character states were recognised for the relative position of the apophyses on metatarsus I: (i) prolateral apophysis in the mid portion; retrolateral apophysis in the basal quarter (ii), both apophyses situated in the mid-portion; and (iii) prolateral apophysis situated in the apical third, retrolateral apophysis in the mid portion.

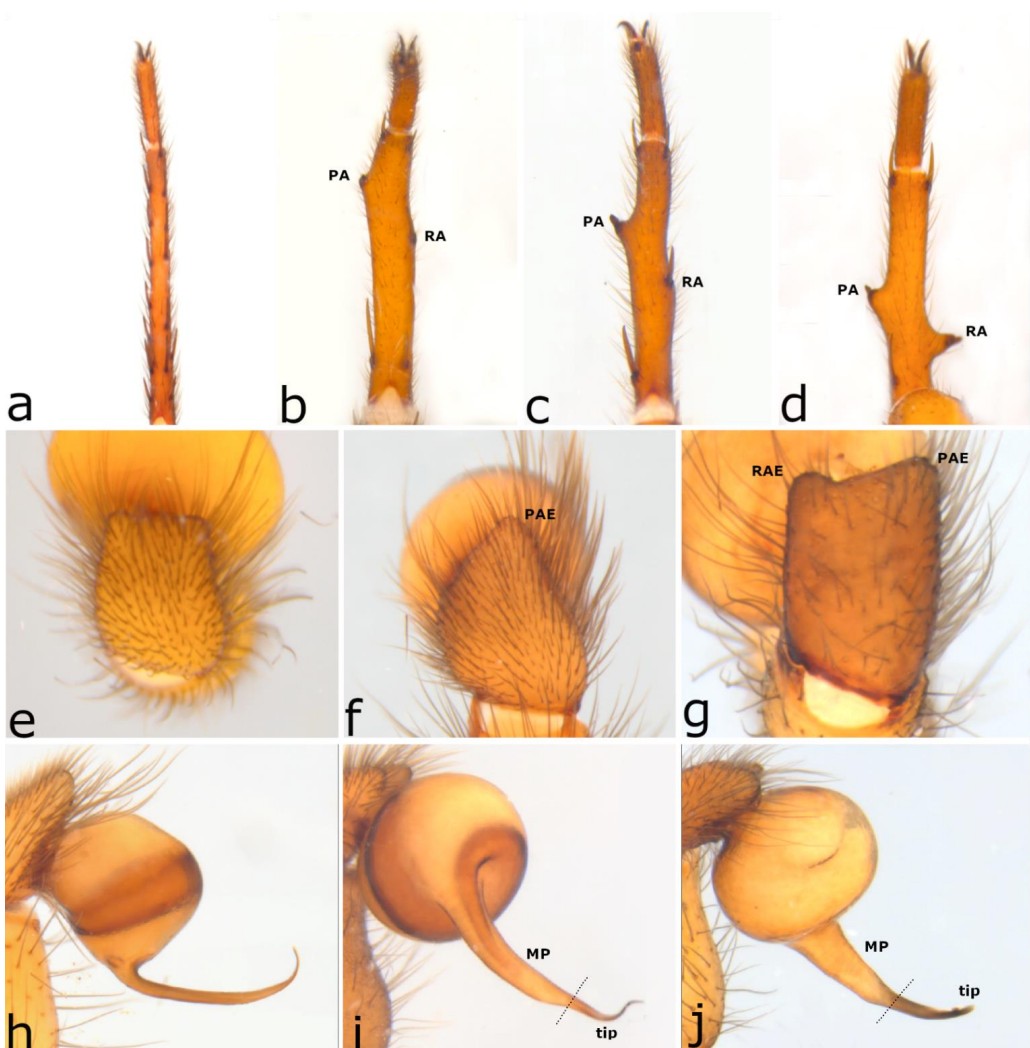

**Figure 3.** Summary of key diagnostic features for species delineation of male *Ariadna*: (**a**) metatarsus I *A. otwayensis* sp. nov., no apophyses; (**b**) same, *A. subplana* sp. nov., two reduced apophyses; (**c**) same, *A. inflata* sp. nov., two apophyses, one reduced, one prominent; (**d**) same, *A. propria* sp. nov., two prominent apophyses; (**e**) left pedipalp cymbium *A. propria* sp. nov. blunt apically, with no obvious PAE or RAE; (**f**) same, *A. otwayensis* sp. nov. PAE prominent, RAE highly reduced; (**g**) same, *A. umbra* sp. nov. PAE and RAE prominent; (**h**,**i**) details of embolus in (**h**) *A. sinuosa* sp. nov., (**i**) *A. formosa* sp. nov. and (**j**) *A. umbra* sp. nov.; the relative location of the transition from the mid portion (MP) to tip, indicated by a dashed line (**i**,**j**).

Whilst the absolute number of macrosetae for each leg segment showed some variation, there was within-species consistency in the pattern of macrosetae in (1) femur I and II dorsal prolateral macrosetae; (2) patella I and (3) tibia I, II prolateral, retrolateral, ventroprolateral and ventroretrolateral macrosetae. For example, in females of *A. woinarskii* sp. nov., the absolute number of prolateral macrosetae of tibia I ranged between 1 and 3 ($n$ = 12, mean = 1.9), and the retrolateral macrosetae ranged between 1 and 2 ($n$ = 12, mean = 1.5); however, at least one prolateral macroseta was always present (Table S3). The exceptions to this rule are species in the *clavata* species group, in which the macrosetae on tibia I and II are non-serial, and as such, the number and placement of macrosetae showed high levels of intraspecies variation.

The relative size and position of the apophyses of metatarsus of leg I were analysed for variation using *A. diucrura* sp. nov., which had apophyses and was known from multiple males. The relative position of apophyses on the metatarsus ranged from 0.60 to 0.64 of the metatarsus length for the prolateral apophysis and from 0.53 to 0.58 for the retrolateral apophysis ($n$ = 10) (see Table S3 for raw data for all species assessed).

## 4. Taxonomy

Family Segestriidae Simon (1893) [1]
Genus *Ariadna* Audouin, 1826 [7]
*Macedonia* Hogg, 1900 [8] (synonymised by [29])
*Pylarus* Hentz, 1842 [30] (synonymised by [6])
*Segestriella* Purcell, 1904 [31] (synonymised by [6])

**Diagnosis.** *Ariadna* can be separated from *Gippsicola* by the labium narrowing distally, as opposed to rectangular in *Gippsicola*; by the labrum lacking median setae dorsally and being longer than the labium; by a basal transverse ridge on the chelicerae; by the posterior eyes being slightly recurved or straight, as opposed to strongly recurved in *Gippsicola*; by the presence of tubular invaginations on the posterior receptaculum and by the tarsus of the female pedipalp having five or more macrosetae [9].

**Description.** *Cephalothorax:* Carapace oval; in lateral view, ranging from strongly flattened (Figure 17c and Figure 57c) to moderately domed (Figure 9c, Figure 13c and Figure 48c). Labium arrow-shaped, narrowing anteriorly, shorter than maxillae (Figure 6b, Figure 15b and Figure 17b). Chelicerae with basal transverse ridge (Figure 13d and Figure 15d), retromargin with single tooth, promargin with three teeth. Sternum oval, with scattered setae and precoxal triangles. Posterior eye row ranging from slightly recurved to straight (Figure 15d, Figure 20d and Figure 24d).

*Abdomen:* Oval and with a covering of fine setae.

*Legs:* Relative leg lengths variable but always with leg III shortest. Leg segments with varying densities of strong macrosetae, denser ventrally and with fine to dense coverings of setae. Metatarsus of some males with prolateral and/or retrolateral apophyses (Figure 35f, Figure 37f and Figure 40f). Femur I slightly to moderately bowed in dorsal view. Tarsus IV with a retrolateral distal preening comb. STC I and II with variable number of teeth, and ITC with a small tooth.

*Male pedipalp:* Tibia shape variable. Cymbium with either a defined PAE, with both a PAE and RAE or with neither and blunted apically (Figure 3). Bulb large and globular.

### 4.1. Key to the Ariadna of South Australia and Victoria

4.1.1. Males

1.  Metatarsus of leg I simple with macrosetae but without prolateral or retrolateral apophyses (Figure 11f, Figure 20f, Figure 22f and Figure 73f) . . . 2

    • Metatarsus of leg I with prolateral and/or retrolateral apophyses (Figure 55f, Figure 57f, Figure 62f and Figure 69f) . . . 11

2.  Abdomen with clear, well-defined and regular dark transverse markings on a pale cream background (Figure 4a, Figure 6a, Figure 9a and Figure 15a) . . . 3

- Abdomen dorsally grey in colour, without distinct, regular, transverse striations (Figure 20a, Figure 22a, Figure 29a and Figure 31a) . . . 6

3. In lateral view, tibia I with a slight but distinct ventral curvature (Figure 4g,h and Figure 9g,h) . . . 4

- In lateral view, tibia I straight (Figure 11g,h and Figure 13g,h) . . . 5

4. Tibia I without a cluster of macrosetae ventrally (Figure 4g,h), preening comb on metatarsus IV with five macrosetae (Figure 4e) . . . *A. clavata*

- Tibia I with a distinct ventral cluster of macrosetae situated basally (Figure 9f–h), preening comb on metatarsus IV with seven macrosetae (Figure 9e) . . . *A. spinosa* sp. nov.

5. Cymbium of palp with a PAE and a smaller but distinct RAE (Figure 14c) . . . A. sinuosa sp. nov.

- Cymbium of palp with a single, large PAE (Figure 12c) . . . *A. otwayensis* sp. nov.

6. Cymbium of palp blunt and squared apically, with no extensions (Figure 21c and Figure 54c) . . . 7

- Cymbium of palp with distinct PAE and/or RAE (Figure 18c, Figure 23c and Figure 27c) . . . 8

7. Pedipalp tibia elongate around 1.8 the length times the width (Figure 21a,b) . . . *A. tria* sp. nov.

- Pedipalp tibia stout around 1.2 the length times the width (Figure 54a,b) . . . *A. simplex* sp. nov.

8. Embolus distinctly thickened basally, remaining broad until around $\frac{3}{4}$ of its length and tapering to a gently curved and robust tip (Figure 23a,b) . . . *A. umbra* sp. nov.

- Embolus elongate, thin and ending in a thin, defined tip (Figure 18a,b, Figure 43a,b, and Figure 74a,b) . . . 9

9. Pedipalp tibia distinctly bulbous ventrobasally (Figure 18a,b). Preening comb with three elongate, widely separated macrosetae (Figure 17e) . . . *A. formosa* sp. nov.

- Pedipalp tibia not distinctly bulbous, preening comb with macrosetae clustered in a closely fitting row . . . 10

10. Metatarsus I straight when viewed ventrally, and with prolateral and retrolateral macrosetae in addition to ventral retrolateral and ventral prolateral macrosetae (Figure 42f–h) . . . *A. insula* sp. nov.

- Metatarsus I distinctly, broadly sinuous when viewed ventrally, and with only ventral retrolateral and ventral prolateral macrosetae (Figure 73f–h) . . . *A. woinarskii* sp. nov.

11. Metatarsus I with both the prolateral and retrolateral apophyses ill-defined, not strongly raised from metatarsus, resulting in metatarsus I appearing sinuous in ventral view (Figure 33e) . . . *A. deserta* sp. nov.

- Metatarsus I with at least one well-defined apophysis projecting from the metatarsus . . . 12

12. Metatarsus I with the retrolateral apophysis situated basally, located in the first quarter of the metatarsus (Figure 46f and Figure 62f) . . . 13

- Metatarsus I with the retrolateral apophysis situated more distally . . . 14

13. Pedipalp cymbium squared apically, with small, indistinct apical extensions (Figure 47c) . . . *A. propria* sp. nov.

- Pedipalp cymbium with a distinct PAE (Figure 63c) . . . *A. una* sp. nov.

14. Metatarsus I with the prolateral apophysis situated in the apical third of metatarsus I (Figure 37f, Figure 40f and Figure 55f) . . . 15

- Metatarsus I with the prolateral apophysis situated centrally (Figure 29f, Figure 31f and Figure 50f) ... 23

15. Prolateral apophysis a broad-based, low pyramidal mound, retrolateral apophysis indistinct, only slightly raised from metatarsus (Figure 55f) ... *A. subplana* sp. nov.

- Prolateral apophysis distinctly elevated, projecting clear of the metatarsus ... 16

16. Retrolateral apophysis distinctly smaller than prolateral apophysis (Figure 26f and Figure 37f) ... 17

- Prolateral and retrolateral apophyses of metatarsus I around the same size (Figure 44f and Figure 65f) ... 22

17. Retrolateral apophysis of metatarsus I very indistinct, only slightly raised from the metatarsus (Figure 26f, Figure 37f and Figure 40f) ... 18

- Retrolateral apophysis of metatarsus I, although smaller than prolateral apophysis, distinct and projecting from the metatarsus (Figure 35f, Figure 51f and Figure 69f) ... 20

18. Embolus thickened up to midway from where it gradually narrows to a curved tip (Figure 38a,b) ... *A. flavescens* sp. nov.

- Embolus thickened up to $\frac{3}{4}$ of its length, from where it narrows abruptly to a sinuous tip (Figure 27a,b and Figure 41a,b) ... 19

19. Embolus not elongated, around the same length as the width of the bulb (Figure 27a,b) ... *A. arenacea* sp. nov.

- Embolus elongated, clearly longer than the width of the bulb (Figure 41a,b) ... *A. inflata* sp. nov.

20. Retrolateral surface of tibia I with a group of three robust macrosetae on a shared base (Figure 57h) ... *A. tangara*

- Tibia without a group of macrosetae sharing a common base ... 21

21. Cymbium with distinct but apically rounded PAE and indistinct RAE (Figure 36c) ... *A. diucrura* sp. nov.

- Cymbium with a distinct, angular PAE and RAE separated by a distinct 'v'-shaped notch (Figure 70c) .... *A. valida* sp. nov.

22. Pedipalp tibia relatively bulbous, 1.3 times the length than the width (Figure 45a,b) ... *A. pollex* sp. nov.

- Pedipalp tibia roughly rectangular-shaped, not bulbous with a straight, ventral edge (Figure 66a,b) ... *A. ungua* sp. nov.

23. Retrolateral apophysis on metatarsus I indistinctly raised from the metatarsus and substantially smaller than the prolateral apophysis (Figure 29f and Figure 50f) ... 24

- Prolateral and retrolateral apophysis similar in size, both raised distinctly from the metatarsus; both apophyses in the basal half of the metatarsus and more or less opposing (Figure 31f) ... *A. curvata* sp. nov.

24. Distinct ventral thickening of the embolus at around half of its length, from which it tapers to a sinuous tip (Figure 30a,b) ... *A. bellatula* sp. nov.

- Embolus smoothly curved along the length (Figure 51a,b) ... *A. rutila* sp. nov.

4.1.2. Females

(Females only known for *A. clavata*, *A. sinuosa* sp. nov. *A. propria* sp. nov., *A. tangara*, *A. umbra* sp. nov., *A. ungua* sp. nov., *A. valida* sp. nov. and *A. woinarskii* sp. nov.)

1. Abdomen with dorsal transverse striations (Figure 6a and Figure 15a) ... 2

- Abdomen dorsally uniform grey (Figure 48a, Figure 59a and Figure 67a) ... 3

2.  Ventral lobe of anterior receptaculum strongly sinuous in ventral view (Figure 16a); retrolateral preening comb on metatarsus IV consisting of seven macrosetae (Figure 15e) . . . *A. sinuosa* sp. nov.

    - Ventral lobe of anterior receptaculum more or less straight in ventral view (Figure 7a); retrolateral preening comb on metatarsus IV consisting of five macrosetae (Figure 6e) . . . *A. clavata*

3.  Metatarsus of leg I retrolaterally with a cluster of short, macrosetae (Figure 48h) . . . *A. propria* sp. nov.

    - Metatarsus of leg I with macrosetae but not a cluster of short, stout macrosetae retrolaterally . . . 4

4.  Retrolateral preening comb of metatarsus IV with three macrosetae (Figure 24e) . . . *A. umbra* sp. nov.

    - Retrolateral preening comb of metatarsus IV with four macrosetae . . . 5

5.  Dorsal or ventral lobe of the anterior receptaculum slightly to strongly sinuous (Figure 72a and Figure 76a) . . . 6

    - Dorsal or ventral lobe of the anterior receptaculum not sinuous . . . 7

6.  In lateral view, lobes of the anterior receptaculum elongate and strongly sinuous, with a ventral facing apical fold on the ventral lobe (Figure 76a,b) . . . *A. woinarskii* sp. nov.

    - Ventral lobe gently sinuous in ventral view. Dorsal lobe around the same length as the ventral lobe (Figure 72a,b) . . . *A. valida* sp. nov.

7.  In ventral view, ventral lobe of anterior receptaculum straight and bulbous apically, ventral lobe around twice as long as dorsal lobe, which is short and squat (Figure 68a,b) . . . *A. ungua* sp. nov.

    - Ventral lobe around 1.25 times longer than dorsal lobe (Figure 60a,b) . . . *A. tangara*

*4.2. Species Descriptions*

4.2.1. *Ariadna clavata* **Species Group**

**Diagnosis.** Abdomen with transverse striations. No apophyses on metatarsus I of males. Retrolateral preening comb of tarsus IV consisting of more than five macrosetae. Pedipalp cymbium with a strongly projecting and anteriorly narrowed PAE and a small, or absent, RAE; embolus simple, long, thin and hooked apically.

**Species included.** South Australia: *A. clavata* and *A. spinosa* sp. nov.

Victoria: *A. burchelli*; *A. otwayensis* sp. nov.; *A. sinuosa* sp. nov.

Tasmania: *A. abbreviata* Marsh, Stevens & Framenau, 2022 [15]; *A. fragilis* Marsh, Stevens & Framenau, 2022 [15] (Tasmania); *A. tigrina* Marsh, Stevens & Framenau, 2022 [15].

*Ariadna clavata* **Marsh, Baehr, Glatz & Framenau, 2018**

Figure 4a–h, Figure 5a–c, Figure 6a–h, Figure 7a,b and Figure 8

urn:lsid:zoobank.org:act:6C44FD1E-17E3-4F3A-A289-778289106615

**Type material.** *Holotype* ♂ AUSTRALIA: *South Australia:* Kangaroo Island, American River 35.787329° S, 137.767213° E, under bark at 1.5 m height, *Eucalyptus diversifolia*, 3 June 2017, coll. J. Marsh (SAMA NN29861).

*Paratypes:* 1♀ Same data as holotype, except 8 July 2017 (SAMA NN29862). 1♂ Pelican Lagoon, American River, Kangaroo Island, 35.796496° S, 137.750963° E, under bark of *Eucalyptus cneorifolia*, coll. J. Marsh (LIB ZMH-A0003051); 1♀ Tangara Drive, American River, 35.782887° S, 137.771559° E, in tube web at 1 m height, in crevice of bark of *Allocasuarina muelleriana*, coll. J. Marsh (LIB ZMH-A0003052).

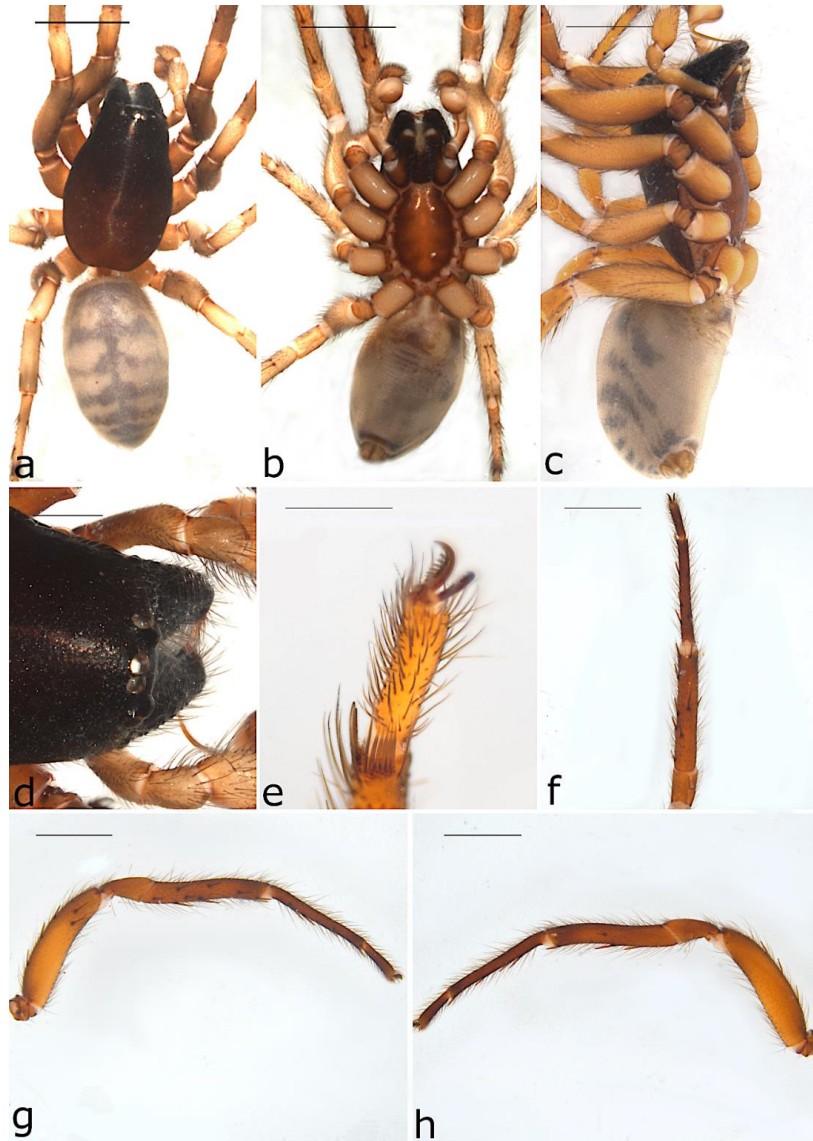

**Figure 4.** *Ariadna clavata* Marsh, Baehr, Glatz & Framenau, 2018. ♂holotype (SAMA NN29861) from Kangaroo Island (SA): (**a**) habitus, dorsal view; (**b**) same, ventral view; (**c**) same, lateral view; (**d**) eyes, dorsal view; (**e**) left metatarsus IV, preening comb, retrolateral view; (**f**) left leg I, ventral view; (**g**) same, prolateral view; and (**h**) same, retrolateral view. Scale bars (**a–d,f–h**) = 1 mm and (**e**) = 0.5 mm.

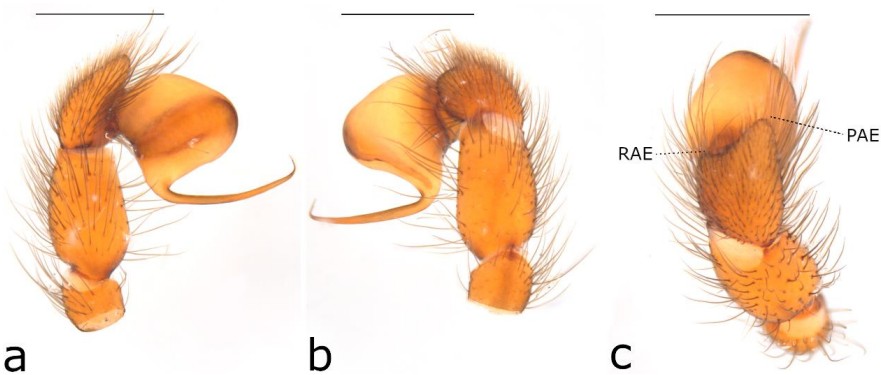

**Figure 5.** *Ariadna clavata* Marsh, Baehr, Glatz & Framenau, 2018. ♂holotype (SAMA NN29861) from Kangaroo Island (SA): (**a**) left pedipalp, prolateral view; (**b**) same, retrolateral view; (**c**) same, cymbium. Scale bar = 0.5 mm.

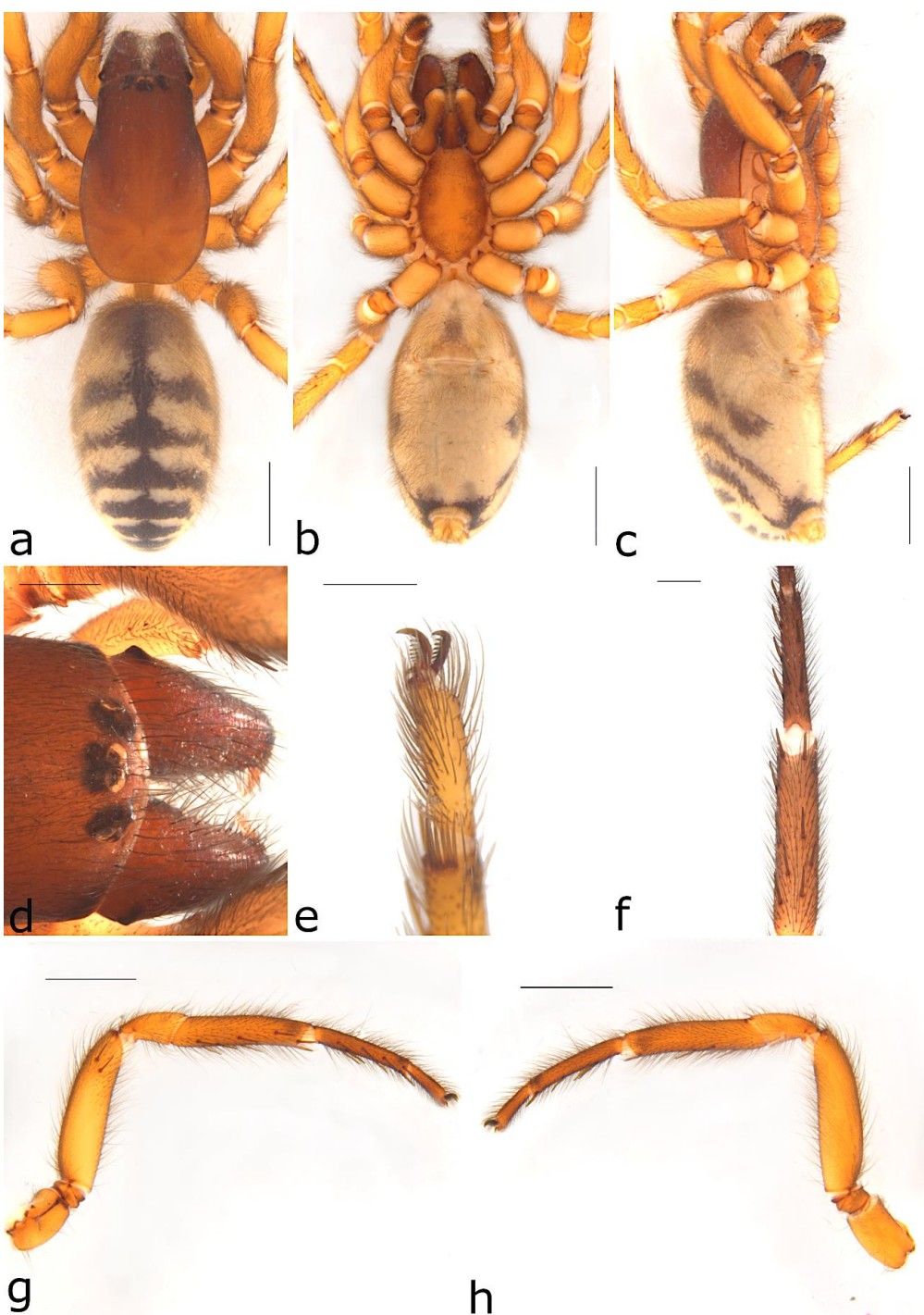

**Figure 6.** *Ariadna clavata* Marsh**,** Baehr, Glatz & Framenau, 2018**.** ♀paratype (SAMA NN29862) from Kangaroo Island (SA): (**a**) habitus, dorsal view; (**b**) same, ventral view; (**c**) same, lateral view; (**d**) eyes, dorsal view; (**e**) left metatarsus IV, preening comb, retrolateral view; (**f**) left leg I, ventral view; (**g**) same, prolateral view; (**h**) same, retrolateral view. Scale bars (**a**–**c**) = 1 mm, (**d**) = 0.5 mm, (**e**) = 1 mm, (**f**) = 0.5 mm and (**g,h**) = 1 mm.

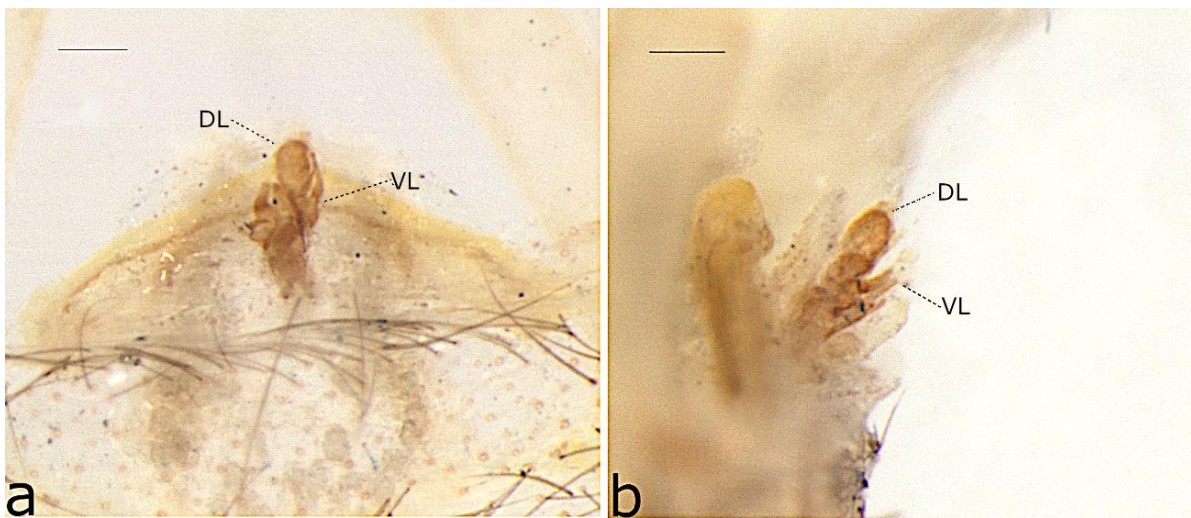

**Figure 7.** *Ariadna clavata* Marsh**,** Baehr, Glatz & Framenau, 2018. ♀paratype (SAMA NN29862) from Kangaroo Island (SA): anterior receptaculum (**a**) ventral view; (**b**) same, lateral view. Scale bar = 0.2 mm. DL = dorsal lobe, VL = ventral lobe.

**Figure 8.** Map showing the currently known distribution of *A. burchelli*, *A. clavata*, *A. sinuosa* sp. nov., *A. otwayensis* sp. nov. and *A. spinosa* sp. nov.

**Other material examined.** AUSTRALIA: *South Australia:* Kangaroo Island: 1♂Tangara Drive, American River, 35.78718° S, 137.78111° E, 2 June 2017, coll. J. Marsh (SAMA NN29865); 1♂same data, except 3 June 2017 (SAMA NN29866); 1♂same data, except 12 June 2017 (SAMA NN29870); 1♂Cannery Walking trail, American River, 35.77370° S, 137.78111° E, coll. J. Marsh, 16 August 2017 (SAMA NN29879); 1♀Pelican Lagoon Walking Trail, American River, 35.79258° S, 137.75790° E, 1 September 2017, coll. J. Marsh (SAMA NN29876); 2# same data, except 10 October 2017 (SAMA NN29881); 1♀, 4♂same data except 35.79649° S, 137.75096° E, 2 November 2017 (SAMA NN29899, NN29897); 1♀Pelican Lagoon Conservation Park, Dudley West, 35.80168° S, 137.77421° E, in tube web in bark of *Eucalyptus* sp., 13 October 2017, coll. J. Marsh; 3♂, 3♀Simpson Road, Dudley West, 35.82352° S, 137.83103° E, 22 June 2017, coll. J. Marsh (SAMA NN29910); 1♀Dudley Conservation Park, Dudley West, 35.79531° S, 137.85000° E, 22 June 2017, coll. J. Marsh (SAMA NN29911); 2♂Baudin Conservation Park, Dudley Peninsula, 35.73484° S, 137.95921° E, 15 July 2017, coll. J. Marsh (SAMA NN29908); 1♂same data, except 35.79594° S, 137.86079° E (SAMA NN29868); 1♀same data, except 35.79595° S, 137.86018° E (SAMA NN29872); 1♂, 3♀Antechamber Bay, Chapman River 35.79073° S, 138.06555° E, 1 September 2017, coll. J. Marsh (SAMA NN29892); 2♀Three Chain Road, MacGillivray, 35.91067° S, 137.55049° E, 26 July 2017, coll. J. Marsh (SAMA NN29909, NN29874); 3♀same data, except 7 August 2017 (SAMA NN29888, NN29889, NN29890); 1♂Penneshaw, Dudley Peninsula, 5–11 November 1990 (SAMA NN30524); 1♂, 2♀Yacca Jacks track, near Parndana Conservation Park, Parndana, 35.75718° S, 137.30731° E, roadside, under bark of *Eucalyptus cosmophylla*, 10 August 2017, coll. J. Marsh (SAMA NN29883); 1♀same data, except 35.75670° S, 137.30652° E (SAMA NN29895); 1♂same data, except 35.75704° S, 137.30723° E (SAMA NN29893); 1♂same data, except 35.75713° S, 137.30719° E (SAMA NN29880); 1♂Ravine Des Casoars Wilderness Protection Area, Kangaroo Island, 35.87924° S, 134.57320° E (JRM research collection, JRM0091). Mount Lofty Ranges: *South Australia:* 2♂Devils Gully Native Forest Reserve, Kersbrook, 34.7555556° S, 138.818889° E, October November 2000 (SAMA NN30523, NN30500); 1♂, Sixth Creek Native Forest Reserve, Forest Range, 34.90027° S, 138.79416° E, October/November 2000 (SAMA NN30501). Fleurieu Peninsula: 7♂Spring Mount, Minnawarra, 35.43222° S, 138.52388° E, 04–05 January 2006 (SAMA NN30581, NN30580, NN30577, NN30579, NN30574, NN30576, NN30578); 1♀Spring Mount, Minnawarra, 35.43222° S, 138.52388° E, 21–27 April 2005 (SAMA NN30582).

**Diagnosis.** Males and females of *A. clavata* can be differentiated from other species in the *clavata* group by the number of macrosetae in the preening comb of the retrolateral surface of leg IV, which has five macrosetae in *A. clavata* and six, seven or eight in other species in the group (Figures 4e and 6e vs. Figure 9e, Figure 11e, Figure 13e and Figure 15e).

**Remarks.** *Ariadna clavata* was described in detail recently [4].

**Distribution.** Known from SE South Australia (Figure 8).

*Ariadna spinosa* **sp. nov.**

Figures 8, 9a–h and 10a–c

urn:lsid:zoobank.org:act:74A09E5B-5ADC-4FF2-A738-5E02748A7066

**Type material.** *Holotype* ♂ AUSTRALIA: *South Australia:* 1.0 km ENE Apoinga, 33.93277° S, 138.95388° E, pitfall traps, hill slope, *Eucalyptus odorata* mallee, 27 October–1 November 2003 (SAMA NN22461).

**Etymology.** The specific epithet is a Latin adjective meaning 'spined' and refers to the spiny arrangement of the clustered macrosetae on the ventral surface of metatarsus I of this species.

**Diagnosis.** *Ariadna spinosa* sp. nov. can be separated from the males of other species in the *clavata* group by a combination of the cluster of macrosetae on the ventral surfaces of metatarsus I and by the shape of metatarsus I, which is bowed ventrally in lateral view in *A. spinosa* sp. nov., but not in other groups (Figure 9f–h vs. Figure 4f–h, Figure 11f–h and Figure 13f–h). It can be further differentiated from males and females of *A. sinuosa* sp. nov. and *A. clavata* by interspecies divergence in the 12S gene (Figure 2).

**Description.** ♂(based on holotype; SAMA NN22461). Total length 7.2.

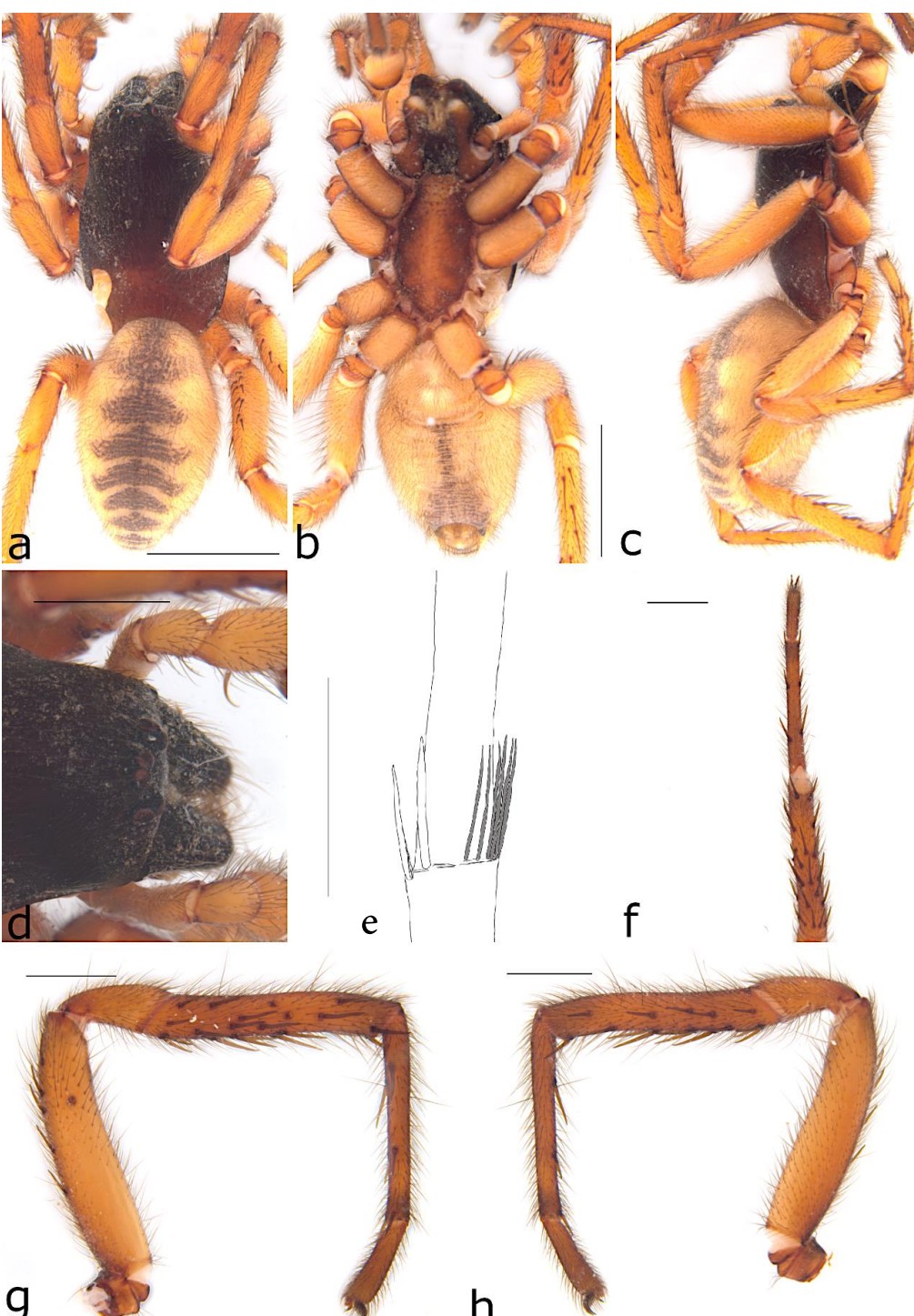

**Figure 9.** *Ariadna spinosa* sp. nov. ♂ holotype (SAMA NN22461) from Apoinga (SA): (**a**) habitus, dorsal view; (**b**) same, ventral view; (**c**) same, lateral view; (**d**) eyes, dorsal view; (**e**) left metatarsus IV, preening comb, retrolateral view, macrosetae of comb coloured grey for clarity; (**f**) left leg I, ventral view; (**g**) same, prolateral view; and (**h**) same, retrolateral view. Scale bars (**a–c**) = 2 mm, (**d**) = 1 mm, (**e**) = 0.5 mm and (**f–h**) = 1 mm.

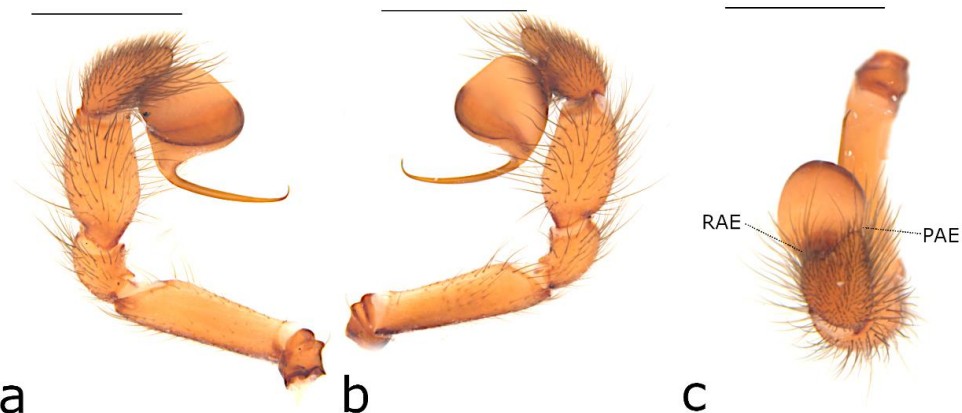

**Figure 10.** *Ariadna spinosa* sp. nov. ♂ holotype (SAMA NN22461) from Apoinga (SA): (**a**) left pedipalp, prolateral view; (**b**) same, retrolateral view; (**c**) same, cymbium. Scale bar = 1 mm.

*Cephalothorax:* 3.6 long, 2.4 wide, 2.1 high. Carapace dark red-brown, darker anteriorly, covered with sparse dark setae. Sternum dark golden-brown, lighter medially, with scattered black setae. Maxillae dark brown, white apically, labium dark brown, cream apically. Chelicerae dark brown to black. Figure 9a–c. Carapace oval, narrowing anteriorly to form a broad, squared 'neck'; fovea is an indistinct and shallow indented pit (Figure 9a). Labium about one-third the length of maxillae; sternum elongate oval, with blunt straight anterior edge and pointed posteriorly and with intercoxal extensions and with pre-coxal triangles (Figure 9b). In lateral view, carapace raised to the anterior, highest just behind eyes (Figure 9c). Eye group 0.8 wide, occupying 0.5 of the width of the carapace anteriorly; posterior eye row straight (Figure 9d).

*Abdomen:* 3.6 long. Dorsally yellow-cream with seven broad dark grey transverse abdominal striations medially, striations enlarged and connected medially. Ventrally yellow grey, with a violet, grey medial strip and surrounding the spinnerets. With brown setae (Figure 9a–c).

*Legs*: Leg length ratio II > I > IV > III. Leg I total 9.2: femur 2.7, patella 1.1, tibia 2.3, metatarsus 2.1, tarsus 1.0 Leg II 9.8: 2.6, 1.2, 2.5, 2.4, 1.1; Leg III 6.9: 2.1, 0.9, 1.7, 1.5, 0.7; Leg IV 8.2: 2.5, 1.0, 1.9, 1.9, 0.9. Legs orange–brown. Metatarsus I without apophyses; in lateral view, basal half of the tibia expanded slightly ventrally and covered with a cluster of non-serial macrosetae; femur I slightly bowed (Figure 9f–h). Macrosetae: Leg I: femur dp1ap, d1-1-1-1-1ap; tibia (non-serial) dp1-1-1, p1-2-1-1-1, vp1-1-1-1ap, v1-1-1, vr1-1-1-1ap, r1-1-1-1-1; metatarsus vp1-1-1ap, vr1-1-1ap. Leg II dp1, d1-1-1-1-1ap, tibia p1-1-2, vp1ap, v1-1-1, vr1-1-1-1-1ap, r1-1-1-1; metatarsus p1, vp1-1-1ap, v1, vr1-1-1ap. Leg IV femur 2-2-1-1, tibia p1-1, vp1ap, v1, vr1-1-1-1ap; metatarsus p1, v1, vr1-1-1-1ap, r1. Retrolateral distal preening comb with seven macrosetae (Figure 9e). STC 1 and 2 with around eight teeth, ITC with a small tooth.

*Pedipalp.* Tibia 1.2 long, 0.6 wide; cymbium 0.6 long, 0.4 wide. Pedipalp tibia moderately elongate, swollen basally in the dorsoventral plane. Bulb pyramidal with embolus arising ventrally. Cymbium with a large PAE; truncate and square edged retrolaterally. Embolus long, slender, hooked apically (Figure 10a–c).

**Distribution.** Only known from type locality in central N South Australia (Figure 8).

### *Ariadna burchelli* **(Hogg, 1900)**

Figure 8; plate 13, Figure 4a,b (Hogg 1900)

**Type material.** *Holotype* The holotype is presumed to be at the MV or the NHM, London, where much of Hogg's material was deposited, but could not be located at either institution for this project (S. Hinkley, J. Beccaloni, pers. comm. to JRM).

**Remarks.** The original description of *A. burchelli* does not include usable diagnostic features, and there are no written descriptions of the macrosetae or genitalia. The figure accompanying the description indicates strong ventral, or pro/retro-ventral macrosetae

on metatarsal segments of all four legs, with no other macrosetae on other leg segments; a combination of characters not seen in any species of *Ariadna* examined by the lead author (Hogg 1900; Figure 4). Hogg (1900) does not clearly indicate whether the specimen is mature or juvenile or male or female. Whilst reference is made to the 'genital aperture', thus suggesting the specimen may be female, there is some swelling of the pedipalp tarsus observable in Hogg's (1900) illustration, suggesting the specimen could possibly be a male or subadult male. Given the diversity of *Ariadna* species in Australia described to date, Hogg's (1900) description of *A. burchelli* cannot be ascribed to any of the species considered in this study. The species is tentatively placed in the *clavata* species group due to the transverse abdominal striations.

**Distribution.** The Macedon region, central Victoria [32].

*Ariadna otwayensis* **sp. nov.**

Figures 8, 11a–h and 12a–c

urn:lsid:zoobank.org:act:D3B9E10A-847F-4AA6-939F-679BB726C162

**Type material.** *Holotype* ♂ AUSTRALIA: *Victoria:* Otway Ranges, Young Creek Road 0.4 km NW of Triplet Falls, 38.67° S, 143.48° E, pitfall trap, *Eucalyptus* forest, 15 November 1994–31 January 1995, coll. G. Milledge (MV K-4985).

**Other material examined:** 1♂ AUSTRALIA: *Victoria:* Phillips Track, Young's Creek crossing, 0.5 km N of Triplet Falls, Otway Ranges, 38.67° S, 143.49° E, pitfall trap, *Nothofagus cunninghamii*, 4–10 December 1991, coll. G. Milledge, P. Lillywhite, C. McPhee, & B. Vanpraagh (MV K-14553).

**Etymology.** The specific epithet refers to the Otway Ranges, the only known locality for this species.

**Diagnosis.** *Ariadna otwayensis* sp. nov. can be differentiated from other species in the *clavata* species group by the shape of the cymbium, which has a single, smoothly triangular PAE in *A. otwayensis* sp. nov., without an RAE, whereas, in other species, there is a large PAE and a small, defined and angular RAE (Figure 12c vs. Figure 5c, Figure 10c and Figure 14c). It can be further differentiated from Tasmanian species in the *clavata* group by the shape of the bulb, which is bulbous and expanded dorsally and roughly pyramidal-shaped in *A. fragilis* and *A. tigrina* but is rounded oblong-shaped in *A. otwayensis* sp. nov. (Figure 12a,b vs. Figure 21a,b and Figure 38a,b [15]). The species can be differentiated from *A. alta* Marsh, Stevens & Framenau, 2022 by the number of macrosetae in the preening comb, being eight thick and well separated macrosetae in *A. alta* but seven fine and not well-separated macrosetae in *A. otwayensis* sp. nov. (Figure 11e vs. Figure 6e [15]).

**Description.** ♂ (based on holotype; MV K-4985). Total length 8.7.

*Cephalothorax:* 4.4 long, 2.9 wide, 3.1 high. Carapace red-brown, dark anteriorly and lighter red-brown posteriorly, with darker striations radiating out from fovea and with scattered dark setae. Sternum golden-red-brown, lighter medially, dark brown pre-coxal triangles. Maxillae dark orange-brown, white apically, labium dark red-brown, paler apically, and chelicerae dark red-brown. Figure 11a–c. Carapace rounded oval, narrowing into broad neck anteriorly; fovea a shallow indented pit (Figure 11a). Labium about three-quarters the length of maxillae. Sternum elongate oval, with scattered setae and with pre-coxal triangles and with rounded intercoxal extensions (Figure 11b). Carapace domed in lateral view, highest midway between eye group and fovea (Figure 11c). Eye group 1.1 wide, occupying 0.6 of the width of the carapace anteriorly; posterior eye row slightly recurved.

*Abdomen:* 4.3 long. Dorsally grey-cream, with around six indistinct broad, longitudinal dark grey striations broken up and mottled medially; ventrally pale grey-cream, with an indistinct broad darker grey band medially. With dense red-brown setae (Figure 11a–c).

*Legs*: Leg length ratio II > I > IV > III. Leg I total 14: femur 4.2, patella 1.4, tibia 3.6, metatarsus 3.6, tarsus 1.2. Leg II 14.3: 4.2, 1.5, 3.6, 3.9, 1.1. Leg III 10.7: 3.4, 1.3, 2.3, 2.6, 1.1. Leg IV 11.4: 3.5, 1.3, 2.5, 2.9, 1.2. Legs golden-orange-brown. Metatarsus I without apophyses (Figure 11f–h). Femur I slightly bowed. Tarsus I-IV straight in lateral view. Tarsi and metatarsi I–IV with sparse scopulate setae apically. Macrosetae: Leg I Femur dp2ap,

d1-1-1-1-1-1ap; tibia numerous non-serial macrosetae; metatarsus p1, vp1-1-1-1-1-1-1-1ap, vr1-1-1-1-1-1-1ap, r1. Leg II femur dp1ap, d1-1-1-1-1ap; tibia sparse non-serial macrosetae; metatarsus p1, vp1-1-1-1-1ap, vr1-1-1-1-1ap, r1. Leg IV femur d1-1-1; tibia p1-1-1, Vp1ap, v1, vr1-1-. v1-1; metatarsus p1, v1, vr1-1-1-1ap. Retrolateral distal preening comb with 7 macrosetae, equal in length (Figure 11e). STC 1, 2 with around eight teeth, ITC with a long, well-defined tooth.

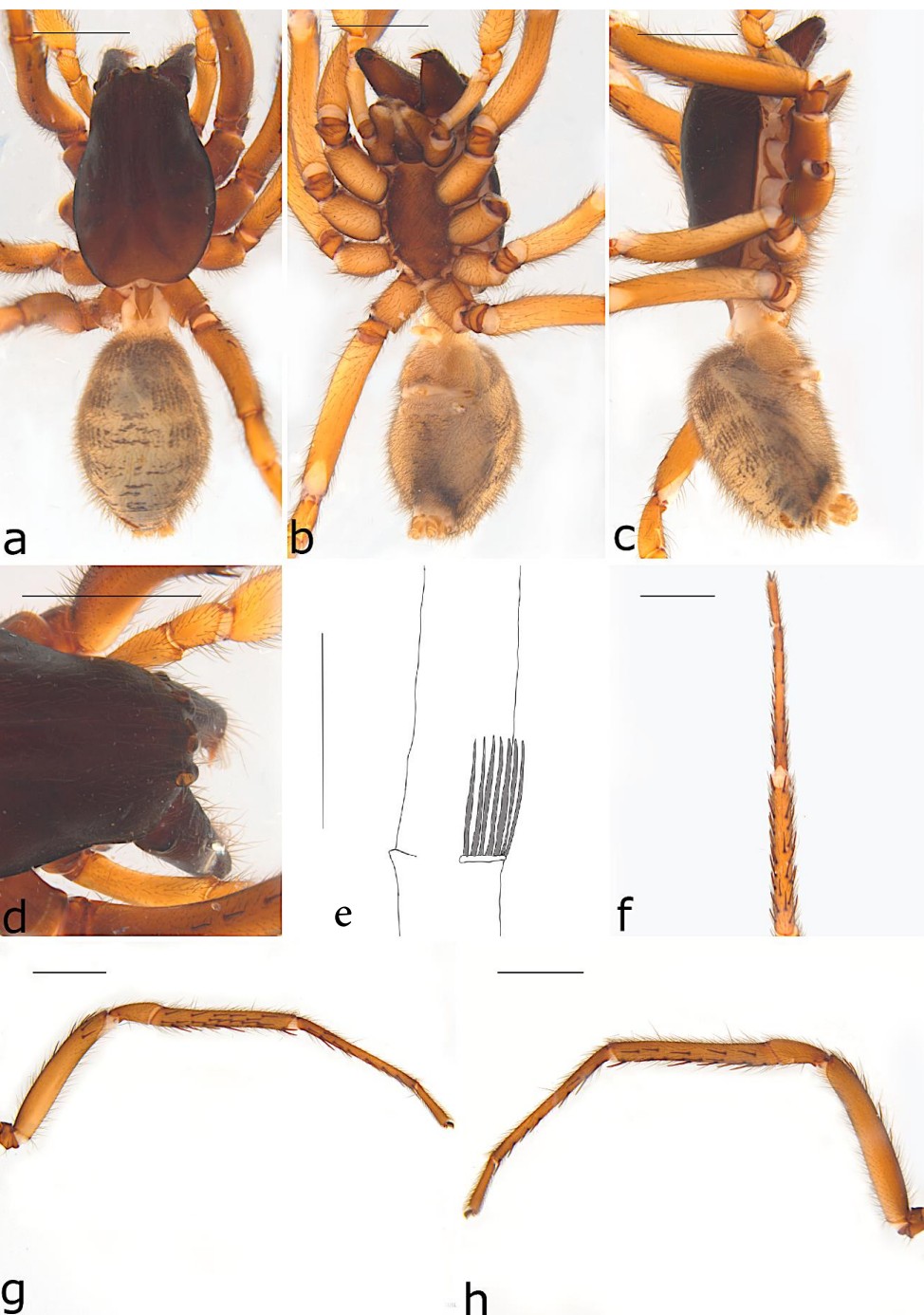

**Figure 11.** *Ariadna otwayensis* sp. nov. ♂ holotype (MV K-4985) from the Otway Ranges (Vic): (**a**) habitus, dorsal view; (**b**) same, ventral view; (**c**) same, lateral view; (**d**) eyes, dorsal view; (**e**) left metatarsus IV, preening comb, retrolateral view, macrosetae of preening comb shaded grey for clarity; (**f**) left leg I, ventral view; (**g**) same, prolateral view; (**h**) same, retrolateral view. Scale bars (**a**–**d**) = 2 mm, (**e**) = 0.5 mm and (**f**–**h**) = 2 mm.

*Pedipalp*: Tibia 1.1 long, 0.7 wide; cymbium 0.8 long, 0.6 wide. Tibia expanded proximally, narrowing apically. Cymbium squared, with single, large triangular prolateral extension apically and with a covering of scopulate setae. Bulb rounded oblong, embolus originating ventrally, thin along its length, elongate, curving smoothly to a gently hooked tip (Figure 12a–c).

**Variation.** Variation in carapace length was not calculated due to problems with access to material. The colour showed little to no variation in the two specimens examined.

**Distribution.** Only known from the Otway Ranges, Southern Victoria (Figure 8).

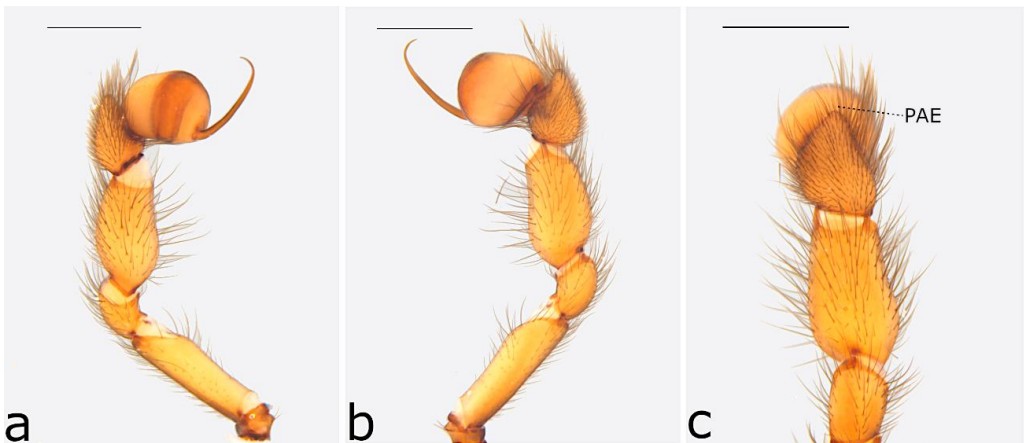

**Figure 12.** *Ariadna otwayensis* sp. nov. ♂holotype (MV K-4985) from the Otway Ranges (Vic): (**a**) left pedipalp, prolateral view; (**b**) same, retrolateral view; (**c**) same, cymbium. Scale bar = 1 mm.

### *Ariadna sinuosa* **sp. nov.**

Figure 8, Figure 13a–h, Figure 14a–c, Figures 15a–h and 16a,b
urn:lsid:zoobank.org:act:27027EAB-8393-4287-B8AE-87E9AF59327B

**Type material.** *Holotype* ♂ AUSTRALIA: *Victoria*, Central Highlands, Archeron Gap, 6 km NE of Mount Donna Buang, 37.67861° S, 145.73889° E, pitfall trap, *Nothofagus cunninghamii* forest, 28 December–21 February 1996, coll. G. Milledge (MV K-4989).

*Paratype* ♀AUSTRALIA: *Victoria*, Central Highlands, 0.7 km N of Archeron Gap, 7 km NE of Mount Donna Buang, 37.67138° S, 145.73889° E, pitfall trap, *Eucalyptus* sp. forest, 28 December–21 February 1996, coll. G. Milledge (MV K-4990).

**Other material examined.** *Victoria:* 1♂ same data as paratype (MV K-4990a); 1♂Central Highlands, The Big Culvert, 2.5 km ENE of Mount Observation, 37.56° S, 145.87° E, direct search, *Nothofagus cunninghamii* forest, 19 February 1996, coll. G. Milledge (MV K-4992).

**Etymology.** The specific epithet is a Latin adjective meaning 'sinuous' and refers to the sinuous anterior receptaculum of the female in ventral view.

**Diagnosis.** This species is most similar to *A. tigrina*, from which it can be differentiated by the porrect chelicerae in males and females of *A. sinuosa* sp. nov., contrasting with the hypognathous chelicerae of *A. tigrina* (Figures 13c and 15c vs. Figure 37c and Figure 39c [15]).

**Description.** ♂(based on holotype; MV K-4989). Total length 6.1.

*Cephalothorax:* 3.4 long, 2.3 wide, 1.8 high. Carapace dark red-brown, lighter posteriorly, with indistinct dark striae radiating out from the fovea, and with sparse light-brown setae, denser anteriorly. Sternum orange-brown, lighter medially and posteriorly, with broad darker striations extending inwardly between coxae. Maxillae orange-brown, pale cream apically; labium dark brown, paler apically. Chelicerae dark red-brown. (Figure 13a–c). Carapace oval, narrowing anteriorly and forming a broad squared neck; lateral margins slightly undulating; fovea is a shallow indented pit (Figure 13a). Labium about four fifths length of maxillae, chelicerae porrect. Sternum oval, with intercoxal extensions and with pre-coxal triangles. scattered setae and with pre-coxal triangles and with rounded intercoxal extensions (Figure 13b). In lateral view, carapace flattened, highest midway between fovea

and eye group (Figure 13c). Eye group 0.8 wide, occupying 0.6 of the width of the carapace anteriorly; posterior eye row slightly recurved (Figure 13d).

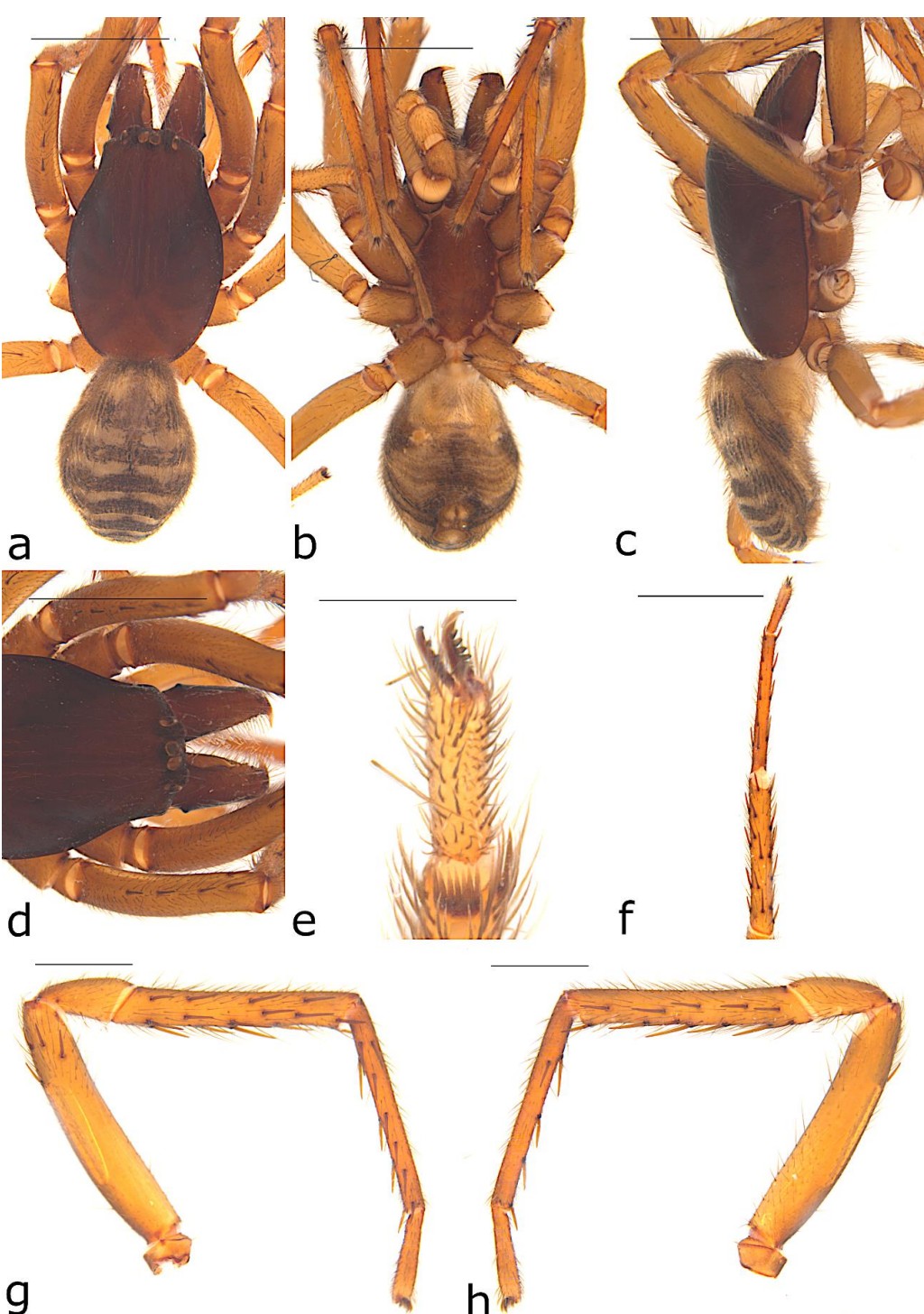

**Figure 13.** *Ariadna sinuosa* sp. nov. ♂holotype (MV K-4989) from Mount Donna Buang (Vic): (**a**) habitus, dorsal view; (**b**) same, ventral view; (**c**) same, lateral view; (**d**) eyes, dorsal view; (**e**) left metatarsus IV, preening comb, retrolateral view; (**f**) left leg I, ventral view; (**g**) same, prolateral view; (**h**) same, retrolateral view. Scale bars (**a–d**) = 2 mm, (**e**) = 0.5 mm, (**f**) = 2 mm and (**g,h**) = 1 mm.

*Abdomen:* 2.7 long. Dorsally with seven dark grey transverse abdominal striations on a pale-yellow base colour, striations connected by thin dark grey strip medially; ventrally with broad dark grey medial strip, laterally yellow-grey. With orange-brown setae (Figure 13a–c).

*Legs*: Leg length ratio I > II > IV > III. Leg I total 9.2: femur 2.7, patella 1.0, tibia 2.3, metatarsus 2.2, tarsus 1.0. Leg II 8.9: 2.6, 1.0, 2.3, 2.1, 0.9. Leg III 6.3: 2.0, 0.9, 1.2, 1.5, 0.7. Leg IV 7.2: 2.1, 1.1, 1.7, 1.6, 0.7. Legs orange-brown. Metatarsus I without apophyses, in lateral view tibia and metatarsus I straight; femur I slightly bowed (Figure 13f–h). Sparse scopulate setae on tarsi and metatarsi I–III, only on tarsi IV. Macrosetae: Leg I: femur dp2ap, d1-1-1ap; tibia dp1-1-1-1, p1-1, vp1-1-1ap, v1-1, vr1-1-1-1-1ap, r1-1-1-1-1; metatarsus p1, vp1-1-1-1-1-1ap, vr1-1-1-1ap, r1. Leg II femur dp1ap, d1-1-1-1, tibia p1-1-1, vp1-1ap, v1-1, vr1-1-1ap, r1-1-1-1, metatarsus p1, vp1-1-1-1ap, vr1-1-1-1ap, r1. Leg IV femur dp1ap, d1-1-1; tibia vp1ap, v1-1; metatarsus vr1-1-1a-1p, v1. Retrolateral distal preening comb with 7 macrosetae (Figure 13e). STC 1, 2 with around 11 teeth, ITC with small tooth.

*Pedipalp*. Tibia 0.8 long, 0.6 wide; cymbium 0.6 long, 0.5 wide. Pedipalp tibia short, swollen in the dorsoventral plane, length 1.3 times the width. Bulb oblong with embolus arising ventrally. Cymbium with a large PAE, retrolaterally squared and truncate. Scopulate setae covering cymbium. Embolus long, thin, hooked apically (Figure 14a–c).

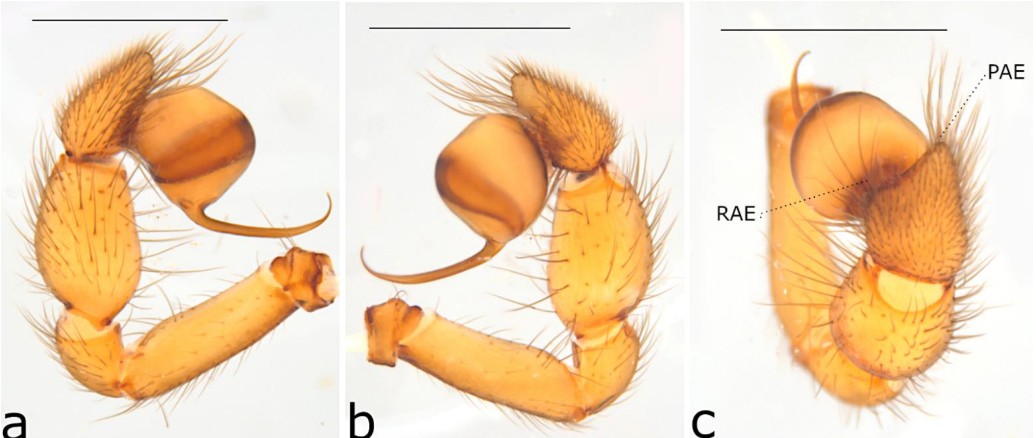

**Figure 14.** *Ariadna sinuosa* sp. nov. ♂holotype (MV K-4989) from Mount Donna Buang (Vic): (**a**) left pedipalp, prolateral view; (**b**) same, retrolateral view; (**c**) same, cymbium. Scale bar = 1 mm.

**Description.** ♀(based on paratype; MV K-4990). Total length 9.0

*Cephalothorax:* 3.7 long, 2.3 wide, 2.7 high. Carapace golden-brown, with sparse setae, small number of elongate setae projecting from between eye groups. Chelicerae orange-brown with few sparse and scattered setae. Sternum golden-brown, with darker intercoxal regions and paler medially. (Figure 15a–c). Carapace elongated oval, the lateral margins slightly undulating; narrowing anteriorly into a broad, squared neck; fovea is a shallow indented pit (Figure 15a). Sternum oval, with small rounded intercoxal extensions and with pre-coxal triangles. Labium about three quarters length of maxillae (Figure 15b). In lateral view, carapace domed, highest just posterior to the eye group (Figure 15c). Eye group 0.9 wide, occupying 0.5 of the width of the carapace anteriorly; posterior eye row slightly recurved.

*Abdomen:* 5.3 long. Dorsally with seven dark grey transverse abdominal striations, on a pale-yellow base colour, striations broadest medially and connected by a thin dark grey medial strip; ventrally grey, with faint darker grey longitudinal line medially. With sparse fark setae (Figure 15a–c).

*Legs*: Leg length ratio II > I > IV > III. Leg I total 7.8: femur 2.5, patella 0.9, tibia 2.0, metatarsus 1.7, tarsus 0.7. Leg II 7.9: 2.5, 1.0, 1.9, 1.6, 0.9. Leg III 5.8: 1.9, 0.8, 1.3, 1.2, 0.6. Leg IV 7.1: 2.4, 0.9, 1.6, 1.5, 0.7. Legs yellow-brown, leg I darker red-brown, with sparse dark long setae. Femur I bowed (Figure 15f–h). Macrosetae: Leg I: femur dp2ap; tibia p1-1, vp1-1-1ap, vr1-1-1-1ap; metatarsus v2-1-2-1-2-2. Leg II femur dp1ap, tibia p1ap, vr1-1-1ap; metatarsus 2-1-2-2-1. Leg IV femur dp1ap; tibia r1; metatarsus vr1-1. Retrolateral distal preening comb with 7 macrosetae (Figure 15e). STC 1, 2 with around 11/12 teeth, ITC with a small tooth.

*Pedipalps:* Multiple strong prolateral macrosetae on the tarsus and tibia and a single toothless claw.

*Genitalia:* Anterior receptaculum unilobed, strongly sinuous in ventral view (Figure 16a–c).

**Variation.** Variations in the carapace length were not calculated. Colour showed little to no variation in the specimens examined.

**Distribution.** Only known from the Central Highlands, Victoria (Figure 8).

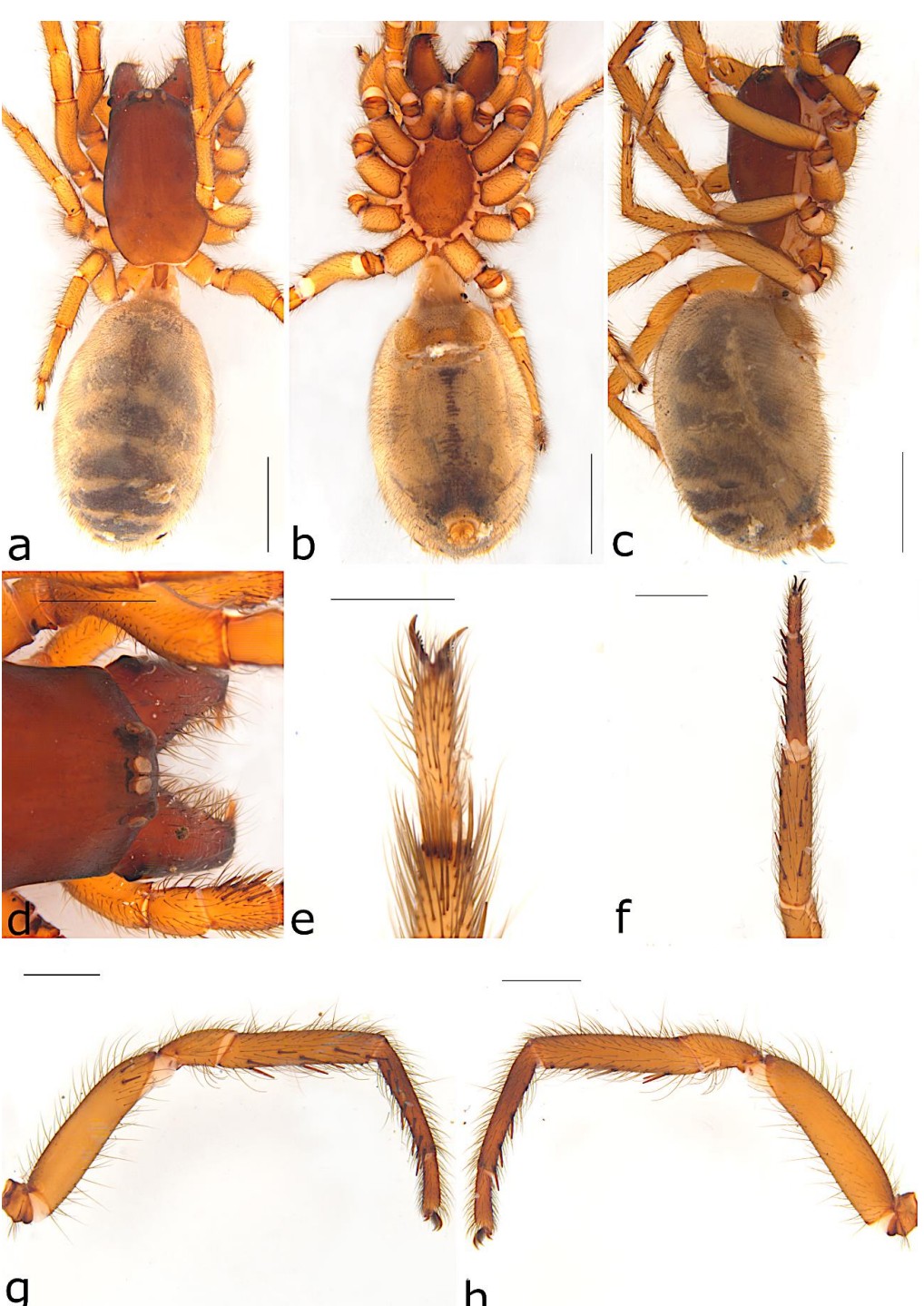

**Figure 15.** *Ariadna sinuosa* sp. nov. ♀paratype (MV K-4990) from Archeron Gap (Vic): (**a**) habitus, dorsal view; (**b**) same, ventral view; (**c**) same, lateral view; (**d**) eyes, dorsal view; (**e**) left metatarsus IV, preening comb, retrolateral view; (**f**) left leg I, ventral view; (**g**) same, prolateral view; (**h**) same, retrolateral view. Scale bars (**a–c**) = 2 mm, (**d**) = 1 mm, (**e**) = 0.5 mm and (**f–h**) = 1 mm.

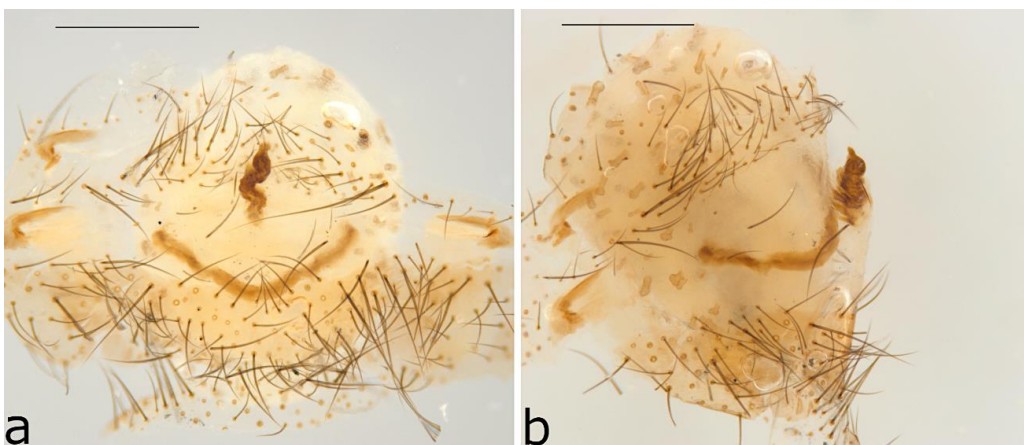

**Figure 16.** *Ariadna sinuosa* sp. nov. ♀(MV K-4990a) from Archeron Gap (Vic): anterior receptaculum (**a**) ventral view; (**b**) same, lateral view. Scale bar = 0.5 mm.

### 4.2.2. *Ariadna formosa* Species Group

**Diagnosis.** No apophyses on metatarsus I of males. Retrolateral preening comb of tarsus IV consisting of three macrosetae. Overall dark colouration, with dark brown cephalothorax, dark grey abdomen.

**Species included.** *South Australia: A. formosa* sp. nov., *A. umbra* sp. nov. *Victoria: A. tria* sp. nov.

### *Ariadna formosa* sp. nov.

Figure 17a–h, Figures 18a–c and 19

urn:lsid:zoobank.org:act:CBAF57C9-E4CE-48D7-A628-5F3F38480DA0

**Type material.** *Holotype* ♂ AUSTRALIA: *South Australia*, 19.9 km WNW Indulkana, 26.89889° S, 133.12277° E, *Acacia aneura* open woodland plain, pitfall trap, 23–31 October 1998, Pitjantjatjara Land Survey, ANMO6 (SAMA NN10565).

**Other material examined.** 7♂ same data as holotype (SAMA NN10564, NN10566, NN10567, NN10568, NN10569, NN10570, NN10571). 2♂ *South Australia*, 26.3 km ENE Mimli 25–31 October 1998, 26.91306° S, 132.95083° E, Sandy plain, very open woodland, *Acacia aneura/Acacia estrophiolata*, pitfall trap, Pitjantjatjara Land Survey, ANMO1 (SAMA NN10562, NN105623).

**Etymology.** The specific epithet is the Latin adjective *formosa* meaning 'beautiful' and refers to the violet-grey colouration of the abdomen, which contrasts with the dark carapace, and to the elongate, graceful form of the species.

**Diagnosis.** Of the species in the *formosa* species group, *A. formosa* sp. nov. can be differentiated from *A. tria* sp. nov. by the absence of PAE or RAE in the cymbium of *A. tria* sp. nov. (Figure 18a,b vs. Figure 21a,b) and can be further separated from *A. tria* sp. nov. and separated from *A. umbra* sp. nov. by the embolus, which is thickened in *A. tria* sp. nov. and *A. umbra* sp. nov. but is elongate and apically sinuous in *A. formosa* sp. nov. (Figure 18a,b, vs. Figure 21a,b and Figure 23a,b).

**Description.** ♂ (based on holotype; SAMA NN10565). Total length 7.3.

*Cephalothorax:* 3.5 long, 2.2 wide, 1.6 high. Carapace dark red-brown, strong black markings surrounding eye group. Sternum red-brown with darker brown mottled patches posteriorly, with dark red-brown pre-coxal triangular extensions. Maxillae orange-brown, white apically, labium mid brown, chelicerae red-brown. Figure 17a–c. Carapace elongate oval, narrowing anteriorly with broadly undulating lateral edges and with scattered sparse setae; fovea a shallow longitudinal furrow (Figure 17a). Labium elongate, about four-fifths length of maxillae; sternum elongated oval with pre-coxal triangles and with smaller, rounded intercoxal extensions, with scattered, long, black setae (Figure 17b). In lateral view, carapace flattened (Figure 17c). Eye group 0.7 wide, occupying 0.6 of the width of the carapace anteriorly; posterior eye row slightly recurved (Figure 17d).

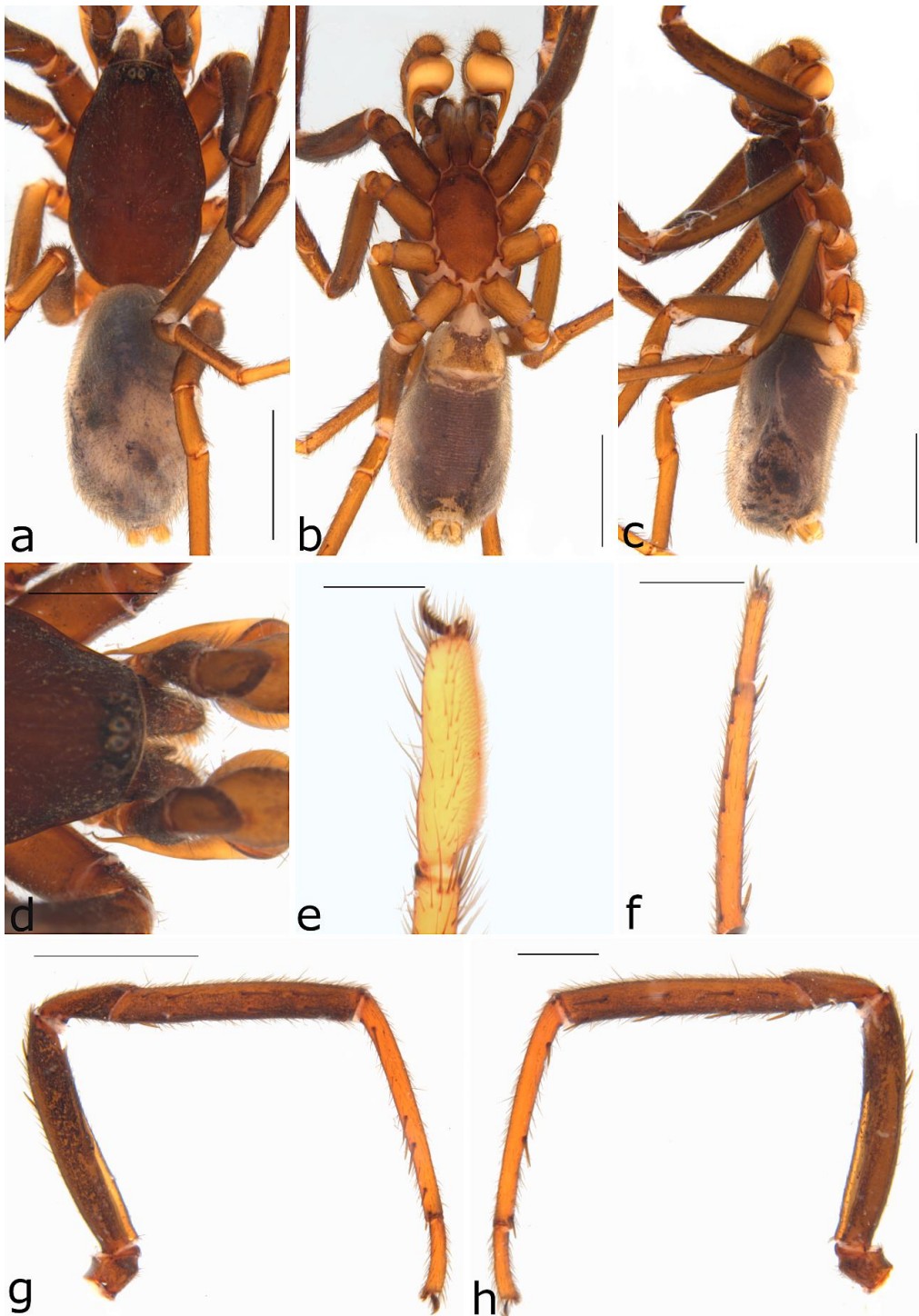

**Figure 17.** *Ariadna formosa* sp. nov. ♂holotype (SAMA NN10565) from Indulkana (SA): (**a**) habitus, dorsal view; (**b**) same, ventral view; (**c**) same, lateral view; (**d**) eyes, dorsal view; (**e**) left metatarsus IV, preening comb, retrolateral view; (**f**) left leg I, ventral view; (**g**) same, prolateral view; (**h**) same, retrolateral view. Scale bars (**a**–**c**) = 2 mm, (**d**) = 1 mm, (**e**) = 0.5 mm, (**f**) = 1 mm and (**g**,**h**) = 2 mm.

*Abdomen:* 3.8 long. Dorsally with dark grey and lighter grey mottled patches, pale yellow-cream patches covering book lungs, but otherwise dark violet-grey. With a covering of fine dark red-brown setae (Figure 17a–c).

*Legs:* Leg length ratio II > I > IV > III. Leg I total 10.5: femur 3.2, patella 1.1, tibia 2.7, metatarsus 2.6, tarsus 0.9. Leg II 11: 3.0, 1.1, 3.0, 2.9, 1.0. Leg III 8.4: 2.6, 1.0, 2.0, 1.8, 1.0. Leg IV 10.3: 3.1, 1.1, 2.9, 2.0, 1.2. Legs dark brown, metatarsus and tarsus of all legs paler orange-brown. Femur I slightly bowed in dorsal view. Metatarsus I straight in ventral view, no apophyses (Figure 17f–h). Tarsi I slightly curved ventrally in lateral view, slightly broader at apex than at base. Tarsi II, III straight in lateral view, tarsi IV inflated ventrally, tarsi II with sparse scopulate setae apically, tarsi III, IV with scopulate setae along entire ventral length. *Macrosetae:* Leg I: femur dp2ap, d1-1; tibia p1-1-1, vp1-1-1, vr1-1-1-1ap, r1-1-1-1; metatarsus vp1-1-1-1-1ap, vr1-1-1ap. Leg II: femur dp1ap, d1-1-1; tibia p1-1-2, r1-1-1; metatarsus vp1-1-1-1-1ap, vr1-1-1ap, r1. Leg IV: femur d1, tibia 1-1-1apretrolateral distal preening comb with 3 elongate, fine macrosetae, widely separated, 1 longer and 2 shorter (Figure 17e). STC I, II with 10 teeth, ITC with a small tooth.

*Pedipalp:* Tibia 0.9 long, 0.7 wide, 1.3; cymbium 0.6 long, 0.4 wide. Pedipalp tibia stout, pear-shaped and bulbous. Cymbium rectangular, with prolateral and retrolateral apical expansions. The bulb is globular. The embolus originates retrolaterally from the bulb. The mid portion is broad, tapering to the tip at around four-fifths of the length of the embolus; tip elongated, hooked and sinuous apically (Figure 18a–c).

**Variation.** Carapace lengths of males examined (*n* = 10) ranged from 2.9–3.8 (mean 3.4); colouration was consistent among the specimens examined. For variation in leg macrosetae, see Table S3.

**Distribution.** Only known from the Pitjantjatjara Lands in the arid NW of South Australia (Figure 19).

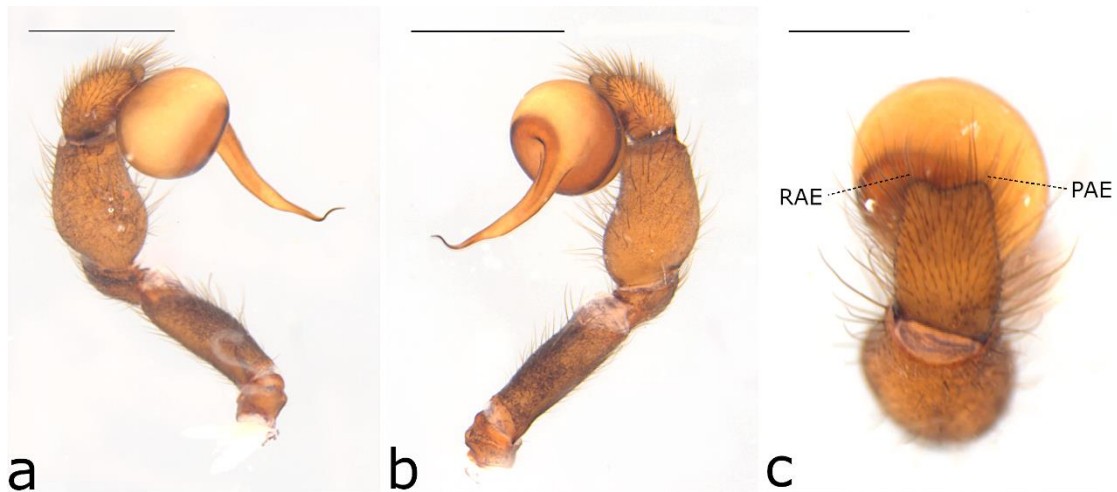

**Figure 18.** *Ariadna formosa* sp. nov. ♂ holotype (SAMA NN10565) from Indulkana (SA): (**a**) left pedipalp, prolateral view; (**b**) same, retrolateral view; (**c**) same, cymbium. Scale bar = 1 mm.

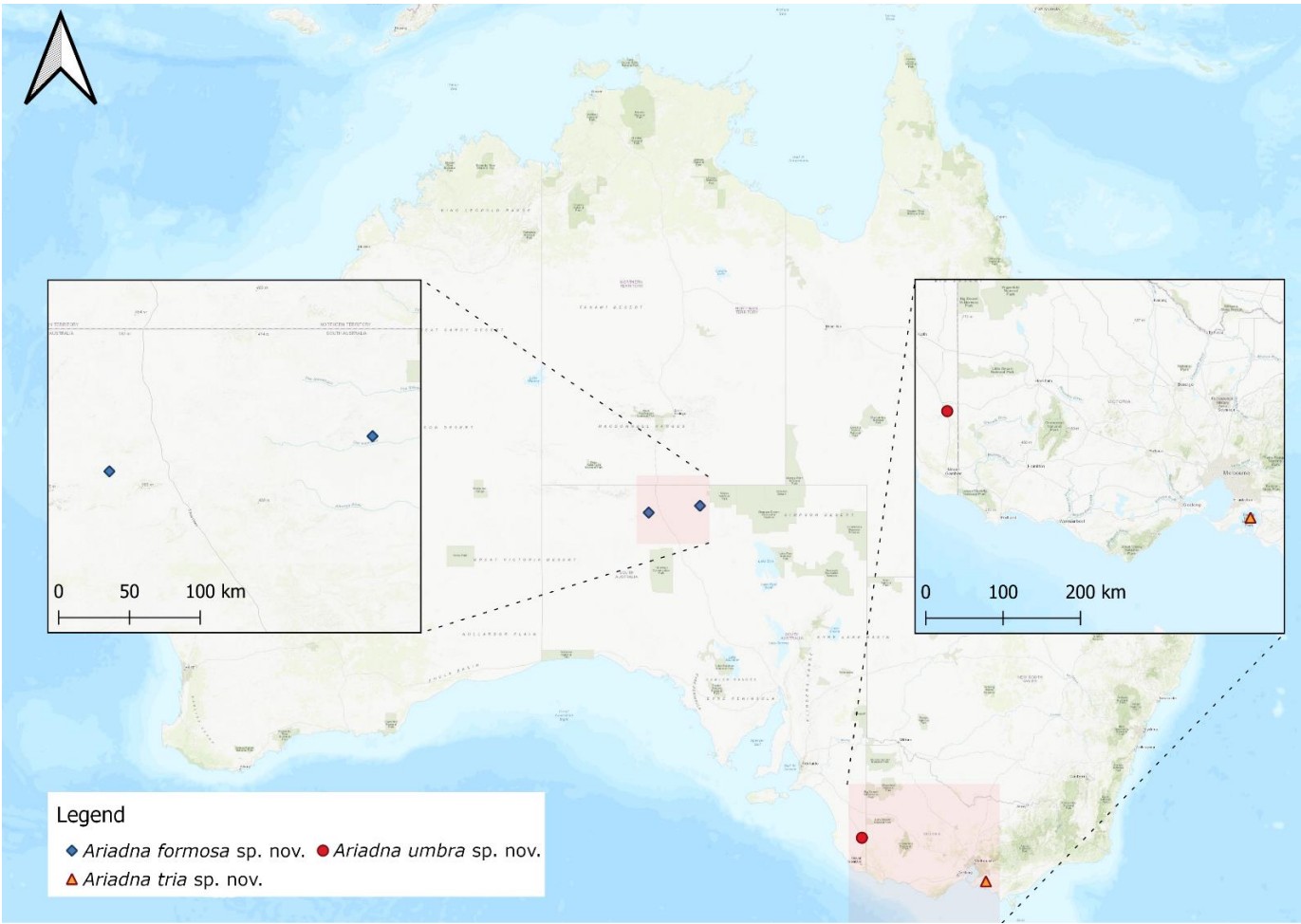

**Figure 19.** Map showing the currently known distribution of *A. formosa* sp. nov., *A. tria* sp. nov. and *A. umbra* sp. nov.

### *Ariadna tria* **sp. nov.**

Figures 19, 20a–h and 21a–c

urn:lsid:zoobank.org:act:15496011-C171-465A-8D6B-BE91D99645EF

**Type material.** *Holotype* ♂ AUSTRALIA: *Victoria:* French Island, 38.35° S, 145.35° E,2016, coll. Zoos Victoria staff (MV K-14905).

**Etymology.** The specific epithet a Latin adjective 'tria' meaning three and referring to the three macrosetae that form the preening comb of this species.

**Diagnosis.** Of the species in the *formosa* species group, *A. tria* sp. nov. can be separated from *A. formosa* sp. nov. and *A. umbra* sp. nov. by the shape of the cymbium apically, which is squared off and blunt, with no apical extensions in *A. tria* sp. nov. but has prolateral and retrolateral apical extensions in these other species (Figure 21c vs. Figure 18c and Figure 23c).

**Description.** ♂ (based on holotype; MV K-14905). Total length 6.0.

*Cephalothorax:* 3.2 long, 1.9 wide, 1.4 high. Carapace dark red-brown, lighter posteriorly, with scattered dark setae. Sternum dark golden-brown, dark brown pre-coxal triangles. Maxillae dark yellow-brown, white apically, labium dark brown, paler apically, chelicerae dark red-brown (Figure 20a–c). Carapace elongate oval, narrowing anteriorly; fovea a shallow indented pit (Figure 20a). Labium elongate, about three-quarters length of maxillae. Sternum elongate oval, with scattered setae and with pre-coxal triangles and with small, indistinct rounded intercoxal extensions (Figure 20b). Flattened in lateral view, highest point is posterior eye row (Figure 20c). Eye group 0.7 wide, occupying 0.7 of the width

of the carapace anteriorly, eyes large and elevated on mounds. Posterior eye row slightly recurved (Figure 20d).

*Abdomen:* 2.8 long. Dorsally uniform dark slate grey; ventrally pale grey-cream, with cream book lungs; with dense red-brown setae (Figure 20a–c).

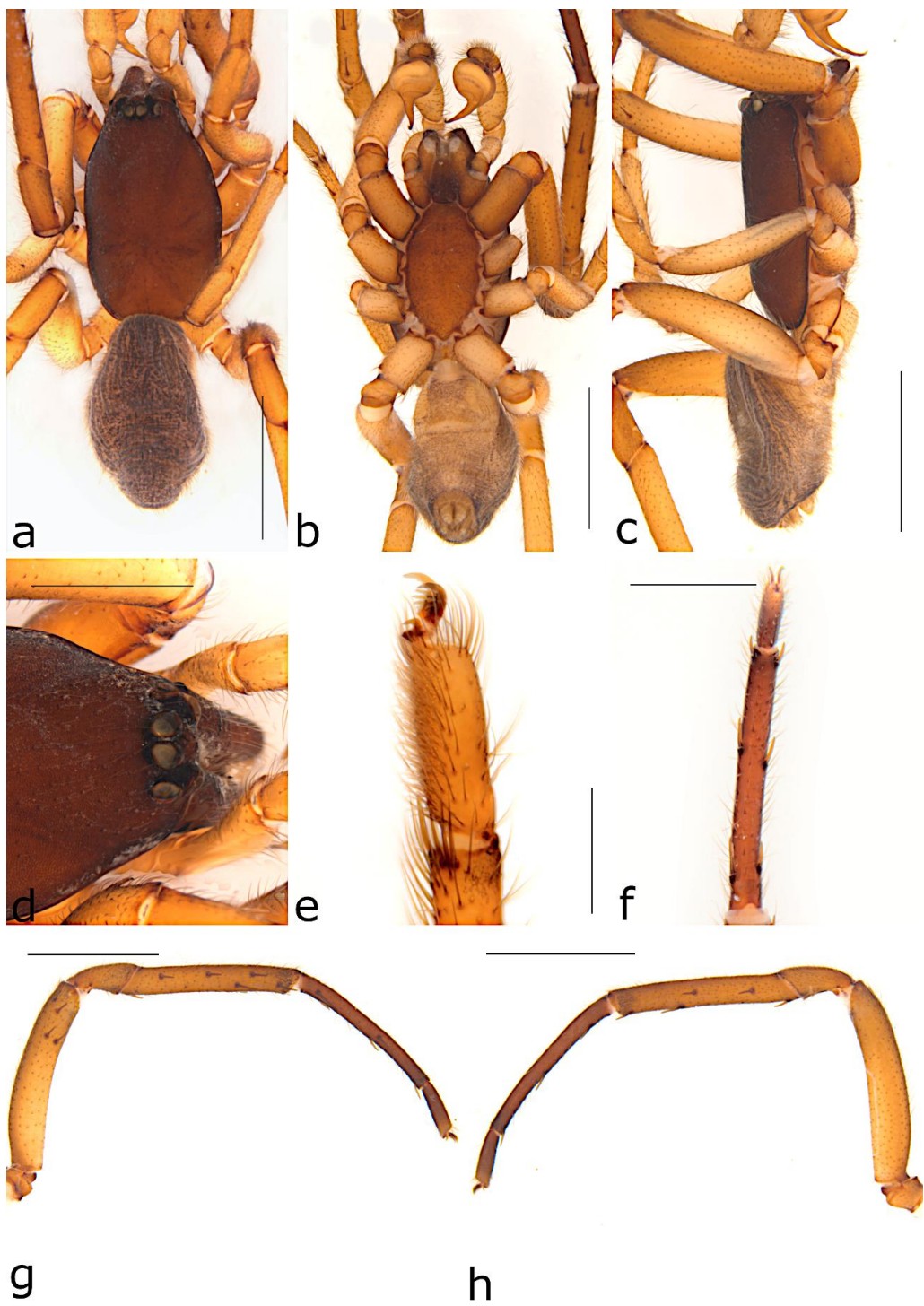

**Figure 20.** *Ariadna tria* sp. nov. ♂ holotype (MV K-14905) from French Island (Vic): (**a**) habitus, dorsal view; (**b**) same, ventral view; (**c**) samIlateral view; (**d**) eyes, dorsal view; (**e**) left metatarsus IV, preening comb, retrolateral view; (**f**) left leg I, ventral view; (**g**) same, prolateral view; (**h**) same, retrolateral Iw. Scale bars (**a**–**c**) = 2 mm, (**d**) = 1 mm, (**e**) = 0.5 mm, (**f**) = 1 mm and (**g**,**h**) = 2 mm.

*Legs*: Leg length ratio I > II > IV > III. Leg I 8.6: femur 2.6, patella 0.8, tibia 2.4, metatarsus 2.2, tarsus 0.6. Leg II 8.3: 2.5, 1.0, 2.1, 2.0, 0.7. Leg III 6.3: 1.9, 0.8, 1.4, 1.5, 0.7. Leg IV 8: 2.5, 1.1, 2.1, 1.6, 0.7. Legs pale golden-orange-brown, legs I and II darker orange-brown. Metatarsus I straight, without apophyses (Figure 20f–h). Femur I slightly bowed. Tarsus I-IV straight in lateral view, tarsi II–IV with sparse scopulate setae. Macrosetae: Leg I Femur dp3ap; tibia dp1-1-1, p1-1ap, vr1-1-1ap, r1-1; metatarsus vp1-1-1ap, vr1-1-1ap. Leg II femur dp1ap; tibia p1-1-1, vp1ap, vr1-1-1ap; metatarsus vp1-1-1ap, vr1-1-1 (small)-1ap. Leg IV retrolateral distal preening comb with three macrosetae, equal in length (Figure 20e). STC 1 with around 8 teeth, inferior tarsal claw with a long, well-defined tooth.

*Pedipalp*: Tibia 0.7 long, 0.4 wide, 1.8; cymbium 0.5 long, 0.4 wide. Tibia slender, elongate, expanded proximally, narrowing apically and prolaterally with two thickened setae apically (Figure 21a). Cymbium square in shape and blunt apically, with no obvious apical extensions. Bulb globular, embolus originating ventrally, mid portion broad until around half of the embolus length, from where it narrows only slightly into a broad, curved tip. (Figure 21a–c).

**Distribution.** Only record is from an unknown locality on French Island, a small island located SE of Melbourne, off the southern coast of mainland Victoria (Figure 19).

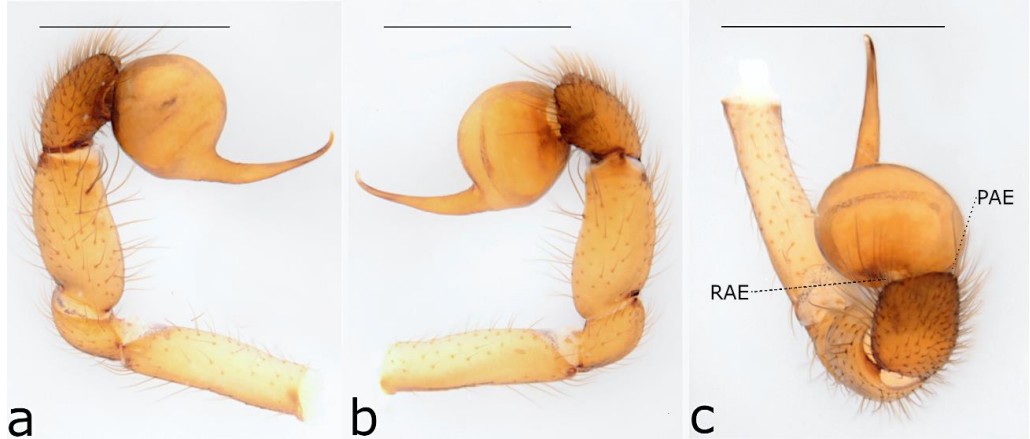

**Figure 21.** *Ariadna tria* sp. nov. ♂holotype (MV K-14905) from French Island (Vic): (**a**) left pedipalp, prolateral view; (**b**) same, retrolateral view; (**c**) same, cymbium. Scale bar = 1 mm.

### *Ariadna umbra* sp. nov.

Figure 19, Figure 22a–h, Figure 23a–c, Figures 24a–h and 25a,b
urn:lsid:zoobank.org:act:B7CE04E9-F64D-43F5-8A70-CB24621191C0

**Type material.** *Holotype* ♂ AUSTRALIA: *South Australia*, Naracoorte Cave Park Cabins 37.04985° S, 140.80821° E, night collection, roaming on bark of stringybark c. 2 m above ground of trees near tube webs, 19 Jul 2020, coll. Matthew Shaw (SAMA NN30621).

*Paratypes:* 1♀: *South Australia*, Naracoorte Cave Park Cabins 37.04985° S, 140.80821° E, removed from tube webs in stringybark trunk, c. 2 m above ground of tree near tube webs, 19 July 2020, coll. Matthew Shaw (SAMA NN30624); 1♀same data (SAMA NN30622).

**Other material examined.** 1♀*South Australia*, Naracoorte Cave Park Cabins 37.04985° S, 140.80821° E, removed from tube webs in stringybark trunk, 19 July 2020, coll. Matthew Shaw (SAMA NN30623).

**Etymology.** The specific epithet is a Latin noun meaning 'shadow' and referring to the dark colour of the male and female of this species.

**Diagnosis.** Of the other species in the *formosa* species group, *A. umbra* sp. nov. can be differentiated from *A. tria* sp. nov. by the shape of the cymbium apically, which is blunt and squared in *A. tria* sp. nov. but has distinct prolateral and retrolateral apical extensions in *A. umbra* sp. nov. (Figure 23c vs. Figure 21c). The species can be separated from *A. formosa* sp. nov. by the shape of the embolus, which is elongate and hooked apically

in *A. formosa* sp. nov. but is thickened and roundly curved in *A. umbra* sp. nov. (Figure 23a,b vs. Figure 18a,b).

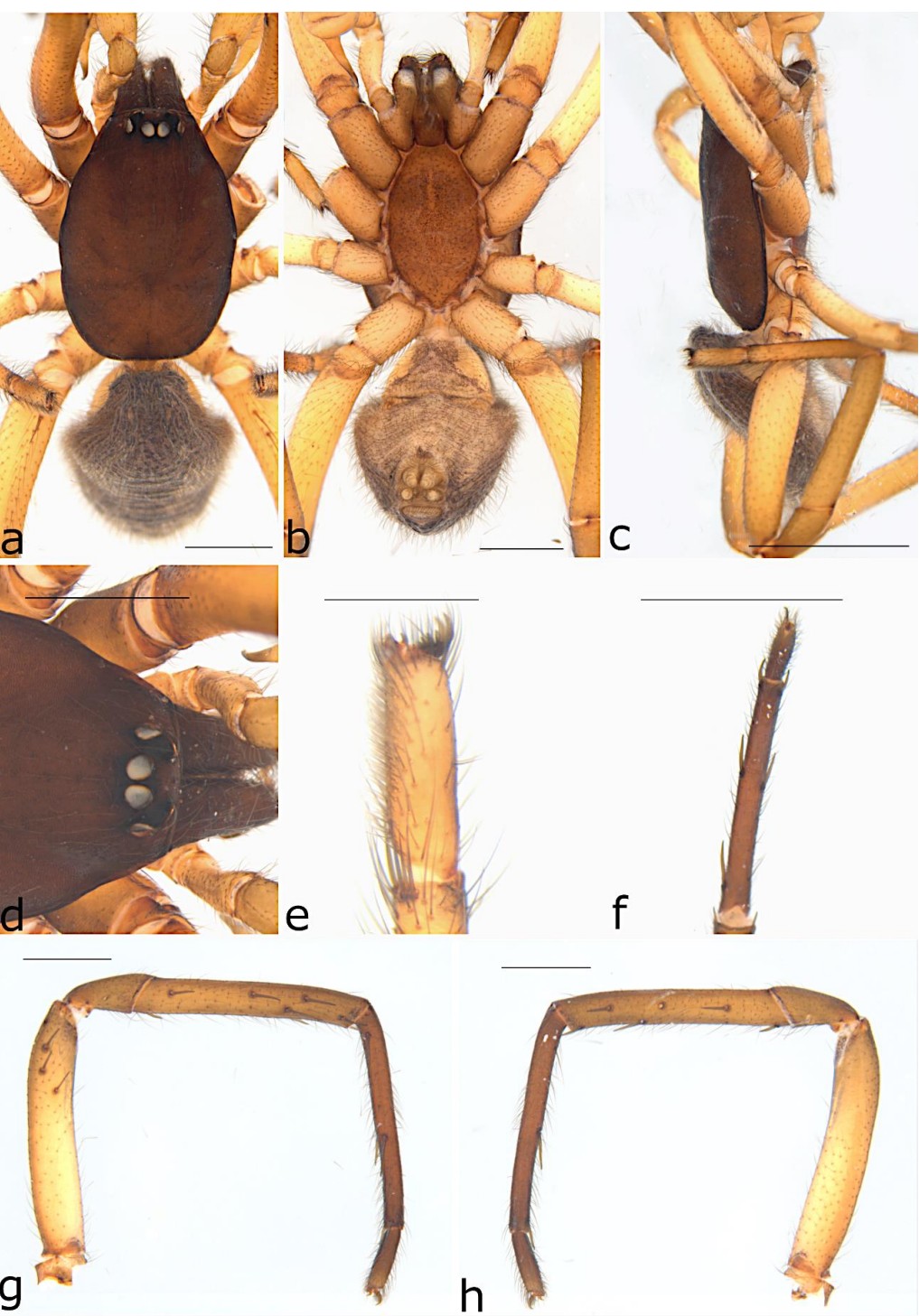

**Figure 22.** *Ariadna umbra* sp. nov. ♂holotype (SAMA NN30621) from Naracoorte (SA): (**a**) habitus, dorsal view; (**b**) same, ventral view; (**c**) same, lateral view; (**d**) eyes, dorsal view; (**e**) left metatarsus IV, preening comb, retrolateral view; (**f**) left leg I, ventral view; (**g**) same, prolateral view; (**h**) same, retrolateral view. Scale bars (**a**–**d**) = 1 mm, (**e**) = 0.5 mm, (**f**) = 2 mm and (**g,h**) = 1 mm.

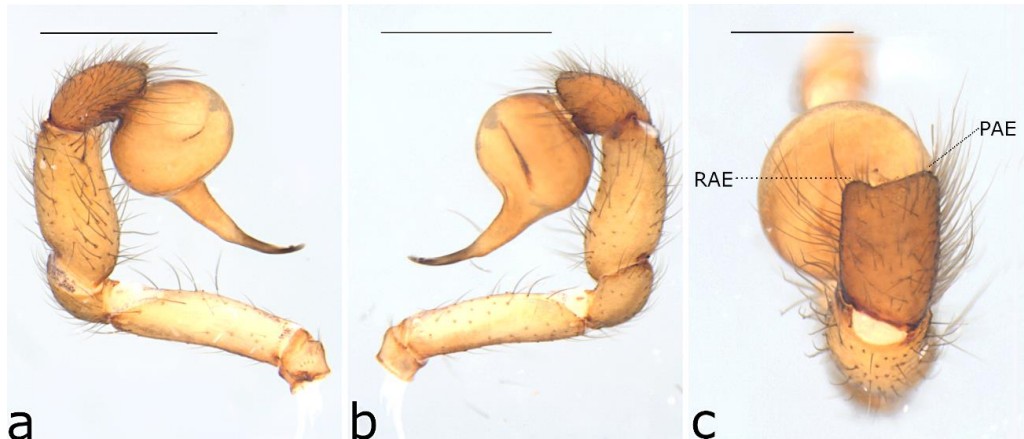

**Figure 23.** *Ariadna umbra* sp. nov. ♂holotype (SAMA NN30621) from Naracoorte (SA): (**a**) left pedipalp, prolateral view; (**b**) same, retrolateral view; (**c**) same, cymbium. Scale bar (**a**,**b**) = 1 mm and (**c**) = 0.5 mm.

**Description.** ♂(based on holotype; SAMA NN30621). Total length 5.0.

*Cephalothorax:* 2.8 long, 2.0 wide, 1.3 high. Carapace dark brown, with small scale-like markings and with scattered dark setae, arranged in longitudinal rows from fovea to eye group. Sternum dark red-brown, with a lighter median stripe anteriorly; darker brown in pre-coxal triangles. Maxillae orange-brown, white apically, labium red-brown, chelicerae dark red-brown. Figure 22a–h. Carapace rounded oval, narrowing anteriorly; lateral edges gently undulating; fovea a shallow indented pit, with darker markings radiating out from fovea (Figure 22a). Labium about three-quarters the length of maxillae. Sternum elongate oval, with scattered setae and with pre-coxal triangles (Figure 22b). Carapace flat in lateral view, highest just posterior to fovea, chelicerae semi-porrect (Figure 22c). Eye group 0.7 wide, occupying 0.6 of the width of the carapace anteriorly; posterior eye row straight (Figure 22d).

*Abdomen:* 2.2 long. Dorsally dark grey, uniform in colour, ventrally pale slate-grey, with cream book lungs; with covering of fine brown setae (Figure 22a–c).

*Legs:* Leg length ratio I > II > IV > III. Leg I total 10: femur 3.2, patella 1.0, tibia 2.5, metatarsus 2.5, tarsus 0.8. Leg II 8.6: 2.6, 1.1, 2.1, 2.0, 0.8. Leg III 5.9: 2.0, 0.8, 1.4, 1.0, 0.7. Leg IV 7.8: 2.5, 1.0, 2.1, 1.6, 0.6. Legs golden pale orange-brown. Leg I metatarsus and tarsus darker brow. Femur I slightly bowed in dorsal view, metatarsus I gently sinuous in ventral view, with no apophyses (Figure 22f–h). Tarsi short, tarsi I–IV straight; tarsi II–IV with sparse distal ventral scopulate setae. Macrosetae: Leg I: femur dp3, d1ap; tibia p1-1-1, vp1-1ap, vr1-1-1ap, r1-1-1; metatarsus vp1(long)-1(long)-1ap, vr1(long)-1ap. Leg II: femur dp1ap, dr1ap; tibia p1-1-1, vp1ap, vr1-1-1-1ap; metatarsus vp1-1-1-1ap, vr1-1-1ap. Leg IV: femur d1-1(basal). Retrolateral distal preening comb with 3 macrosetae, equal in length (Figure 22e). STC I, II with 8 teeth, ITC with short, stout tooth.

*Pedipalp:* Tibia 0.8 long, 0.5 wide, 1.6; cymbium 0.6 long, 0.4 wide. Tibia elongate slightly curved, prolaterally with one or two thickened setae apically. Cymbium with a defined apical prolateral triangular extension and a smaller but well-defined retrolateral extension. Bulb globular, embolus originating ventroretrolaterally, mid portion broad until three-quarters of the length of the embolus, where it tapers apically to the tip; the tip is broad and gently curved at the apex (Figure 23a–c).

**Description.** ♀(based on paratype; SAMA NN30624). Total length 10.4.

*Cephalothorax:* 4.7 long, 2.8 wide, 2.6 high. Carapace dark red-brown, darker anteriorly. Sternum orange-brown. Maxillae darker orange-brown, paler apically. Labium red-brown, chelicerae darker red-brown. Figure 24a–c. Carapace elongated oblong, with undulating lateral edges and forming a broad neck anteriorly; fovea a shallow indented pit. (Figure 24a). Labium elongate, about four-fifths length of maxillae. Sternum elongate oval, with long, scattered dark setae; pre-coxal triangles indistinct (Figure 24b. Carapace domed in lateral

view, highest just prior to eye group (Figure 24c). Eye group 1.0 wide, occupying 0.6 of the width of the carapace anteriorly; posterior eye row straight (Figure 24d).

*Abdomen:* 5.7 long. Dorsally dark grey with mottled lighter colouring; ventrally dark grey, with paler white striations, pale colour around book lungs and epigynal plate; with a covering of fine dark brown setae (Figure 24a–c).

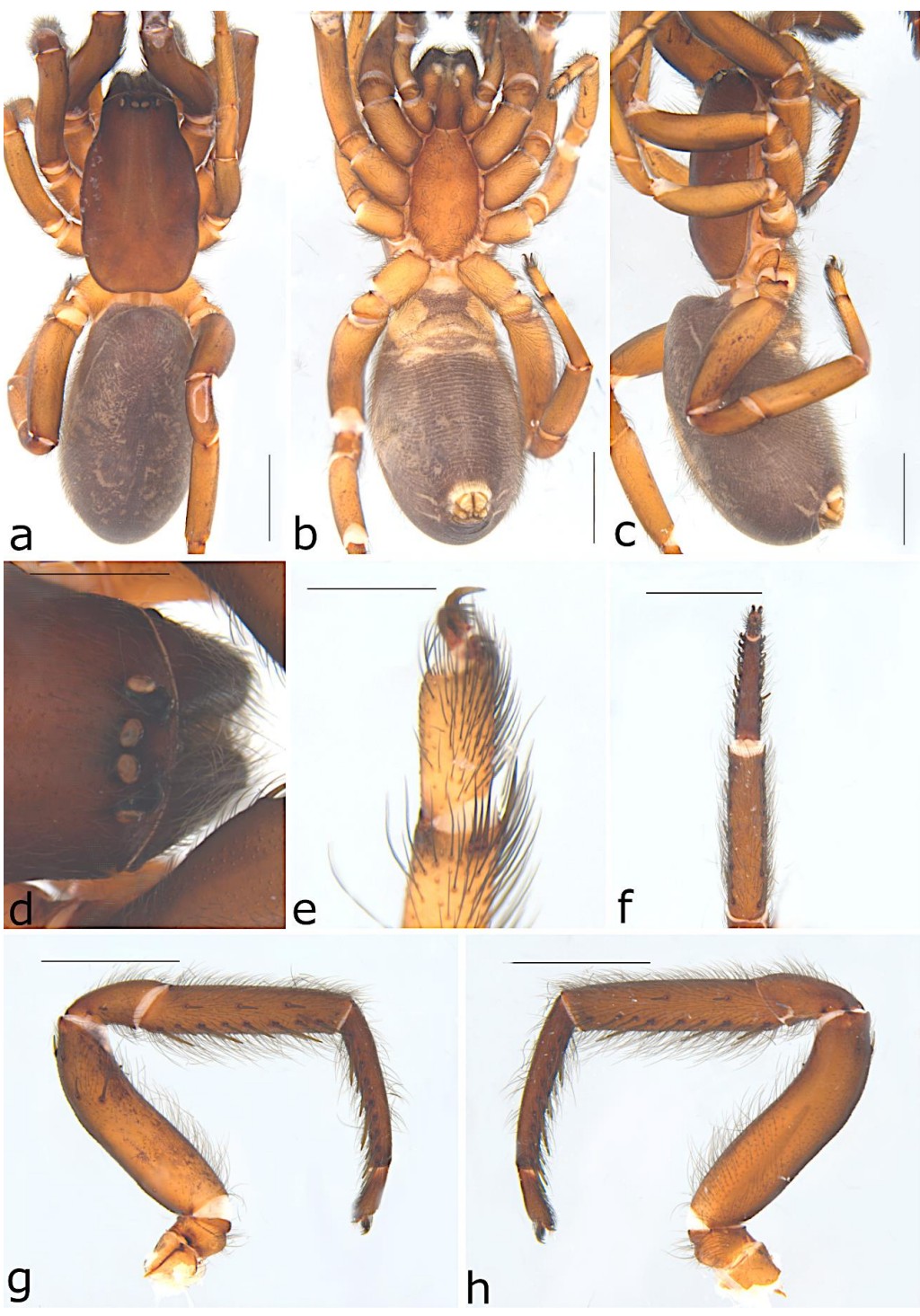

**Figure 24.** *Ariadna umbra* sp. nov. ♀paratype (SAMA NN30624) from Naracoorte (SA): (**a**) habitus, dorsal view; (**b**) same, ventral view; (**c**) same, lateral view; (**d**) eyes, dorsal view; (**e**) left metatarsus IV, preening comb, retrolateral view; (**f**) left leg I, ventral view; (**g**) same, prolateral view;(**h**) same, retrolateral view. Scale bars (**a**–**d**) = 2 mm, (**e**) = 0.5 mm and (**f**–**h**) = 2 mm.

*Legs:* Leg length ratio I > II > IV > III. Leg I total 10.4: femur 3.1, patella 1.3, tibia 2.7, metatarsus 2.3, tarsus 1.0. Leg II 10.3: 3.1,1.4, 2.6, 2.3, 0.9. Leg III 7.8: 2.5, 1.2, 1.8, 1.5, 0.8. Leg IV 9.8: 3.2, 1.6, 2.5, 1.6, 0.9. Legs orange-brown, legs I and II darker red-brown. Femur I strongly bowed in dorsal view. (Figure 24f–h). Tarsi short, stout. Tarsi I, II broader at apex than at the base. All leg segments covered in dense, black setae. Macrosetae: Leg I: femur dp3ap, d1ap, dr1ap; tibia p1-1-1, vp1-1-1-1-1-1-1-1, vr1-1, r1-1-1-1-1-1-1-1; metatarsus v2-2-2-2-2-2-2-2-2-2. Leg II: femur dp1-1ap, d1ap, dr1ap; tibia p1-1-1, vp1-1-1-1-1ap; vr1-1-1-1-1-1-1-1ap; metatarsus v2-2-2-2-2-2-2-2-1-1. Leg IV: metatarsus v1ap. Retrolateral distal preening comb with three macrosetae: two shorter, one longer (Figure 24e). STC I and II with six teeth, ITC with a small, stout tooth.

*Pedipalp:* Densely covered in dense black setae, with numerous macrosetae prolaterally.

*Genitalia:* (based on paratype; SAMA NN30622) The anterior receptaculum is bi-lobed, both lobes around the same size. The dorsal lobe is sinuous, whereas the ventral lobe is more or less straight. (Figure 25a–c).

**Variation.** Carapace lengths of females examined (*n* = 3) ranged from 4.7 to 4.8 (mean 4.7); colouration was consistent among the specimens examined. For variation in leg macrosetae, see Table S3.

**Distribution.** This species is only known from the type locality, in Naracoorte, SE South Australia (Figure 19).

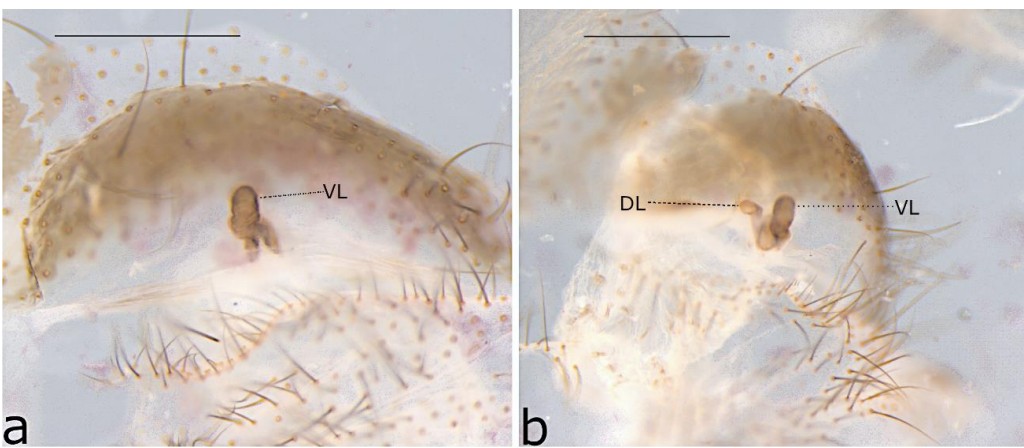

**Figure 25.** *Ariadna umbra* sp. nov. ♀paratype (SAMA NN30622) from Naracoorte (SA): anterior receptaculum (**a**) ventral view; (**b**) same, lateral view. Scale bar = 0.5 mm. DL = dorsal lobe, VL = ventral lobe.

### *Ariadna arenacea* **sp. nov.**

Figure 26a–h, Figures 27a–c and 28

urn:lsid:zoobank.org:act:9C8D9D17-E203-420E-B1FA-70E1A8FB91D4

**Type material.** *Holotype* ♂ AUSTRALIA: *South Australia*, NW region, 173 km SSW Wataru, 28.50667° S, 129.26194° E, very open low mallee, *Eucalyptus youngiana* over *Dicrastylis*, *Newcastelia* & *Triodia*, pitfall trap, 19–23 October 2006, Sandy Desert Survey, SER008 (SAMA NN30585).

**Etymology.** The specific epithet is Latin adjective meaning 'sandy' and referring to the Sandy Desert Survey in which this species was collected.

**Diagnosis.** Of the species with only one well-defined apophysis on metatarsus I, *A. arenacea* sp. nov. can be distinguished from *A. flavescens* sp. nov. by the shape of the embolus, which is thickened until midway from where it gradually narrows to a curved tip in *A. flavescens* sp. nov., whereas, in *A. arenacea* sp. nov., it is thickened until three-quarters of its length, from where it narrows abruptly to a sinuous tip (Figure 27a,b vs. Figure 38a,b). It can be differentiated from *A. inflata* sp. nov. by the relative length of the embolus to the pedipalp bulb, where, in *A. arenacea* sp. nov., the embolus is relatively short, being around the same length as the width of the bulb, whereas, in *A. inflata* sp. nov., the embolus

is elongated, clearly longer than the width of the bulb (Figure 27a,b vs. Figure 41a,b). It can be further separated by the pedipalp cymbium, which has well-defined, angular PAE and low, not well-defined RAE in *A. arenacea* sp. nov., whereas, in *A. inflata* sp. nov. and *A. flavescens* sp. nov., both the PAE and RAE are well-pronounced (Figure 27c vs. Figure 38c, Figure 41c).

**Description.** ♂ (based on holotype; SAMA NN30585). Total length 6.7.

*Cephalothorax:* 3.4 long, 2.1 wide, 1.9 high. Carapace dark red-brown, lighter colour posteriorly. Sternum golden orange-brown, with darker orange-brown pre-coxal triangular extensions. Maxillae orange-brown, white apically, labium darker orange-brown, chelicerae red-brown. (Figure 26a–c). Carapace rounded oval, narrowing anteriorly and with rows of sparse, setae, running longitudinally from fovea to eye group. Fovea a shallow longitudinal furrow (Figure 26a). Labium about $\frac{3}{4}$ length of the maxillae; sternum elongated oval with pre-coxal triangles and with smaller, rounded intercoxal extensions, with scattered, brown setae (Figure 26b). In lateral view, the carapace dome was highest midway between the eye group and fovea (Figure 26c). Eye group 0.7 wide, occupying 0.7 of the width of the carapace anteriorly; posterior eye row slightly recurved (Figure 26d).

*Abdomen:* 3.3 long. Dorsally dark slate-grey, ventrally cream, with a covering of fine dark red-brown setae (Figure 26a–c).

*Legs:* Leg length ratio I > II > IV > III. Leg I total 9.1: femur 3.0, patella 1.1, tibia 2.2, metatarsus 2.0, tarsus 0.8. Leg II 9: 2.7, 1.2, 2.1, 2.0, 1.0. Leg III 6.6: 2.1, 0.8, 1.4, 1.4, 0.9. Leg IV 7.8: 2.7, 0.8, 1.7, 1.7, 0.9. Legs orange-brown, legs I and II darker brown than III or IV. Femur I slightly bowed in dorsal view. Metatarsus I straight in ventral view, with two opposing apophyses; measured proximally to distally, the prolateral apophysis is situated at 0.63 the length of the metatarsus and the retrolateral apophysis at 0.51 of the length of the metatarsus; prolateral apophysis pronounced and bearing a short, stout blunt macroseta; retrolateral apophysis a low, indistinct mound and bearing an elongate macroseta, directed parallel to the plane of the metatarsus (Figure 26f–h). Tarsi I slightly curved ventrally in lateral view, slightly broader at the apex than at the base. Tarsi II–IV straight in lateral view. Macrosetae: Leg I: femur dp3ap, d1-1, dr1ap; tibia p1-1-1-1-1, vp2-1-1ap, vr1-1-1-1ap, r1-1-1-1-1-1; metatarsus p1(Long)-apophysis-1ap, r1-apophysis-1ap. Leg II: femur dp2ap, d1-1, dr1ap; tibia p1-1-1-1-1, vp1ap, vr2-1-1-1ap, r1-1-1-1-1-1; metatarsus p1-1-1ap, vr1-1-1ap, r-1-1-1ap. Leg IV: retrolateral distal preening comb with three macrosetae, more or less equal in length (Figure 26e). STC I, II with nine teeth, ITC with a small tooth.

*Pedipalp:* Tibia 0.7 long, 0.5 wide, 1.4; cymbium 0.7 long, 0.5 wide. Pedipalp tibia oblong, curved ventrally. Cymbium rectangular, with a pointed, triangular PAE and a small, blunt RAE. The bulb is large, globular. The embolus originates retrolaterally from the bulb; the mid portion is broad until around three-quarters of the length of the embolus, from where it tapers apically to tip; tip hooked, then very elongate and sinuous apically. (Figure 27a–c).

**Distribution:** This species is only known from the Sandy Desert in the arid NW of South Australia (Figure 28).

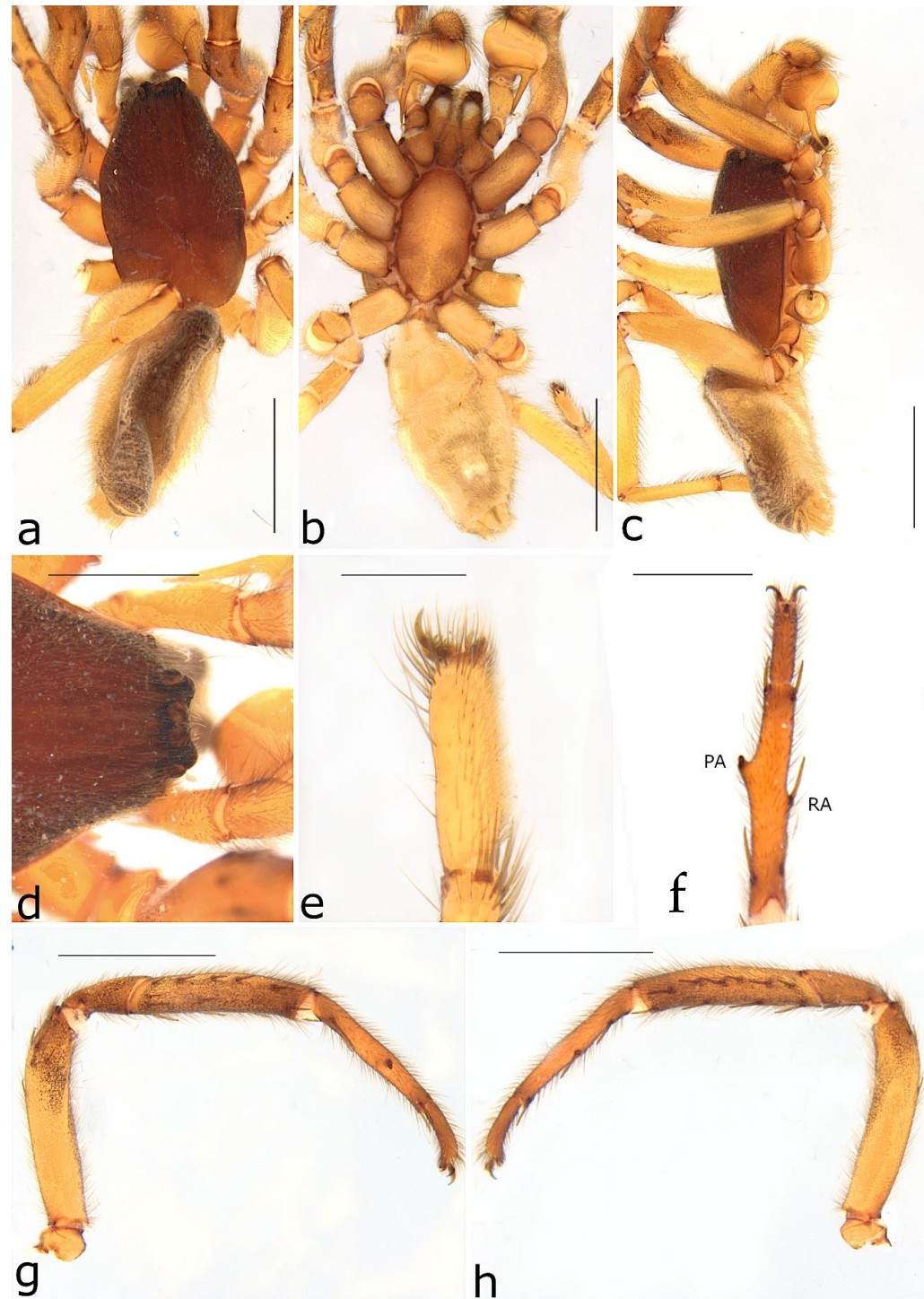

**Figure 26.** *Ariadna arenacea* sp. nov. ♂ holotype (SAMA NN30585) from North Wartaru (SA): (**a**) habitus, dorsal view; (**b**) same, ventral view; (**c**) same, lateral view; (**d**) eyes, dorsal view; (**e**) left metatarsus IV, preening comb, retrolateral view; (**f**) left leg I, ventral view; (**g**) same, prolateral view; (**h**) same, retrolateral view. Scale bars (**a**–**c**) = 2 mm, d, (**f**) = 1 mm, (**e**) = 0.5 mm and (**g**,**h**) = 2 mm.

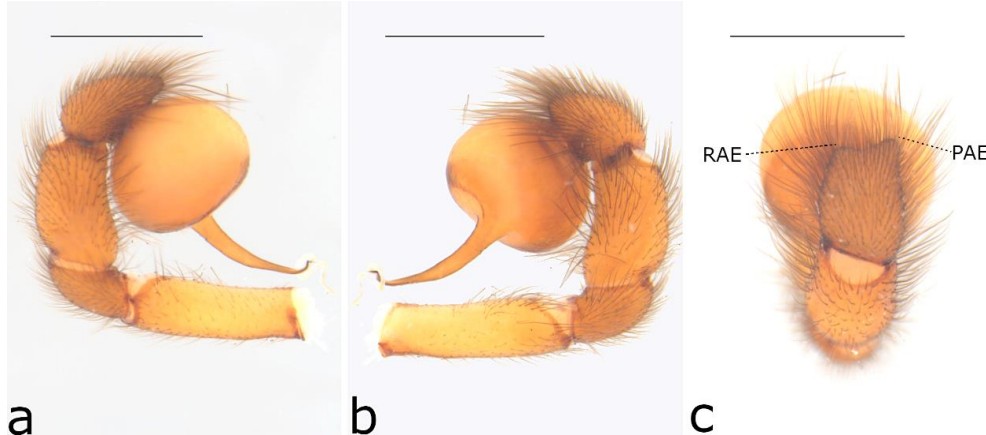

**Figure 27.** *Ariadna arenacea* sp. nov. ♂ holotype (SAMA NN30585) from north Wartaru (SA): (**a**) left pedipalp, prolateral view; (**b**) same, retrolateral view; (**c**) same, cymbium. Scale bar = 1 mm.

**Figure 28.** Map showing the currently known distribution of *A. arenacea* sp. nov., *A. bellatula* sp. nov., *A. curvata* sp. nov., *A. deserta* sp. nov. and *A diucrura* sp. nov.

### *Ariadna bellatula* **sp. nov.**

Figures 28, 29a–h and 30a–c

urn:lsid:zoobank.org:act:B3E55F69-7665-414D-A9C4-25F5B41DCEF1

**Type material.** *Holotype* ♂ AUSTRALIA: *South Australia*, 23.4 km SSW Mundy Dam, 26.6733° S, 133.01667° E, *Sclerolaena uniflora* low open shrubland, pitfall trap, 12–16 August 1998, Pitjantjatjara Lands ITY0501 (SAMA NN10577).

**Etymology.** The specific epithet is a Latin adjective meaning 'neat' and referring to the compact body form of this species.

**Diagnosis.** Of the species with two well-pronounced apophyses situated in the medial section of metatarsus I, *A. bellatula* sp. nov. can be differentiated from *A. curvata* sp. nov. by the retrolateral apophysis of *A. bellatula* sp. nov. being only slightly raised from the metatarsus, whereas, in *A. curvata* sp. nov., the retrolateral apophysis is strongly raised (Figure 29f vs. Figure 31f). It can be separated from *A. rutila* sp. nov. by the mid portion of the embolus, which is distinctly thickened at around half the embolus length, from which it tapers to a sinuous tip in *A. bellatula* sp. nov. but is a smooth curve in *A. rutila* sp. nov. (Figure 30a,b vs. Figure 51a,b). Molecular analyses place *A. bellatula* sp. nov. as being closest to *A. una* sp. nov.; however, the two species differed morphologically and so here are treated as different species; relative position of the apophyses on metatarsus I, *A. una* sp. nov. has the prolateral apophysis at 0.45 and retrolateral apophysis at 0.27 of metatarsus length, *A. bellatula* sp. nov. has the prolateral apophysis at 0.54 and retrolateral apophysis at 0.44 of the metatarsus length; both apophyses subequal in size in *A. una* sp. nov., but the retrolateral apophysis is smaller than the prolateral apophysis in *A. bellatula* sp. nov. (Figure 29f vs. Figure 62f); the pedipalp cymbium of *A. una* sp. nov. has an enlarged PAE compared to the RAE, whereas the PAE and RAE are subequal in *A. bellatula* sp. nov. (Figure 30c vs. Figure 63c).

**Description.** ♂ (based on holotype; SAMA NN10577). Total length 5.1.

*Cephalothorax:* 2.7 long, 1.9 wide, 1.3 high. Carapace orange-brown, lighter colour posteriorly. Sternum pale orange-brown, with darker orange-brown pre-coxal triangular extensions. Maxillae pale orange-brown, white apically, labium darker orange-brown, chelicerae red-brown (Figure 29a–c). Carapace rounded oval, narrowing anteriorly and with rows of sparse, setae, running longitudinally from fovea to eye group; fovea a shallow longitudinal furrow (Figure 29a). Labium about three-quarters length of maxillae. Sternum elongated oval with pre-coxal triangles and with smaller, rounded intercoxal extensions, with scattered, brown setae (Figure 29b). In lateral view, carapace gently domed, highest midway between eye group and fovea (Figure 29c). Eye group 0.6 wide, occupying 0.6 of the width of the carapace anteriorly; posterior eye row slightly recurved (Figure 29d).

*Abdomen:* 2.4 long. Dorsally dark slate-grey, ventrally pale grey; with a covering of fine dark red-brown setae (Figure 29a–c).

*Legs:* Leg length ratio II > I > IV > III. Leg I total 7.4: femur 2.4, patella 0.9, tibia 1.8, metatarsus 1.5, tarsus 0.8. Leg II 7.5: 2.5, 0.9, 1.7, 1.7, 0.7. Leg III 6: 1.9, 0.8, 1.3, 1.3, 0.7. Leg IV 7.2: 2.3, 0.9, 1.8, 1.5, 0.7. Legs pale orange-brown, legs I and II darker orange-brown. Femur I slightly bowed in dorsal view. Metatarsus I straight in ventral view, with two opposing apophyses; measured proximally to distally, the prolateral apophysis is at 0.54 the length of the metatarsus, and the retrolateral apophysis at 0.41 of the length of the metatarsus. The prolateral apophysis is pronounced and bears a short, stout macroseta; the retrolateral apophysis is indistinct, small and bears an elongate macroseta, directed parallel to the plane of the metatarsus (Figure 29f–h). Tarsi I slightly curved ventrally in lateral view, slightly broader at apex than at base. Tarsi II, III slightly inflated, tarsi IV inflated ventrally. Tarsi II with sparse scopulate setae apically, tarsi III, tarsi IV with scopulate setae along entire ventral edge. Macrosetae: Leg I: femur dp1ap, d1ap; tibia dp1-1-1-1-1, vp1-1-1-1ap, vr1-1(small)-1-1-1ap, r1-1-2-1-1; metatarsus p1(long)-appophysis-1ap, r-apophysis-1ap. Leg II: femur dp2, d1; tibia dp1-1, vp1ap, vr2-1-1-1-1ap, r1-1-1-1-1-1; metatarsus vp1-1-1ap, r1-1-1, vp1-1-1ap. Leg IV: retrolateral distal preening comb with four macrosetae: one longer and three shorter (Figure 29e). STC I, II with 12 teeth, ITC with a small tooth.

*Pedipalp:* Tibia 0.6 long, 0.5 wide; cymbium 0.6 long, 0.4 wide. Pedipalp tibia stout. Cymbium-squared rectangular, with small, rounded prolateral and retrolateral apical expansions. The bulb is globular. The embolus originates retrolaterally from the bulb, the

mid portion is broad until around half the length of the embolus, from where it tapers apically to tip; tip hooked and sinuous apically (Figure 30a–c).

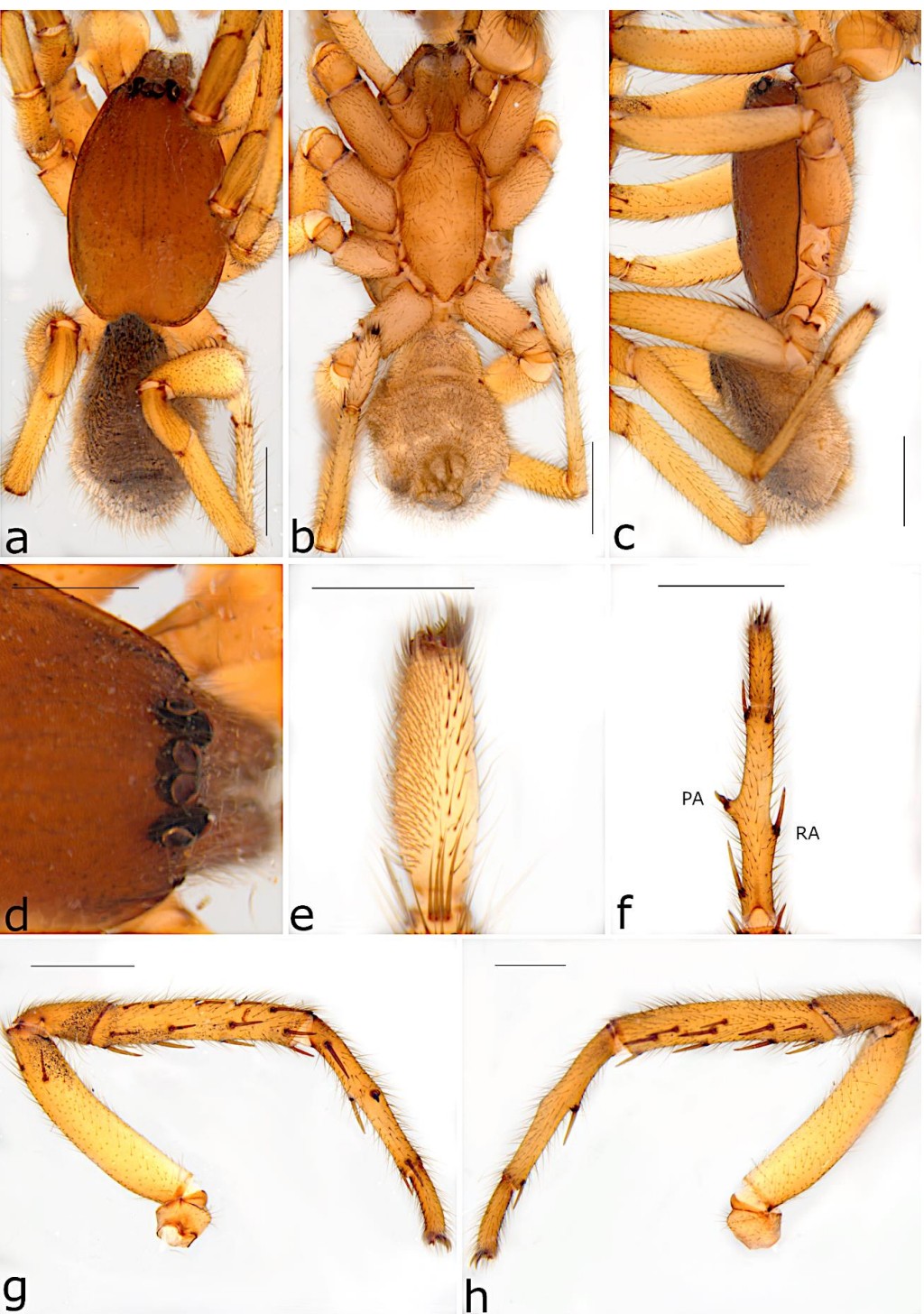

**Figure 29.** *Ariadna bellatula* **sp. nov.** ♂holotype (SAMA NN10577) from Mundy Dam (SA): (**a**) habitus, dorsal view; (**b**) same, ventral view; (**c**) same, lateral view; (**d**) eyes, dorsal view; (**e**) left metatarsus IV, preening comb, retrolateral view; (**f**) left leg I, ventral view; (**g**) same, prolateral view; (**h**) same, retrolateral view. Scale bars (**a**–**c**) = 1 mm, d, (**e**) = 0.5 mm and (**f**–**h**) = 1 mm.

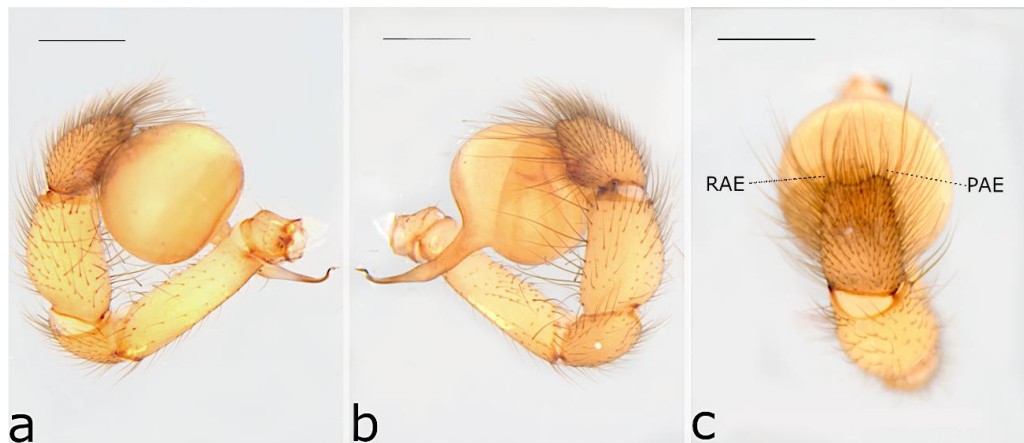

**Figure 30.** *Ariadna bellatula* sp. nov. ♂ holotype (SAMA NN10577) from Mundy Dam (SA): (**a**) left pedipalp, prolateral view; (**b**) same, retrolateral view; (**c**) same, cymbium. Scale bar = 1 mm.

**Distribution.** Only known from the holotype, found near Mundy Dam in the Pitjant-jatjara Lands of NW South Australia (Figure 28).

### *Ariadna curvata* **sp. nov.**

Figures 28, 31a–h and 32a–c

urn:lsid:zoobank.org:act:C905A125-D7D5-4C9C-B46A-E3DC164D952A

**Type material.** *Holotype* ♂ AUSTRALIA: *South Australia*, Flinders Ranges, 4 km N Arkaroola Village, 30.27778° S, 139.34694° E, grassland, *Triodia*, *Ptilotus* & *Acacia* with emergent *Eucalyptus intertexta*, micropitfall trap, 26–30 September 2009, ARK 00101 (SAMA NN30589).

**Etymology.** The specific epithet is a Latin adjective, meaning hooked and refers to the hooked embolus of the male of this species.

**Diagnosis.** Of the species with two well-pronounced apophyses situated in the medial section of metatarsus I, *A. curvata* sp. nov. can be differentiated from *A. bellatula* sp. nov. and *A. rutila* sp. nov. by the shape of the retrolateral apophysis of metatarsus I, which is only slightly raised from the metatarsus in *A. bellatula* sp. nov. and *A. rutila* sp. nov. but is distinctly raised in *A. curvata* sp. nov. (Figure 31f vs. Figure 19f and Figure 50f).

**Description.** ♂ (based on holotype; SAMA NN30589). Total length 8.5.

*Cephalothorax:* 4.2 long, 3.0 wide, 2.2 high. Carapace dark red-brown, lighter posteriorly, irregular and indistinct mottled darker patches posteriorly. Sternum golden orange-brown with darker orange-brown patches extending medially from intercoxal areas, with dark orange-brown pre-coxal triangles. Maxillae orange, white apically, labium orange-brown, chelicerae red-brown (Figure 31a–c). Carapace rounded oval, narrowing anteriorly with gently undulating lateral edges and with scattered sparse, setae; fovea a shallow indented pit (Figure 31a). Labium about four-fifths length of maxillae. Sternum elongated oval with pre-coxal triangles and with smaller, rounded intercoxal extensions, with scattered, long, black setae (Figure 31b). In lateral view, the carapace gently domed, highest midway between fovea and eye group (Figure 31c). Eye group 0.9 wide, occupying 0.6 of the width of the carapace anteriorly; posterior eye row slightly recurved. (Figure 31d).

*Abdomen:* 4.3 long. Abdomen dorsally dark violet-grey, with faint light irregular striations, ventrally cream; with a covering of fine orange-brown setae (Figure 31a–c).

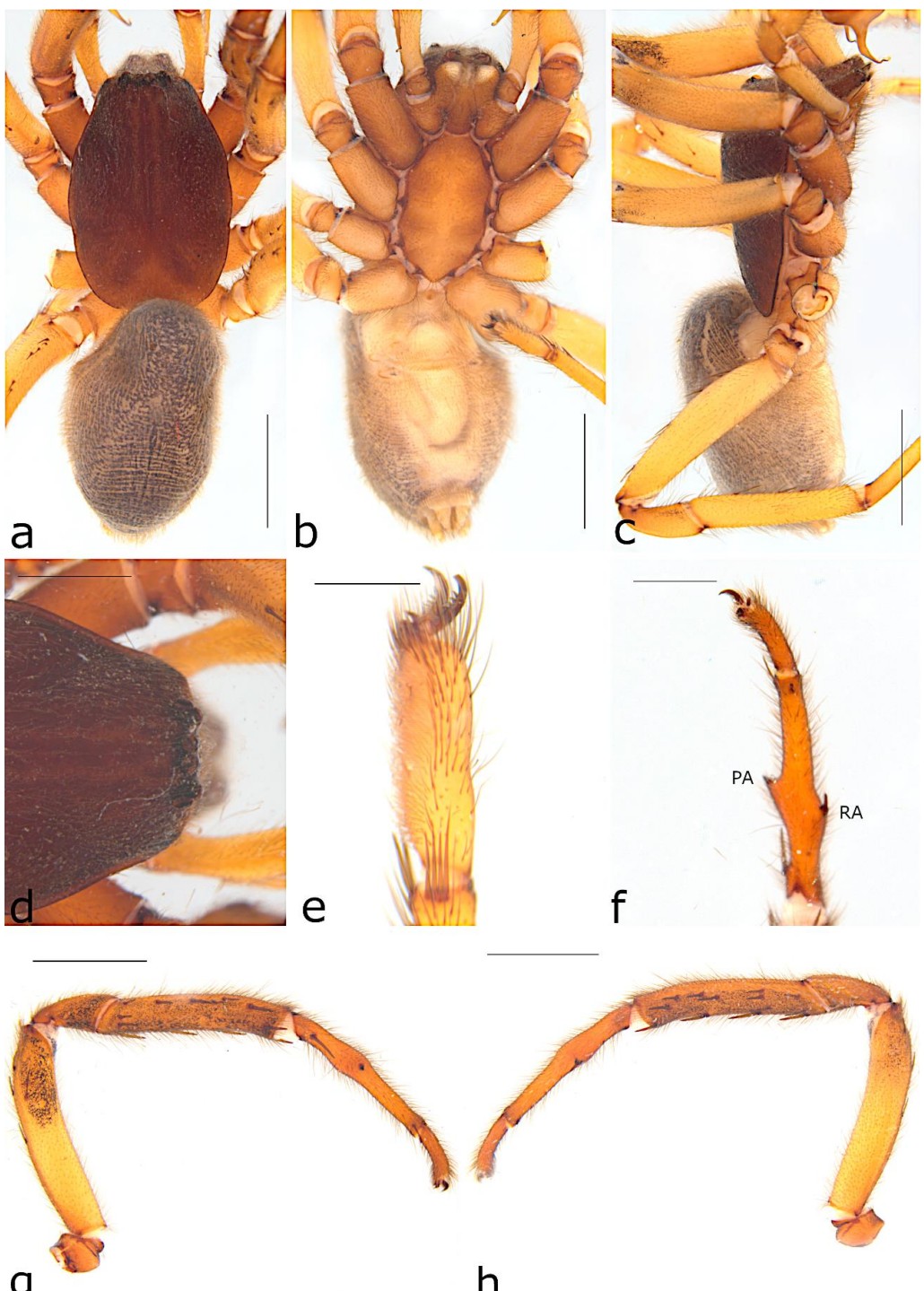

**Figure 31.** *Ariadna curvata* sp. nov. ♂ holotype (SAMA NN30589) from Arkaroola (SA): (**a**) habitus, dorsal view; (**b**) same, ventral view; (**c**) same, lateral view; (**d**) eyes, dorsal view; (**e**) left metatarsus IV, preening comb, retrolateral view; (**f**) left leg I, ventral view; (**g**) same, prolateral view; (**h**) same, retrolateral view. Scale bars (**a**–**c**) = 2 mm, (**d**,**f**) = 1 mm, (**e**) = 0.5 mm and (**g**,**h**) = 2 mm.

*Legs:* Leg length ratio I = II > IV > III. Leg I total 12.3: femur 3.5, patella 1.3, tibia 3.2, metatarsus 3.1, tarsus 1.2. Leg II 12.3: 3.5, 1.4, 3.1, 3.1, 1.2. Leg III 9: 3.1, 0.9, 1.7, 2.2, 1.1. Leg IV 11.6: 3.6, 1.4, 2.8 2.6, 1.2. Legs pale orange-brown, dark brown mottling on patella and tibia of leg I and II. Legs III and IV uniform pale orange. Femur I slightly bowed in dorsal view. Metatarsus I straight in ventral view, with two opposing apophyses; measured proximally to distally, the prolateral apophysis is situated at 0.46 of the length

of the metatarsus, and retrolateral apophysis at 0.40 of the length of the metatarsus. The prolateral apophysis bears a short, stout macroseta, and the retrolateral apophysis bears a moderately elongate macroseta, directed parallel to the plane of the metatarsus. Both apophyses are pronounced and triangular in shape, and the prolateral apophysis is broad and robust, whilst the retrolateral apophysis is narrower (Figure 31f–h). Tarsi I curved ventrally in lateral view, broader at the apex than at the base. Tarsi II–IV straight in lateral view. Macrosetae: Leg I: femur dp2ap, d1-1-1, dr21ap; tibia p1-1-1-1-1-1, vp2-1ap, vr2-2-1-1, r1-2-1-1-1-1; metatarsus p1 (long)-apophysis-1ap, r-apophysis-1ap. Leg II: femur dp1-1, d1-1-1-1, dr1; tibia p1-1-1-1-1, vp1ap, vr2-2-1-1ap, r2-1-1-1-1-1; metatarsus vp1-1-1ap, vr1-1-1-1ap, r1-1-1-1. Leg IV: femur dp1, d1-1-1-1-1-1-1; tibia p1, v1-1-1; metatarsus r1. Retrolateral distal preening comb with four macrosetae: one longer and three shorter (Figure 31e). STC I, II with 13 teeth, ITC with small tooth.

*Pedipalp:* Tibia 0.9 long, 0.7 wide; cymbium 0.5 long, 0.3 wide. Pedipalp tibia stout, with rounded ventral expansion basally. Cymbium rectangular elongate, with a pronounced rounded PAE and a smaller, rounded RAE. The bulb is globular. The embolus originates retrolaterally from the bulb and is distinctly curved, the mid portion is broad until about half of the embolus length, at which point, it widens before tapering apically to the tip; embolus tip elongates, strongly hooked and sinuous (Figure 32a–c).

**Distribution:** This species is only known from near Arkaroola, in the Northern Flinders Ranges, South Australia (Figure 28).

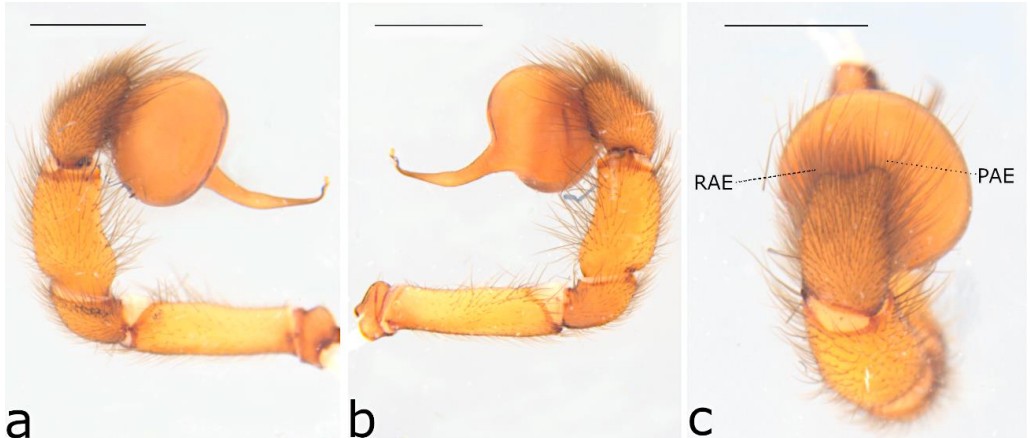

**Figure 32.** *Ariadna curvata* sp. nov. ♂holotype (SAMA NN30589) from Arkaroola (SA): (**a**) left pedipalp, prolateral view; (**b**) same, retrolateral view; (**c**) same, cymbium. Scale bar = 1 mm.

### *Ariadna deserta* **sp. nov.**

Figures 28, 33a–h and 34a–c

urn:lsid:zoobank.org:act:C960DE1E-FC99-4B13-845B-B72E89D460BA

**Type material.** *Holotype* ♂ AUSTRALIA: *South Australia*, 9.8 km SW Mount Goodair, Witjira National Park, 26.69694° S, 135.55278° E, 23 June 1995, pitfall trap, Stony Desert Biological Survey, ME06 (SAMA NN30493).

**Etymology.** The specific epithet is taken from the Latin adjective 'deserta', meaning solitary/lonely, in reference to the collection location of this species.

**Diagnosis.** *Ariadna deserta* sp. nov. can be distinguished from all other described species of *Ariadna* by the prolateral and retrolateral apophyses on metatarsus I, being present but low and not well-defined, so giving the appearance of a ventrally sinuous metatarsus I, whereas other species have either no apophyses, or at least one well-defined apophysis (Figure 33f). It can be distinguished from *A. woinarskii* sp. nov., which also has A sinuous metatarsus I, by the shorter embolus in relation to the bulb in *A. deserta* sp. nov., which contrasts with the strongly elongate tip of the embolus of *A. woinarskii* sp. nov. (Figure 33f vs. Figure 73f).

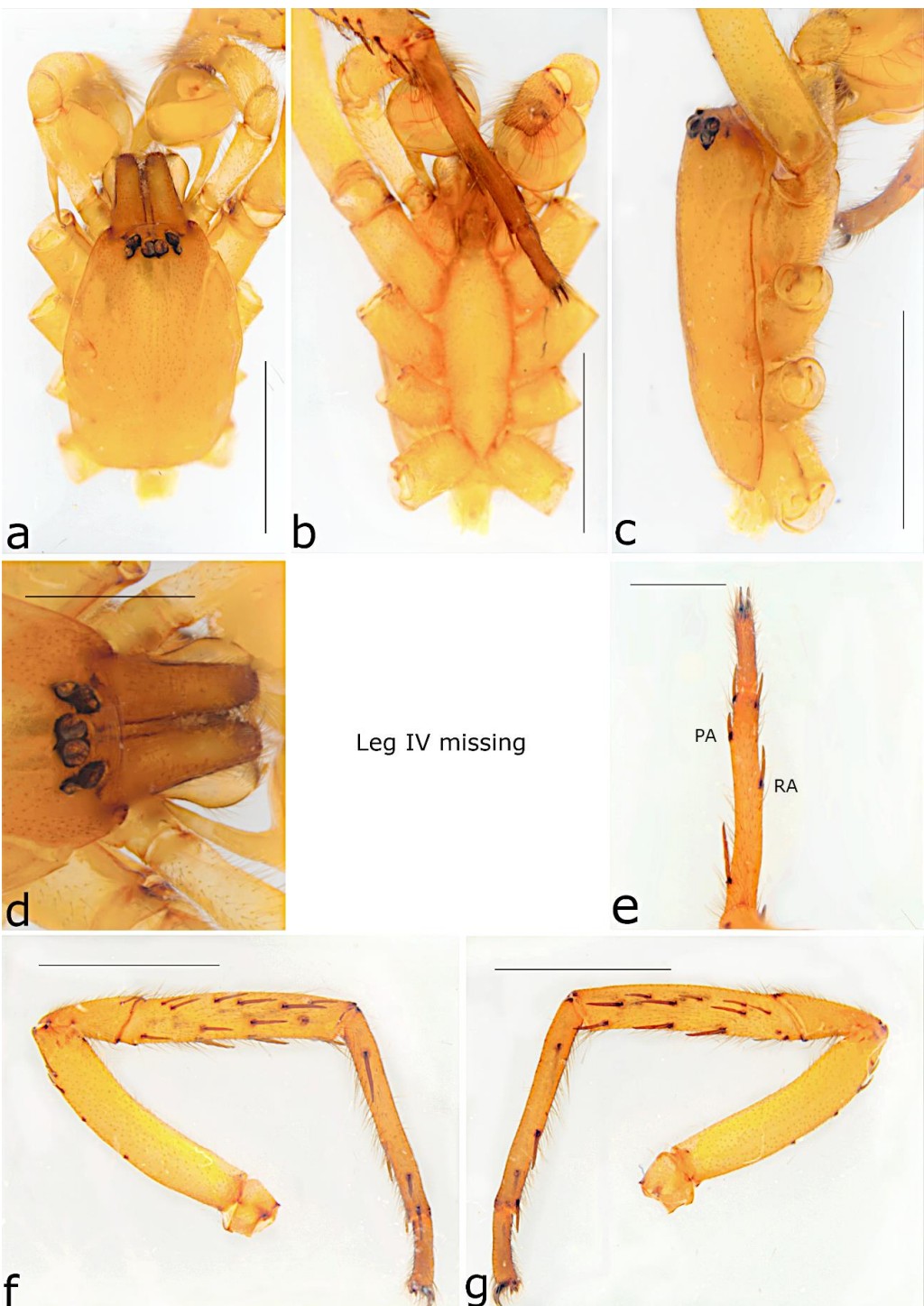

**Figure 33.** *Ariadna deserta* sp. nov. ♂ holotype (SAMA NN30493) from Witjira National Park (SA): (**a**) habitus, dorsal view; (**b**) same, ventral view; (**c**) same, lateral view; (**d**) eyes, dorsal view; (**e**) right leg I (mirrored), ventral view; (**f**) same, prolateral view; (**g**) same, retrolateral view. Scale bars (**a**–**e**) = 1 mm and (**f**,**g**) = 2 mm.

**Description.** ♂ (based on holotype; SAMA NN30493). Total length unknown (abdomen missing).

*Cephalothorax:* 2.8 long, 2.0 wide, 1.3 high. Carapace bright yellow-brown. Sternum yellow-brown. Maxillae, labium, and chelicerae darker orange-brown. Carapace rectangular, narrowing anteriorly; with scattered dark setae; fovea a shallow indented pit. Figure 33a. Labium about four-fifths the length of maxillae. Sternum elongate oval (Figure 33b). In

lateral view carapace flattened, chelicerae semi-porrect (Figure 33c). Eye group 0.7 wide, occupying 0.5 of the width of the carapace anteriorly; posterior eye row slightly recurved (Figure 33d).

Abdomen: Absent.

*Legs*: Leg II, III, IV absent. Leg I total 9.7: femur 2.8, patella 1.2, tibia 2.4, metatarsus 2.4, tarsus 0.9. Leg I yellow-golden-brown, metatarsus and tarsus darker orange-brown.

Tarsus I straight in lateral view and broader at apex than at base. Metatarsus I sinuous in ventral view, with a strongly flattened, indistinct prolateral apophysis, bearing a macroseta at 0.84 of the metatarsus length and a strongly flattened, indistinct retrolateral apophysis, situated at 0.65 of the metatarsus length (Figure 33f–h). Macrosetae: Leg I Femur d-1-1-1-1; patella p1; tibia dp1-1-1-1-1-1, vp1-1-1-1, vr1-1-1-1, r1-2-1-1-1-1; metatarsus p1 (long), apophysis on low, ill-defined mound, 1ap, r1-1. STC with around 11 teeth, ITC with a shortened stout tooth.

Pedipalp: Tibia 0.7 long, 0.6 wide, 1.2; cymbium 0.8 long, 0.5 wide. Tibia short, stout; cymbium rectangular, with two rounded triangular apical projections; prolateral projection larger. Bulb globular, embolus originating retrolaterally, elongate, mid portion broad until around three quarters of the length of the embolus, from where it tapers apically to tip; tip sharply sinuous (Figure 34a–c).

**Remarks.** The holotype is missing the abdomen and legs II–IV; however, the combination of the morphology of the pedipalp and leg I means that the diagnosis is not compromised.

**Distribution.** Only known from Witjira National Park, in the arid north of South Australia (Figure 28).

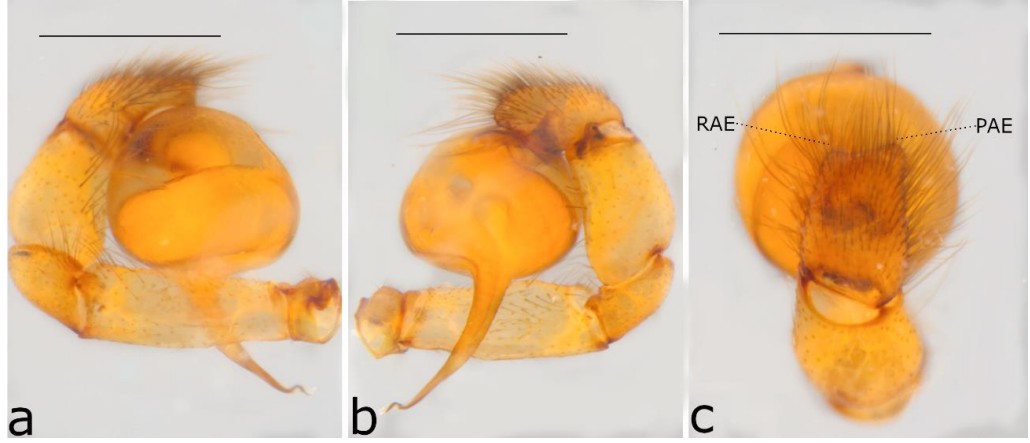

**Figure 34.** *Ariadna deserta* sp. nov. ♂holotype (SAMA NN30493) from Witjira National Park (SA): (**a**) left pedipalp, prolateral view; (**b**) same, retrolateral view; (**c**) same, cymbium. Scale bar = 1 mm.

### *Ariadna diucrura* sp. nov.

Figures 28, 35a–h and 36a–c

http://zoobank.org/xxxxxxxxxx

urn:lsid:zoobank.org:act:F106FE46-581A-41CD-8229-AB925B93F648

**Type material.** *Holotype* ♂ AUSTRALIA: *South Australia*, Olympic Dam site, Roxby Downs, 30.56° S, 136.89° E; March–June 1987, coll. Adam Smith (SAMA NN31206).

**Other material examined.** 9♂same data as holotype (SAMA NN30480).

**Etymology.** The specific epithet is a Latin adverb '*diu*', meaning long and the Latin noun crus (plural *crura*) for leg, referring to the long legs of this species.

**Diagnosis.** Of the species with two well-pronounced apophyses situated in the medial section of metatarsus I, *A. diucrura* sp. nov. can be differentiated by the retrolateral apophysis of metatarsus I being distinctly raised but smaller than the prolateral apophysis; contrasting to *A. flavescens* sp. nov., *A. arenacea* sp. nov., and *A. inflata* sp. nov. in which the retrolateral apophysis is indistinct and only slightly raised from the metatarsus (Figure 35f,

vs. Figure 37f, Figure 26f and Figure 40f), and from *A. pollex* sp. nov. and *A. ungua* sp. nov., in which both apophyses are subequal in size and strongly raised (Figure 35f vs. Figure 44f and Figure 65f). *Ariadna diucrura* sp. nov. can be differentiated from *A. valida* sp. nov. by the shape of the cymbium, which has a distinct and angular PAE and RAE, separated by a distinct 'v' shaped notch in *A. valida* sp. nov., but has an apically rounded PAE and indistinct RAE in *A. diucrura* sp. nov. (Figure 36c vs. Figure 70c).

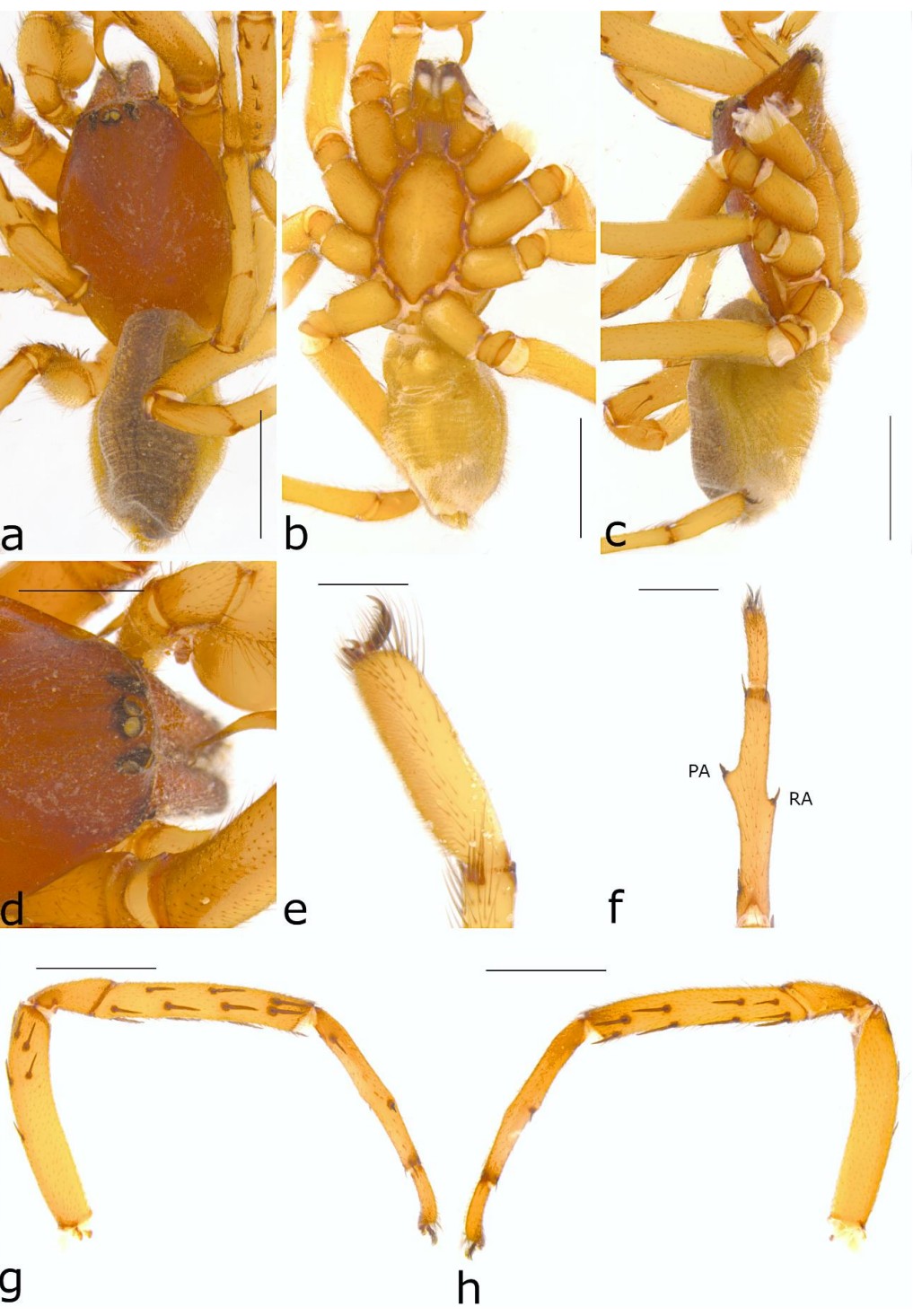

**Figure 35.** *Ariadna diucrura* sp. nov. ♂holotype (SAMA NN31206) from Roxby Downs (SA): (**a**) habitus, dorsal view; (**b**) same, ventral view; (**c**) same, lateral view; (**d**) eyes, dorsal view; (**e**) left metatarsus IV, preening comb, retrolateral view; (**f**) left leg I, ventral view; (**g**) same, prolateral view; (**h**) same, retrolateral view. Scale bars (**a–c**) = 2 mm, (**d**) = 1 mm, (**e**) = 0.5 mm, (**f**) = 1 mm and (**g,h**) = 2 mm.

**Description.** ♂ (based on holotype; SAMA NN31206). Total length 7.6.

*Cephalothorax:* 3.8 long, 2.7 wide, 1.5 high. Carapace golden red-brown. Sternum golden-orange, paler centrally, with darker orange at lateral edges and in pre-coxal triangles and intercoxal extensions. Maxillae orange, white apically; labium pale orange, chelicerae dark red-brown. Figure 35a–c. Carapace oval, fovea indented pit (Figure 35a). Labium about four-fifths length of maxillae. Sternum oval with distinct pre-coxal triangles and intercoxal extensions; with sparse, long, dark setae (Figure 35b). Flattened in lateral view, highest posterior to eye group (Figure 35c). Eye group 0.8 wide, occupying 0.5 of the width of the carapace anteriorly; posterior eye row slightly recurved (Figure 35d).

*Abdomen:* 3.8 long. Dorsally dark slate-grey; ventrally cream; with a covering of golden-brown setae (Figure 35a–c).

*Legs:* Leg length ratio I > II > IV > III. Leg I 12.7: femur 3.7, patella 1.5, tibia 3.3, metatarsus 3.1, tarsus 1.1. Leg II 12.5: 3.8, 1.3, 3.2, 3.1, 1.1. Leg III 9.9: 3.0, 1.1, 2.4, 2.3, 1.1. Leg IV 11.2: 3.8, 1.2, 2.8, 2.3, 1.1. Legs yellow-brown, tarsi and metatarsi I darker orange-brown. Femur I bowed in dorsal view. Metatarsus I straight, with a prolateral apophysis bearing a short, robust macroseta situated at 0.64 of the metatarsus length and a retrolateral apophysis, bearing a longer, elongate macroseta and situated at 0.53 of the metatarsus length (measured proximally to distally) (Figure 35f–h). Tarsus I bent prolaterally and moderately bowed ventrally in lateral view, broader at apex than at base; tarsi II–IV ventrally incrassate and with scopulate setae; tarsi I with no scopulate setae. Macrosetae: Leg I Femur d1-1-1ap, dp2ap, dr1ap; tibia dp1-1-1, vp1-1-1-1ap, vr2-1-1-1ap, r1-1-1-1; metatarsus p1 (short), p-apophysis-1 (robust, short), vp1ap, vr1ap, r-apophysis-1 (moderately elongated). Leg II femur d1-1-1-1-1ap, dp1-1ap, dr1ap; tibia p1-1-2-2, vr1-1-1-1ap, r1-1-1-1; metatarsus vp1-1-1ap, vr1-1-1ap, r1-1-1-1. Leg IV femur d1-1-1-1-1-1ap, dp1ap; metatarsus vp2ap. Retrolateral distal preening comb with four long macrosetae, equal length s(Figure 35e). STCs with around 10 teeth, ITC short, robust, with a short tooth.

*Pedipalp:* Tibia 1.0 long, 0.7 wide, 1.4; cymbium 0.9 long, 0.6 wide. Pedipalp: tibia bulbous anteriorly, curved ventrally; cymbium squared. Bulb, globular, embolus originating retrolaterally, gently curved, mid portion broad until three quarters of the length of the embolus, from where it tapers to tip; tip corkscrewed (Figure 36a–c).

**Variation.** Carapace lengths of males examined (*n* = 10) range from 3.6 to 4.3 (mean 3.9); colouration is consistent among specimens examined. For variation in leg macrosetae and position of apophyses on metatarsus I see Table S3.

**Distribution.** This species is only known from Roxby Downs, in the far north region of South Australia (Figure 28).

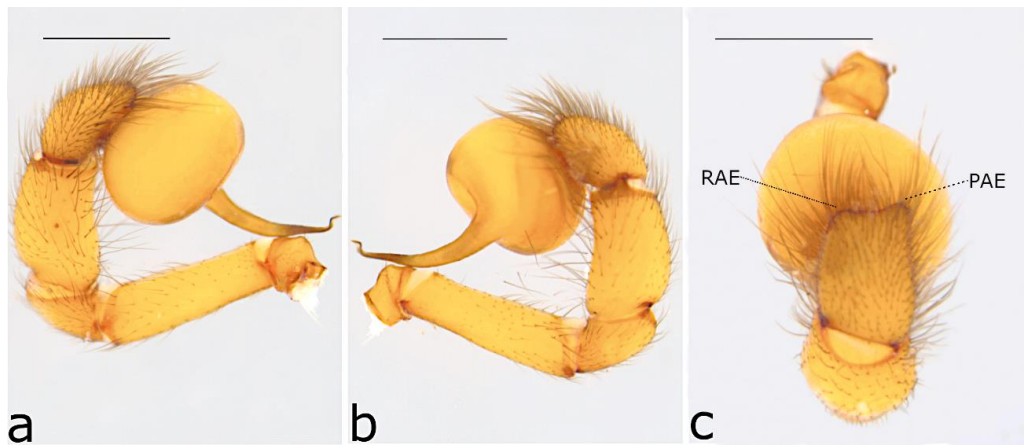

**Figure 36.** *Ariadna diucrura* sp. nov. ♂ holotype (SAMA NN31206) from Roxby Downs (SA): (**a**) left pedipalp, prolateral view; (**b**) same, retrolateral view; (**c**) same, cymbium. Scale bar = 1 mm.

*Ariadna flavescens* **sp. nov.**

Figure 37a–h, Figures 38a–c and 39

urn:lsid:zoobank.org:act:99D60FC3-9366-4257-831D-D3E1A2A83A2F

**Type material.** *Holotype* ♂ AUSTRALIA: *South Australia*, North Arckaringa Creek, 26.68778° S, 134.83972° E, pitfall trap, 26–30 Sept 1995 (SAMA NN30483)

**Etymology.** The specific epithet flavescens is a Latin singular present active participle in the nominative case that means 'yellowish', referring to the yellow colour of this species.

**Diagnosis.** *Ariadna flavescens* sp. nov. can be differentiated from other similar species with one well-defined apophysis on metatarsus I by the shape of the embolus, the mid portion of which is thickened until around half the length of the embolus, from where it gradually narrows to a curved tip in *A. flavescens* sp. nov., whereas, in *A. arenacea* sp. nov. and *A. inflata* sp. nov., the mid portion of the embolus is thickened to three-quarters of the embolus length, from where it narrows abruptly to a sinuous tip (Figure 38a,b vs. Figure 27a,b and Figure 41a,b). It can be further separated by the pedipalp cymbium, in which the PAE and RAE are well-defined and rounded distally in *A. flavescens* sp. nov. (Figure 38c), contrasting with that of *A. inflata* sp. nov. in which the apical extensions are both well-defined, pointed and angular distally, and *A. arenacea* sp. nov. in which the RAE is small and not well-defined (Figure 27c and Figure 41c).

**Description.** ♂ (based on holotype; SAMA NN30483). Total length 6.7.

*Cephalothorax:* 3.1 long, 2.1 wide, 1.8 high. Carapace golden yellow-brown. Scattered dark setae. Sternum golden yellow-brown, darker yellow in pre-coxal triangles. Maxillae yellow-orange-brown, paler apically, labium orange-brown, chelicerae red-brown. Carapace elongate oval, narrowing anteriorly; fovea an indistinct, shallow pit (Figure 37a). Labium about four fifths length of maxillae. Sternum elongate oval, with scattered setae and with pre-coxal triangles (Figure 37b). Carapace flattened in lateral view, highest midway between fovea and eye group; chelicerae semi-porrect (Figure 37c). Eye group 0.7 wide, occupying 0.6 of the width of the carapace anteriorly; posterior eye row straight (Figure 37d).

*Abdomen:* 3.6 long. Dorsally uniform pale grey, ventrally yellow-grey, with scattered red setae (Figure 37a–c).

*Legs:* Leg length ratio II > I > IV > III. Leg I total 9.6: femur 2.9, patella 1.2, tibia 2.3, metatarsus 2.1, tarsus 1.1. Leg II 10.8: 2.9, 1.2, 2.5, 2.2, 2.0. Leg III 7.6: 2.6, 0.9, 1.5, 1.5, 1.1. Leg IV 9.1: 3.0, 0.9, 2.4, 1.8, 1.1. Legs bright yellow-brown. Leg I metatarsus and tarsus darker orange-brown. Metatarsus I straight in ventral view, with an elevated and well-pronounced prolateral apophysis, bearing a short, stout blunt macroseta and an indistinct, low retrolateral apophysis, bearing an elongate macroseta, directed parallel to the plane of the metatarsus. Measured proximally to distally, the prolateral apophysis is at 0.76 the length of the metatarsus, and the retrolateral apophysis is at 0.47 of the length of the metatarsus (Figure 37f–h.). Tarsus I slightly bowed ventrally in lateral view, and broader at apex than at base; tarsi II–IV swollen ventrally and with scopulate setae. Tarsi IV elongate, only slightly shorter than the metatarsus. Macrosetae: Leg I Femur dp1ap, d1-1-1-1ap; tibia dp1-1-1-1, vp2 (1 small)-1-1-1, dr1-1-1-1, vr2-2-1-1; metatarsus p1(long), apophysis, p1ap, r apophysis, vr1ap. Leg II femur dp2ap, d1-1-1ap; tibia p1-1-1, vp1-1-1, vr1-1-1-1-1-1, r1-1-1-1-1; metatarsus vp1-1-1ap, vr1-1-1ap, r1-1-1. Leg IV femur d1, dp1ap; tibia v1-1; metatarsus v2ap. Retrolateral distal preening comb with 4 long macrosetae, 3 shorter, 1 longer (Figure 37e). STCs with around 11 teeth, ITC with shortened stout tooth.

*Pedipalp:* Tibia 0.7 long, 0.5 wide, 1.4; cymbium 0.7 long, 0.5 wide. Tibia expanded proximally, narrowing apically and curved; cymbium with small rounded triangular PAE. Bulb large oval-globular, embolus originating retrolaterally, elongate, mid portion broad until around half the length of the embolus, from where it tapers apically to tip; tip long, sinuous and corkscrewed at apex (Figure 38a–c).

**Distribution:** This species is only known from the type locality, near North Arckaringa Creek, in northern South Australia (Figure 39).

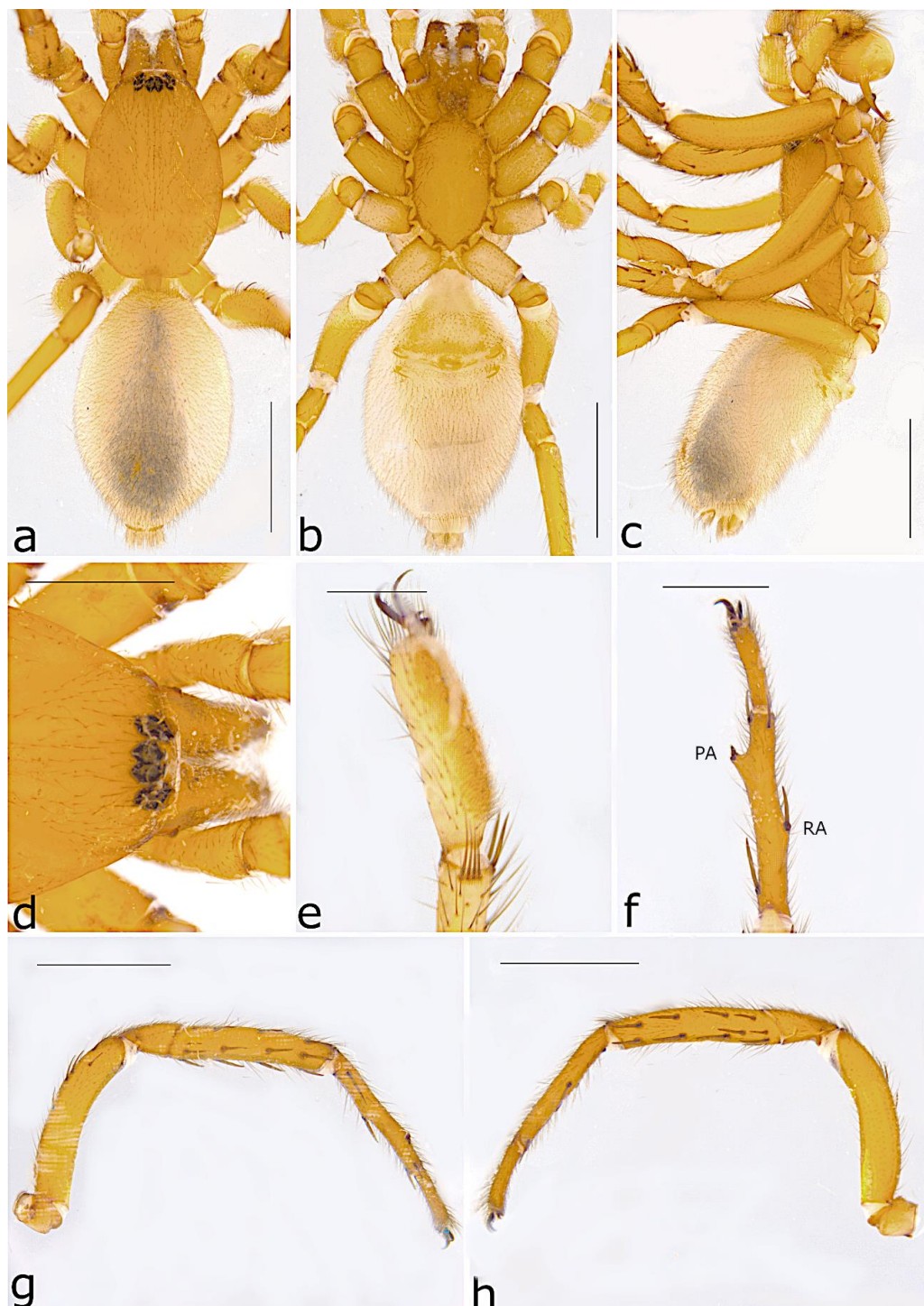

**Figure 37.** *Ariadna flavescens* sp. nov. ♂ holotype (SAMA NN30483) from north Arckaringa Creek (SA): (**a**) habitus dorsal view; (**b**) same, ventral view; (**c**) same, lateral view; (**d**) eyes, dorsal view; (**e**) left metatarsus IV, preening comb, retrolateral view; (**f**) left leg I, ventral view; (**g**) same prolateral view; (**h**) same, retrolateral view. Scale bars (**a–c**,**g**,**h**) = 2 mm, (**d**) = 1 mm, (**e**) = 0.5 mm, (**f**) = 1 mm and (**g**,**h**) = 2 mm.

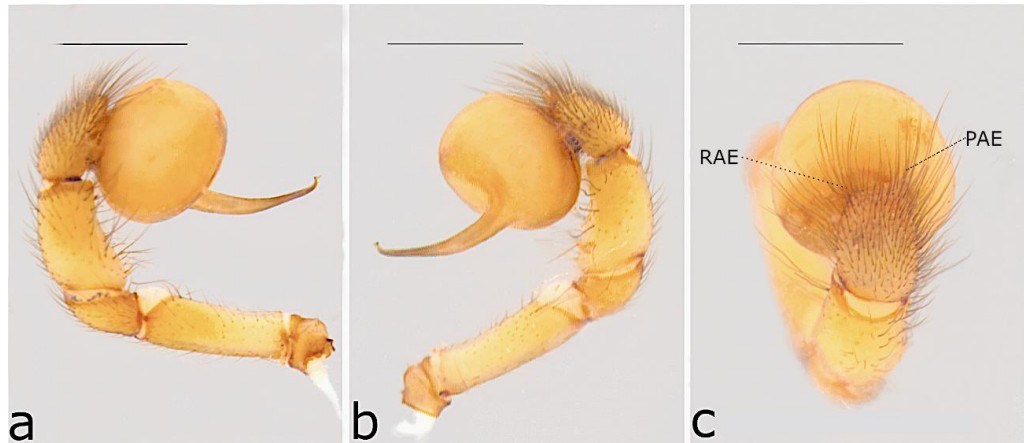

**Figure 38.** *Ariadna flavescens* **sp. nov.** ♂ holotype (SAMA NN30483) from north Arckaringa Creek (SA); (**a**) left pedipalp, prolateral view; (**b**) same, retrolateral view; (**c**) same, cymbium. Scale bar = 1 mm.

**Figure 39.** Map showing the currently known distribution of *A. flavescens* sp. nov.: *A.* inflata **sp. nov.,** A. insula **sp. nov.** A. pollex **sp. nov.** and A. propria **sp. nov**.

### *Ariadna inflata* **sp. nov.**

Figures 39, 40a–h and 41a–c

urn:lsid:zoobank.org:act:303B8E74-5ADA-40D1-AFE7-060A0C9375DA

**Type material.** *Holotype* ♂ AUSTRALIA: *South Australia*, Yorke Peninsula, 12.7 km ENE Marion Bay, 35.15944° S, 137.08777° E, open mallee *Eucalyptus diversifolia*, *Eucalyptus*

*oleosa*, 16–21 October 2004, pitfall trap, Mid North Yorke Peninsula Survey HIL 02201, Department of Environment and Heritage (SAMA NN22460).

**Etymology.** The specific epithet is a Latin adjective meaning 'inflated' and refers to the inflated fourth tarsi of this species.

**Diagnosis.** Of the species with only one well-defined apophysis on metatarsus I, *A. inflata* sp. nov. can be distinguished from *A. flavescens* sp. nov. by the shape of the embolus, the mid portion of which is thickened until around half the length of the embolus from where it gradually narrows to a curved tip in *A. flavescens* sp. nov., whereas, in *A. inflata* sp. nov., it is thickened to three-quarters of the embolus length, from where it narrows abruptly to a sinuous tip (Figure 41a,b vs. Figure 38a,b). It can be differentiated from *A. arenacea* sp. nov., by the relative length of the embolus to the pedipalp bulb, where in *A. arenacea* sp. nov. the embolus is relatively short, being around the same length as the width of the bulb; whereas in *A. inflata* sp. nov. the embolus is elongate, clearly longer than the width of the bulb (Figure 41a,b vs. Figure 27a,b). It can be further separated by the pedipalp cymbium, which has well-defined, angular, triangular extensions in *A. inflata* sp. nov., with both the PAE and RAE well-pronounced, which contrasts with the rounded PAE and RAE in *A. flavescens* sp. nov. (Figure 41c vs. Figure 39c) and an angular and well-defined PAE in *A. arenacea* sp. nov. but with a low and not well-defined RAE (Figure 41c vs. Figure 28c).

**Description.** ♂ (based on holotype; SAMA NN22460). Total length 5.9.

*Cephalothorax:* 3.3 long, 2.3 wide, 1.6 high. Carapace golden red-brown, darker brown anteriorly, golden yellow-brown posteriorly. Scattered dark setae. Sternum golden-brown, darker mottled markings anteriorly, darker orange at lateral edges and faintly in pre-coxal triangles. Maxillae orange-brown, white apically, labium orange-brown, chelicerae red-brown(Figure 40a–c). Carapace oval, narrowing anteriorly, posteriorly strongly indented at midsection; fovea an elongate, shallow indented pit (Figure 40a). Labium about four fifths length of maxillae. Sternum rounded oval with pre-coxal triangles, with scattered, long, dark setae (Figure 40b). Flattened in lateral view, the highest midway between the fovea and eye group (Figure 40c). Eye group prominent, projecting over the chelicerae; 0.7 wide, occupying 0.6 of the width of the carapace anteriorly; posterior eye row slightly recurved (Figure 40d).

*Abdomen:* 2.6 long. Dorsally dark violet-grey, uniform in colour; ventrally grey medially, light cream-brown anteriorly; with scattered red-brown setae (Figure 40a–c).

*Legs:* Leg length ratio I > II > IV > III. Leg I total 9.9: femur 2.9, patella 1.1, tibia 2.6, metatarsus 2.5, tarsus 0.8. Leg II 8.6: 2.7, 1.1, 2.2, 1.8, 0.8. Leg III 6.2: 1.8, p1.0, 1.3, 1.2, 0.9. Leg IV 7.5: 2.4, 1.2, 2.1, 1.1, 0.7. Legs yellow-brown. Leg I metatarsus and tarsus darker orange-brown. Femur I bowed in dorsal view. Metatarsus I sinuous in ventral view, with a distinct, raised prolateral apophysis arising at 0.75 of the length of metatarsus, and bearing a stout, blunt macroseta; the retrolateral apophysis is flattened and indistinct and is situated at 0.57 of the metatarsus length (Figure 40f–h). Tarsus I short, curved ventrally in lateral view, slightly broader at apex than at base. Tarsi II, III straight in lateral view, tarsi IV inflated, cigar shaped, widest basally and narrowing anteriorly. Tarsi II with sparse scopulate setae anteriorly, tarsi III, IV along entire length ventrally. Coxae elongate. Macrosetae: Leg I Femur dp2ap, d1; tibia dp1-1-1-1, vp1-1-1, vr1-1-1-1ap, r1-1-1-1-1; metatarsus p1-apophysis-1ap, r-apophysis-1ap. Leg II femur dp2ap, dr1ap; tibia dp1-1-1, vp1-1, vr1-1-1-1ap, r1-1-1; metatarsus p1-1ap, vr1-1ap, r1-1. Leg IV femur d1-1-1; retrolateral distal preening comb with four long macrosetae, three longer, one shorter (Figure 40e).

*Pedipalp:* Tibia 0.7 long, 0.5 wide, 1.4; cymbium 0.6 long, 0.5 wide. Pedipalp tibia short and curved. Cymbium rectangular, with triangular prolateral and retrolateral apical expansions. The bulb is globular. The embolus originates retrolaterally from the bulb; the mid portion is broad until around three-quarters of the length of the embolus, from which it tapers apically to tip; tip hooked (Figure 41a–c).

**Distribution.** Only known from the southern Yorke Peninsula, South Australia (Figure 39).

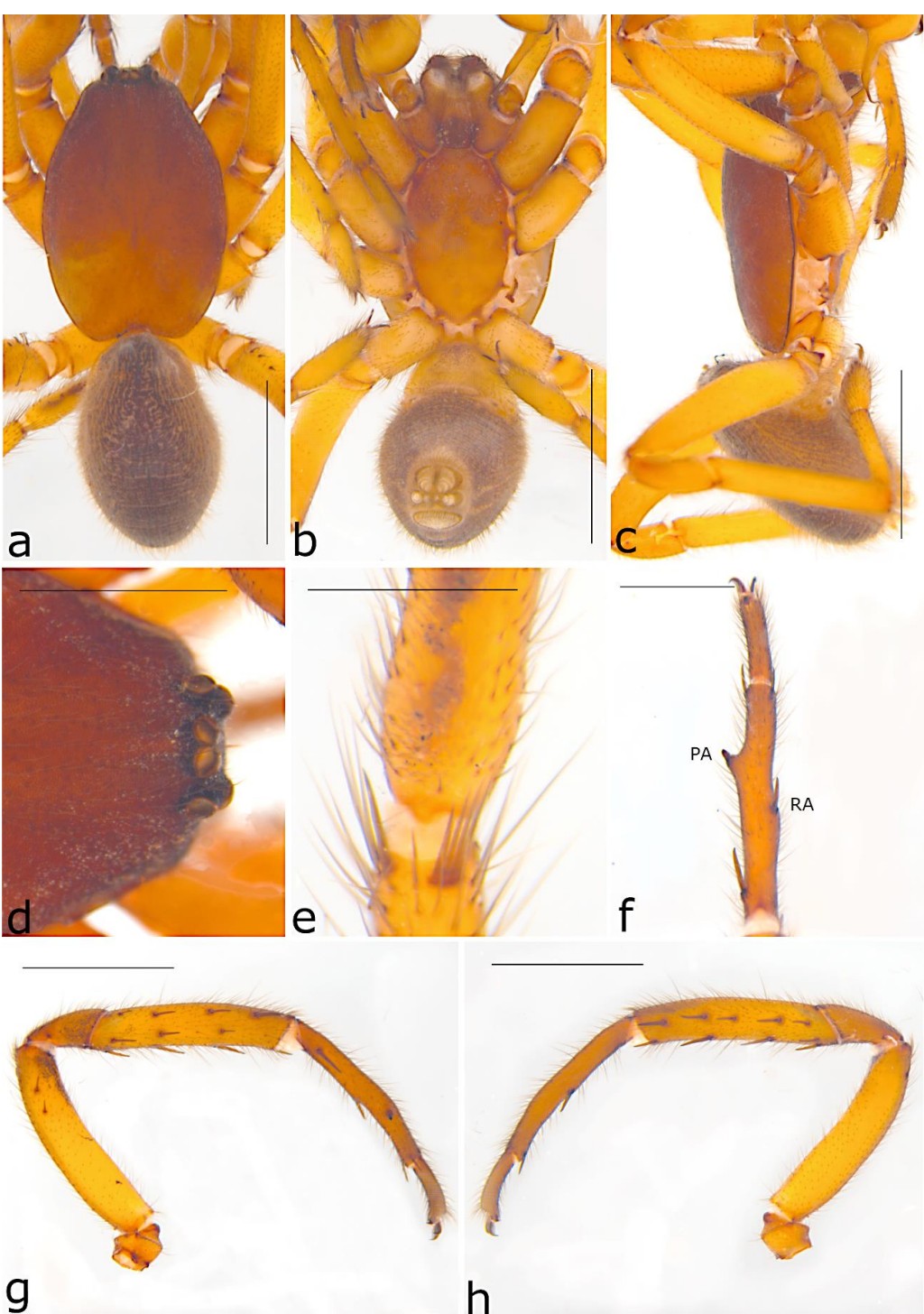

**Figure 40.** *Ariadna inflata* sp. nov. ♂ holotype (SAMA NN22460) from Yorke Peninsula (SA): (**a**) habitus, dorsal view; (**b**) same, ventral view; (**c**) same, lateral view; (**d**) eyes, dorsal view; (**e**) left metatarsus IV, preening comb, retrolateral view; (**f**) left leg I, ventral view; (**g**) same, prolateral view; (**h**) same, retrolateral view. Scale bars (**a**–**c**) = 2 mm, (**d**) = 1 mm, (**e**) = 0.5 mm, (**f**) = 1 mm and (**g**,**h**) = 2 mm.

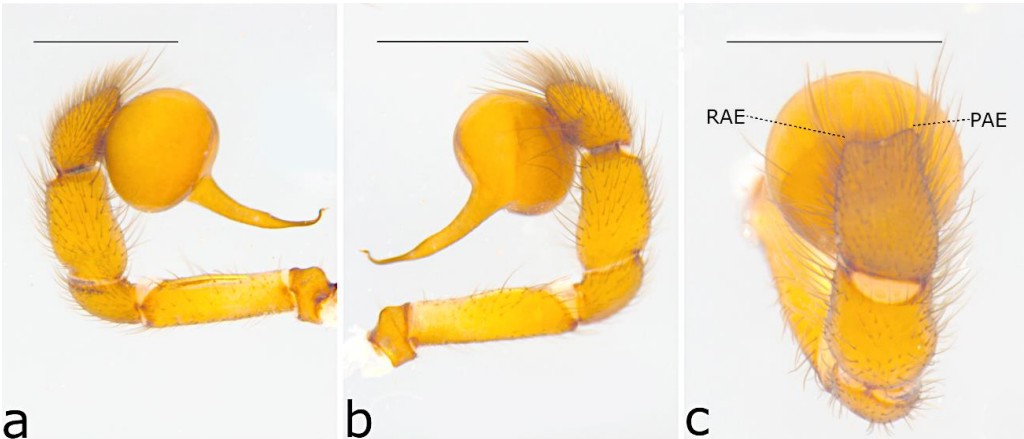

**Figure 41.** *Ariadna inflata* sp. nov. ♂holotype (SAMA NN22460) from Yorke Peninsula (SA): (**a**) left pedipalp, prolateral view; (**b**) same, retrolateral view; (**c**) same, cymbium. Scale bar = 1 mm.

### *Ariadna insula* **sp. nov.**

Figures 39, 42a–h and 43a–c

urn:lsid:zoobank.org:act:A114B860-80C4-435D-AD00-A054C9256D1B

**Type material.** *Holotype* ♂ AUSTRALIA: *South Australia*, Franklin Island, 32.28° S; 133.57° E; 27 November 1984, coll. B. Guerin (SAMA NN30573).

**Other material examined.** 1♂ Franklin Island, 32.27° S; 133.58° E; spotlighting, November 1985, coll. B. Guerin (SAMA NN30568).

**Etymology.** The specific epithet is from the Latin *insula* meaning 'island', referencing the type locality of this species.

**Diagnosis.** *Ariadna insula* sp. nov. can be differentiated from the morphologically closest species, *A. woinarskii* sp. nov. by the macrosetae of metatarsus I, where there is prolateral and retrolateral macrosetae in addition to ventral retrolateral and ventral prolateral *A. insula* sp. nov., whereas, in *A. woinarskii* sp. nov., there are only ventral retrolateral and ventral prolateral macrosetae (Figure 42f–h vs. Figure 73f–h).

**Description.** ♂ (based on holotype; SAMA NN30573). Total length 5.6.

*Cephalothorax:* 2.9 long, 2.0 wide, 1.6 high. Carapace golden-brown, darker on lateral edges, with faint small, scale-like markings. Scattered dark setae. Faint, mottled medial longitudinal darker striations, extending out from fovea and around eye mound. Sternum orange-cream, with faint darker broad medial line. Darker orange at lateral edges and faintly in pre-coxal triangles and intercoxal extensions. Maxillae orange-cream, white apically, labium orange-brown, chelicerae red-brown (Figure 42a–c). Carapace oval, narrowing anteriorly; fovea a shallow indented pit (Figure 42a). Labium about four fifths length of maxillae. Sternum rounded oval with pre-coxal triangles and with smaller, rounded intercoxal extensions, with scattered, long, dark setae (Figure 42b). Flattened in lateral view; chelicerae semi-porrect (Figure 42c). Eye group 0.6 wide, occupying 0.6 of the width of the carapace anteriorly; posterior eye row slightly recurved (Figure 42d).

*Abdomen*: 2.7 long. Dorsally uniformly dark violet-grey; ventrally grey medially, light cream at lateral edges; with scattered red-brown setae (Figure 42a–c).

*Legs:* Leg length ratio I > II > IV > III. Leg I total 9: femur 2.6, patella 1.1, tibia 2.4, metatarsus 2.3, tarsus 0.6. Leg II 8.2: 2.4, 0.9, 2.2, 2.0, 0.7. Leg III 6.1: 1.8, 0.8, 1.4, 1.3, 0.8. Leg IV 7: 2.2, 0.8, 1.9, 1.3, 0.8. Legs orange-brown, darker towards apex (Figure 42a–h). Femur I bowed in dorsal view. Metatarsus I without apophyses (Figure 42f–h). Tarsus I slightly bowed ventrally in lateral view and broader at the apex than at the base; tarsi II–IV straight. Tarsi I, II, III with sparse scopulate setae apically, tarsi IV with no scopulate macrosetae. Macrosetae: Leg I Femur dp3ap, dr1ap, d1-1-1-1ap; tibia dp1, p1-1-1-2ap, vp1, vr2-1-1-1ap, r1-1-1; metatarsus p1-1-1, vp1-1-1ap, vr1-1-1ap, r1-1. Leg II femur dp2ap, dr1ap, d1-1-1-1ap; tibia dp1-1-1-1, vp1-1ap, vr1-1-1-1-1ap, r1-1-1-1; metatarsus dp1-1, p1-1-1ap, vr1-1, r1-1-1ap. Leg IV femur d1-1-1-1-1, dp1ap; tibia v1-1; metatarsus v2ap. Retrolateral distal preening

comb with five macrosetae, three longer, two shorter (Figure 42e). STCs with around nine teeth, ITC with a shortened stout tooth.

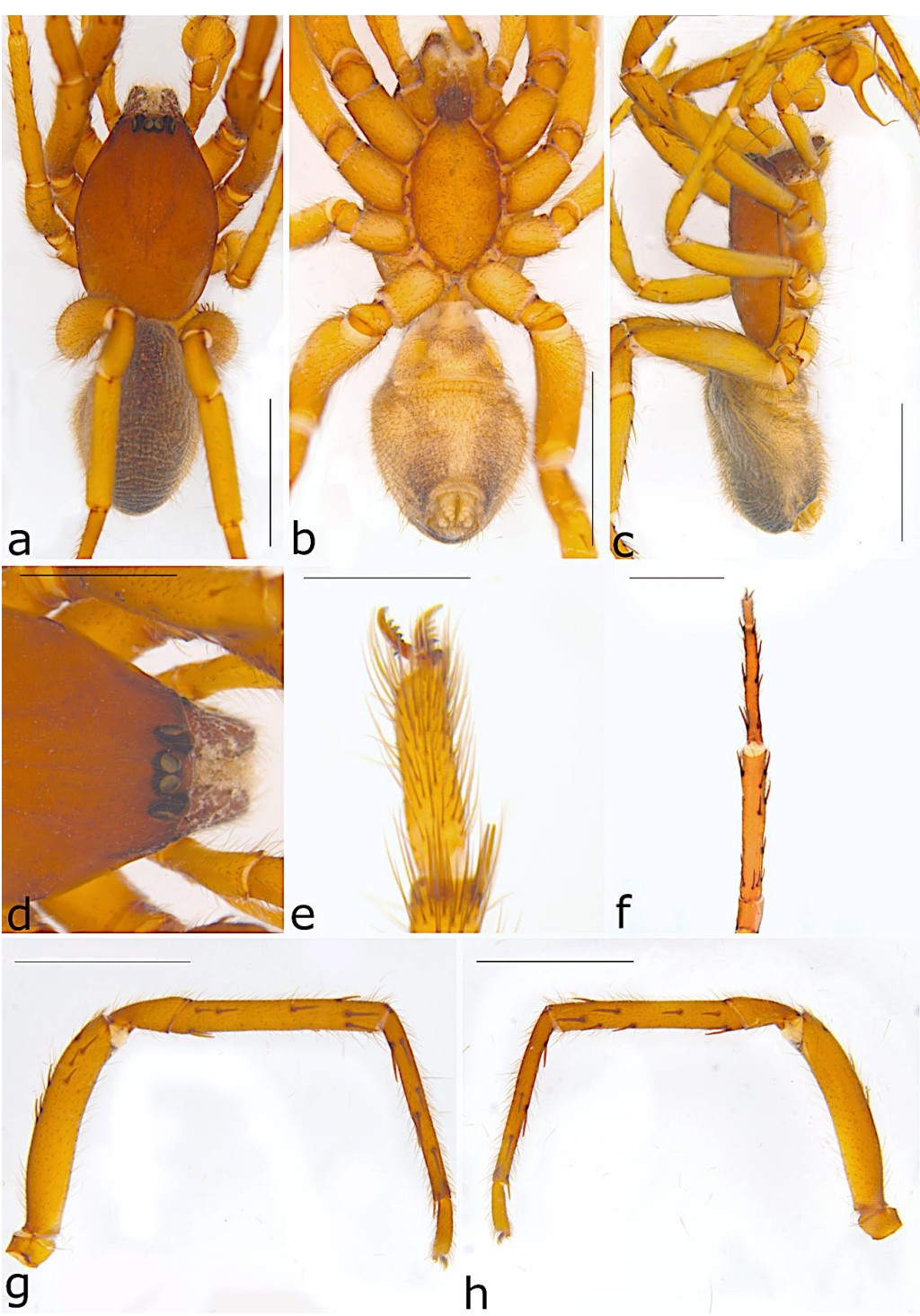

**Figure 42.** *Ariadna insula* sp. nov. ♂ holotype (SAMA NN30573) from Franklin Island (SA): (**a**) habitus, dorsal view; (**b**) same, ventral view; (**c**) same, lateral view; (**d**) eyes, dorsal view; (**e**) left metatarsus IV, preening comb, retrolateral view; (**f**) left leg I, ventral view; (**g**) same, prolateral view; (**h**) same, retrolateral view. Scale bars (**a–c**) = 2 mm, (**d**) = 1 mm, (**e**) = 0.5 mm and (**f–h**) = 2 mm.

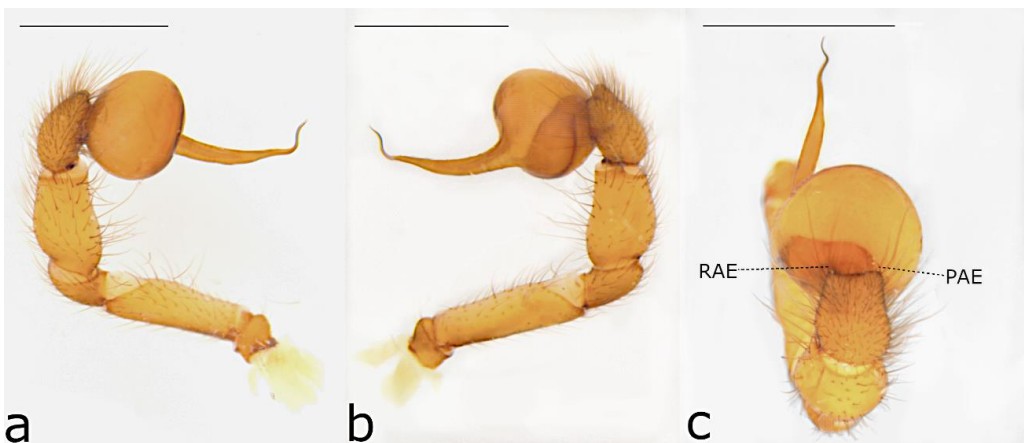

**Figure 43.** *Ariadna insula* sp. nov. ♂holotype (SAMA NN30573) from Franklin Island (SA); (**a**) left pedipalp, prolateral view; (**b**) same, retrolateral view; (**c**) same, cymbium. Scale bar = 1 mm.

*Pedipalp:* Tibia 0.6 long, 0.4 wide, 1.5; cymbium 0.5 long, 0.4 wide. Tibia expanded proximally, narrowing apically; cymbium squared, with small but distinct apical prolateral and retrolateral extensions. Bulb large globular, embolus originating retrolaterally, elongate, mid portion broad until around three-quarters of the length of the embolus, where there is a brief prolateral expansion, from which it tapers apically to the tip; the tip elongated, sinuous and strongly hooked at the apex (Figure 43a–c).

**Variation.** Carapace lengths of males examined (*n* = 2) was 2.6 and 2.9; colouration was consistent for both specimens. For variation in leg macrosetae see Table S3.

**Distribution.** Only known from Franklin Island, a small island off the south-west coast of mainland South Australia (Figure 39).

*Ariadna pollex* **sp. nov.**

Figures 39, 44a–h and 45a–c

urn:lsid:zoobank.org:act:460CD0BB-6B7E-4101-B363-F15473AC060A

**Type material.** *Holotype* ♂ AUSTRALIA: *South Australia:* Purni Bore, 26.2846° S, 136.0978° E, 24–29 August 1991, coll. H. Ehmann (SAMA NN30485).

**Etymology.** The specific epithet is a noun in apposition and is taken from the Latin word pollex meaning 'thumb' and named after the thumb, such as apophysis on leg I.

**Diagnosis.** Of the species with two well-defined apophyses situated in the apical half of metatarsus I, *A. pollex* sp. nov. can be differentiated from *A. ungua* sp. nov. by the shape of the pedipalp tibia, which is slender, being 1.5 the length times the width in *A. ungua* sp. nov.; but is bulbous and stout in *A. pollex* sp. nov., being 1.3 the length times the width (Figure 45a,b vs. Figure 66a,b).

**Description.** ♂(based on holotype; SAMA NN30485). Total length 7.2.

*Cephalothorax:* 3.4 long, 2.2 wide, 2.2 high. Carapace golden orange-brown, with faint small, scale-like markings. Carapace darker red-brown anteriorly of fovea, demarcating cephalic area; with scattered dark setae. Sternum pale yellow-cream, with darker cream at lateral edges and darker orange in pre-coxal triangles; with scattered fine dark setae. Maxillae yellow-cream, white apically, labium yellow-brown, chelicerae orange-brown. Figure 44a–c. Carapace oval and narrowing anteriorly; edges gently undulating; fovea a shallow indentation (Figure 44a). Labium about three-quarters length of maxillae. Sternum elongated oval with pre-coxal triangles and with small, rounded intercoxal extensions (Figure 44b). Flattened in lateral view, highest just posterior to eye row (Figure 44c). Eye group 0.7 wide, occupying 0.6 of the width of the carapace anteriorly; posterior eye row slightly recurved (Figure 44d).

*Abdomen*: 3.8 long. Dorsally with broad violet-grey longitudinal medial strip, pale cream laterally; ventrally uniform yellow-cream; with sparse golden-brown setae (Figure 44a–c).

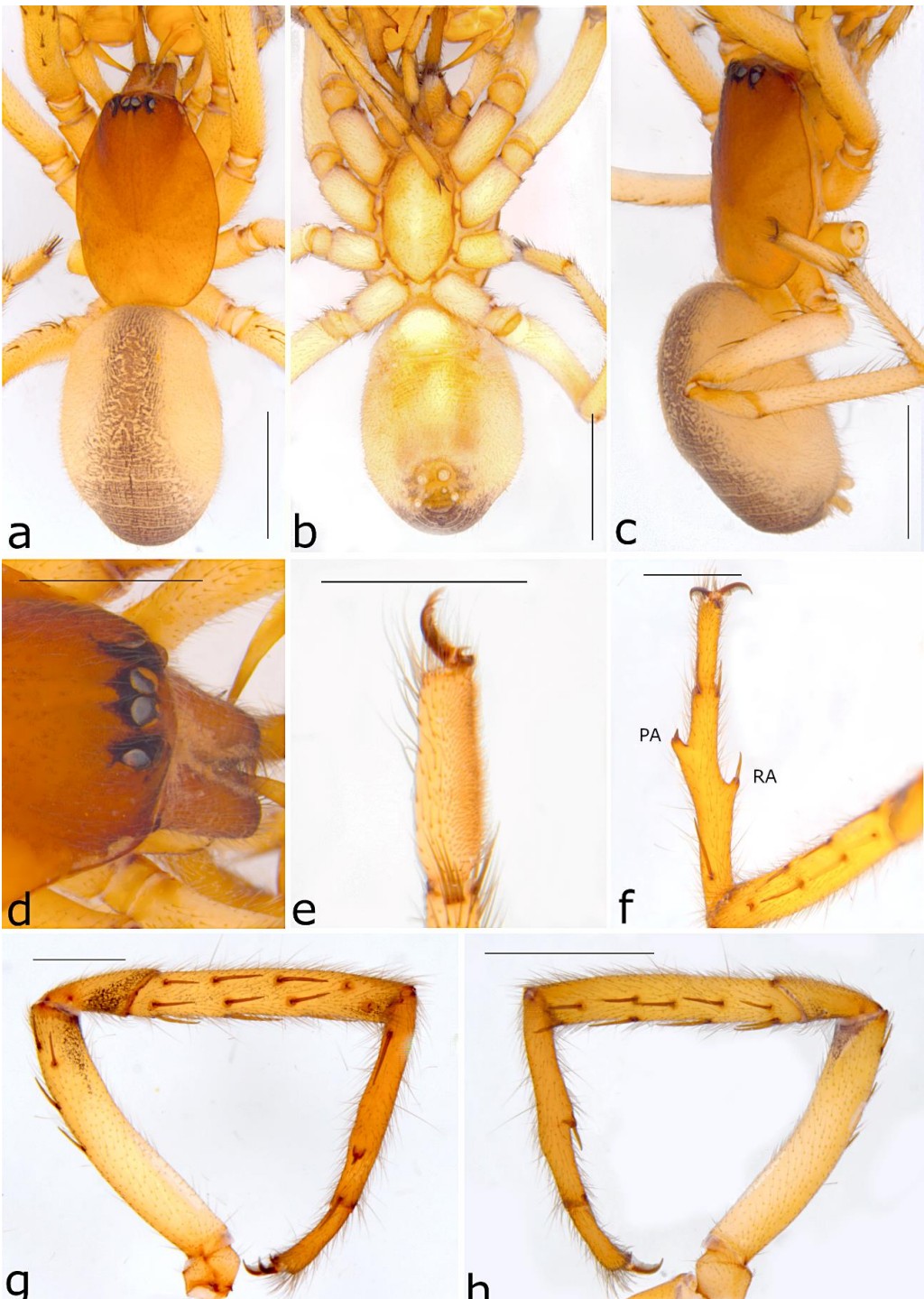

**Figure 44.** *Ariadna pollex* sp. nov. ♂ holotype (SAMA NN30485) from Purni Bore (SA): (**a**) habitus, dorsal view; (**b**) same, ventral view; (**c**) same, lateral view; (**d**) eyes, dorsal view; (**e**) left metatarsus IV, preening comb, retrolateral view; (**f**) left leg I, ventral view; (**g**) same, prolateral view; (**h**) same, retrolateral view. Scale bars (**a**–**c**) = 2 mm, (**d**,**e**) = 1 mm, (**f**,**g**) = 1 mm and (**h**) = 2 mm.

*Legs:* Leg length ratio I > IV > II > III. Leg I total 10.8: femur 3.3, patella 1.3, tibia 2.9, metatarsus 2.4, tarsus 0.9. Leg II 9.8: 3.0, 1.0, 2.6, 2.3, 0.9. Leg III 8.6: 2.8, 1.0, 2.0, 1.8, 1.0. Leg IV 10.4: 3.2, 1.2, 2.8, 2.1, 1.1. Legs yellow-cream, darker orange towards apex; with sparse dark fine setae. Femur I bowed in dorsal view. Metatarsus I sinuous, with two opposing apophyses; measuring proximally to distally, prolateral apophysis arising at 0.71 length of metatarsus and retrolateral apophysis arising at 0.56 of the length of metatarsus. The

prolateral apophysis bears a short, stout macroseta and the retrolateral apophysis, a long macroseta (Figure 44f–h). Tarsus I moderately bowed ventrally in lateral view, broader at the apex than at the base; tarsi II–IV cigar-shaped, the same width along their length. Tarsi II with sparse scopulate setae apically, tarsi III and IV with denser scopulate macrosetae along length; tarsi I with no scopulate setae. Macrosetae: Leg I Femur dp1ap, d1-1-1/0-2; tibia p1-1-1-1, vp1-1-1-1ap, vr1-1-1-1ap, r1-1-1-1-1; metatarsus p1 (elongate)-apophysis (blunt)- p1ap, r-apophysis, r1ap. Leg II femur dp1ap, d1-1-1-2; tibia p1-1-1, vp1-1ap, vr1-1-1-1-1-1ap, r1-1-1-1-1; metatarsus vp1-1-1ap, vr1-1-1-1ap, r1-1-1. Leg IV femur d1-1-1-1; metatarsus vr2ap. Retrolateral distal preening comb with four macrosetae: one short and three long (Figure 44e). STCs, with around 18 teeth, ITC with elongate, thin tooth; claw of legs I, II, IV elongate.

*Pedipalp:* Tibia 0.8 long, 0.6 wide; cymbium 0.8 long, 0.5 wide. Tibia curved ventrally, cymbium rectangular with a small PAE. Bulb large globular, embolus originating retrolaterally, mid portion broad up to half the length of the embolus, from where it tapers apically to tip; tip sinuous and hooked at apex (Figure 45a–c).

**Distribution:** This species is only known from Purni Bore in the arid far north of South Australia (Figure 39).

**Remarks.** Specimen SAMA NN30487 (4 km WSW Damperanie Yard, 26.92694° S, 139.40278° E, pitfall trap, 3–6 May 1995, Stony Desert Survey, coll. P. Copley), was missing pedipalps and legs II–IV. The morphology of leg I is very similar to that of *A. pollex* sp. nov., however, the two specimens cannot be definitively linked based on the data available.

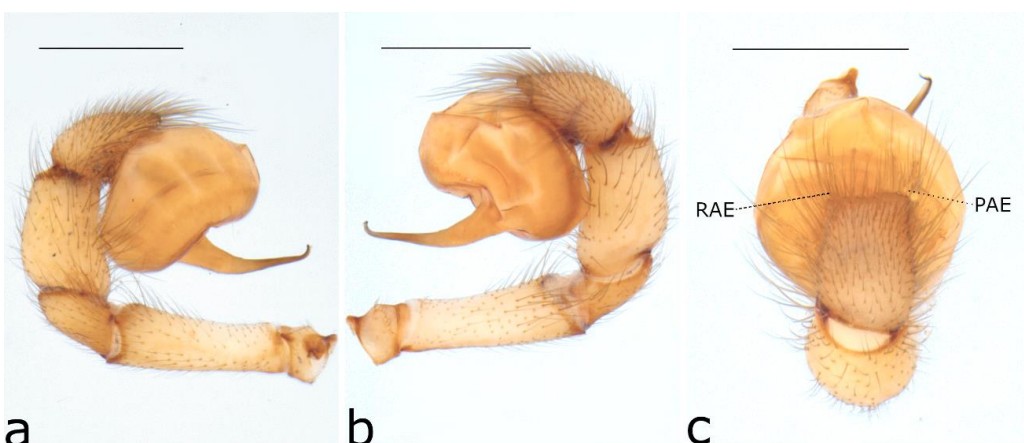

**Figure 45.** *Ariadna pollex* sp. nov. ♂ holotype (SAMA NN30485) from Purni Bore (SA): (**a**) left pedipalp, prolateral view; (**b**) same, retrolateral view; (**c**) same, cymbium. Scale bar = 1 mm.

### *Ariadna propria* **sp. nov.**

Figure 39, Figure 46a–h, Figure 47a–c, Figures 48a–h and 49a,b

urn:lsid:zoobank.org:act:36C54B70-024C-4C63-B4CC-FBC8F759C9DB

**Type material.** *Holotype* ♂ AUSTRALIA: *South Australia*, Melrose, 33.82° S, 138.18° E, alongside walking trail to obelisk at night, 2 September 2000, coll. D. Hirst (SAMA NN30560)

*Paratype* series 8♀ AUSTRALIA: *South Australia*, Melrose, camping area, 18 April 1987, coll. D. Hirst (SAMA NN31205; SAMA NN30536a; SAMA NN30536).

**Etymology.** The specific epithet is a Latin adjective meaning distinct or peculiar and referring to the distinct macrosetae of leg I of the females of this species.

**Diagnosis.** *Ariadna propria* sp. nov. can be differentiated *A. una* sp. nov., with which it shares the retrolateral apophysis being situated in the first quarter of the metatarsus by the shape of the cymbium apically, which is straight and squared in *Ariadna propria* sp. nov., but has a rounded PAE in *A. una* sp. nov. and by the elongate and sinuous embolus tip of *A. propria* sp. nov., compared to the shorter and less sinuous tip of *A. una* sp. nov. (Figure 47a–c vs. Figure 63a–c). Females of *A. propria* sp. nov. can be differentiated from

other similar species by the presence of a cluster of short ventroretrolateral macrosetae on metatarsus I (Figure 48h).

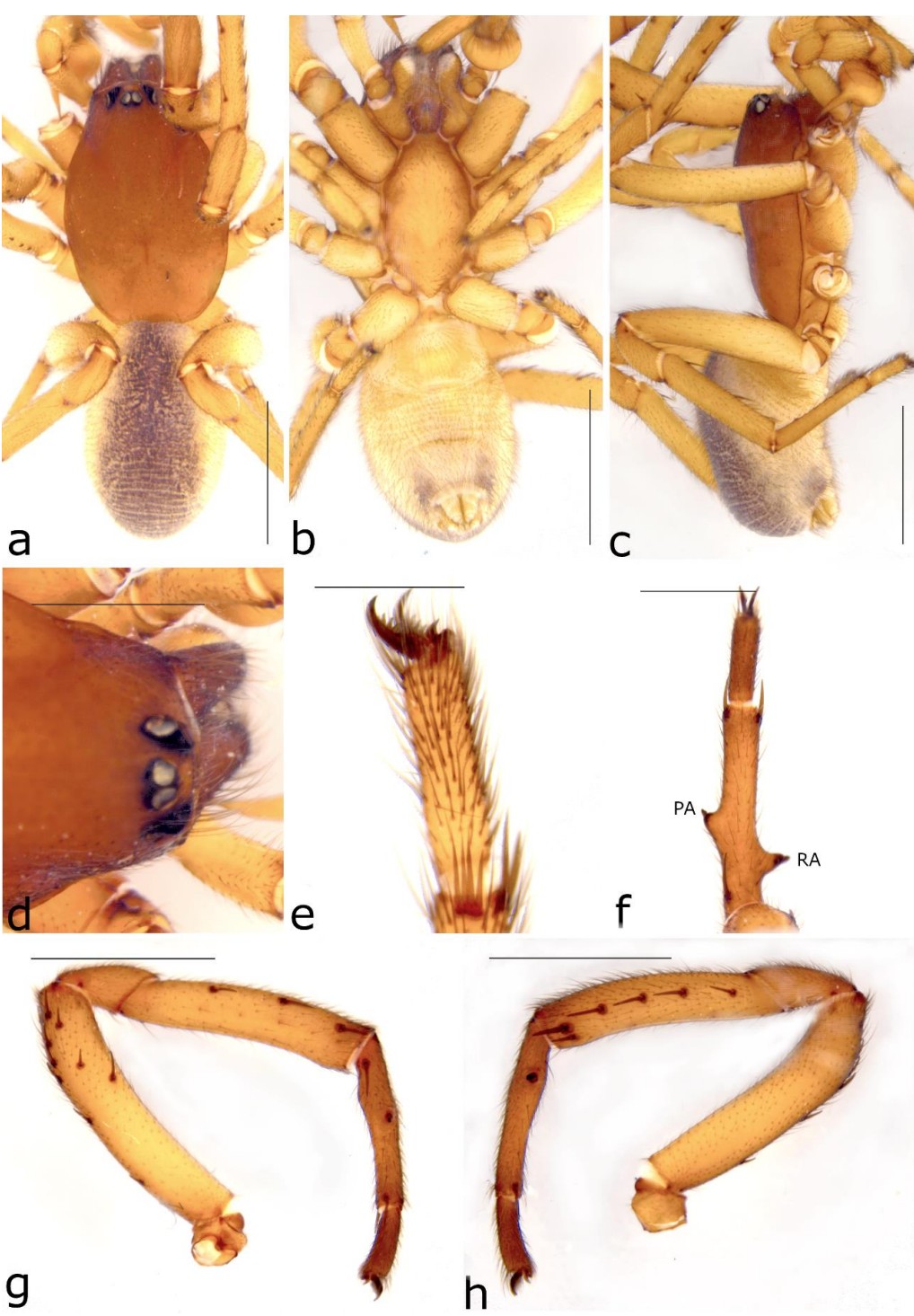

**Figure 46.** *Ariadna propria* sp. nov. ♂holotype (SAMA NN30560) from Melrose (SA): (**a**) habitus, dorsal view; (**b**) same, ventral view; (**c**) same, lateral view; (**d**) eyes, dorsal view; (**e**) left metatarsus IV, preening comb, retrolateral view; (**f**) left leg I, ventral view; (**g**) same, prolateral view; (**h**) same, retrolateral view. Scale bars (**a**–**d**) = 1 mm, (**e**) = 0.5 mm, (**f**) = 1 mm and (**g**,**h**) = 2 mm.

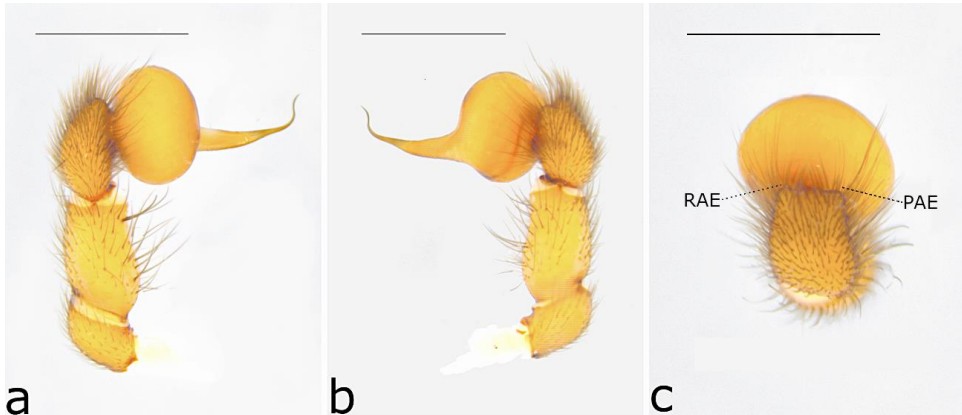

**Figure 47.** *Ariadna propria* sp. nov. ♂holotype (SAMA NN30560) from Melrose (SA): (**a**) left pedipalp, prolateral view; (**b**) same, retrolateral view; (**c**) same, cymbium. Scale bar = (**a**,**b**) 1 mm and (**c**) = 0.5 mm.

**Description.** ♂(based on holotype; SAMA NN30560). Total length 6.6.

*Cephalothorax:* 3.5 long, 2.3 wide, 1.5 high. Carapace golden red-brown. Sternum pale orange-brown, with darker orange at lateral edges and in pre-coxal triangles. Maxillae pale orange, white apically; labium pale orange, brown laterally. Chelicerae dark golden-red-brown (Figure 46a–c). Carapace rounded oval and narrowing anteriorly to form a distinct neck; edges broadly and slightly undulating; fovea an indented line (Figure 46a). Labium about four fifths length of maxillae. Sternum elongated oval with distinct pre-coxal triangles and with rounded, indistinct intercoxal extensions; with scattered, long, dark setae (Figure 46b). Carapace gently raised in lateral view, highest at eye group (Figure 46c). Eye group 0.7 wide, occupying 0.6 of the width of the carapace anteriorly; posterior eye row slightly recurved (Figure 46d).

*Abdomen:* 3.1 long. Dorsally slatey violet-grey, cream laterally; ventrally cream; with a covering of golden-brown setae (Figure 46a–c).

*Legs:* Leg length ratio II > I > IV > III. Leg I total 9.2: femur 3.1, patella 1.0, tibia 2.5, metatarsus 1.8, tarsus 0.8. Leg II 9.3: 3.0, 1.1, 2.5, 1.9, 0.8. Leg III 7.5: 2.4, 0.9, 1.7, 1.6, 0.9. Leg IV 7.7: 2.6, 0.9, 1.8, 1.7, 0.7. Legs uniform orange-brown, tibia I, metatarsus I and tarsus I darker orange. Femur I strongly bowed in dorsal view. Metatarsus I straight in ventral view, with a prominent prolateral apophysis bearing a short, robust macroseta and a prominent retrolateral apophysis, bearing a short, robust macroseta; prolateral apophysis at 0.46 of metatarsus length and retrolateral apophysis at 0.28 (measured proximally to distally) (Figure 46f–h). Tarsi I bent prolaterally and moderately bowed ventrally in lateral view, broader at the apex than at the base; tarsi II, III ventrally incrassate and with scopulate setae; tarsi IV straight and with no scopulate setae. Macrosetae: Leg I Femur d1-1-1-1-1-1ap, dp1-1-1ap, dr1ap; tibia p1-1-1, r1-1-1-1-2 (robust); metatarsus p1 (elongate), p-apophysis-1 (robust, short), vp1ap, r-apophysis-1 (robust, short), vr1ap. Leg II femur d1-1-1-1-1ap, dp1-1ap, dr1ap; patella p1-1, tibia dp1-1-1, vr2-1-1-1ap, r1-1-1-1-1. Leg IV femur d1-1-1-1, dp1ap; tibia v1-1-1, metatarsus vp2ap, v1. Retrolateral distal preening comb with four long macrosetae, three shorter, one longer (Figure 46e). STC leg I, II with around 11 teeth, ITC with short tooth.

*Pedipalp:* Tibia 0.6 long, 0.5 wide; cymbium 0.5 long, 0.4 wide. Tibia curved ventrally, cymbium rectangular, straight edge apically. Bulb globular, embolus originating retrolaterally, gently curved, mid portion broad until around three-quarters of the length of the embolus, from where it tapers slightly to the tip; tip very elongated and sinuous apically (Figure 47a–c).

**Description.** ♀(based on paratype; SAMA NN31205). Total length 11.0.

*Cephalothorax:* 4.4 long, 3.1 wide, 2.5 high. Carapace uniform dark maroon-brown. Sternum golden-red-brown, lighter medially. Maxillae red-brown, paler apically. Labium red-brown, chelicerae dark maroon-brown (Figure 48a–c). Carapace elongated oblong, with

undulating lateral edges and forming a broad neck anteriorly, fovea a shallow indented pit. (Figure 48a). Labium elongate about four-fifths length of maxillae. Sternum elongate oval, inflated ventrally, with long, scattered dark setae. Pre-coxal triangles and indistinct rounded intercoxal extensions (Figure 48b). Carapace domed in lateral view, highest just prior to eye group; chelicerae robust and with a dense covering of fine macrosetae (Figure 48c). Eye group 1.0 wide, occupying 0.5 of the width of the carapace anteriorly; posterior eye row slightly recurved (Figure 48d).

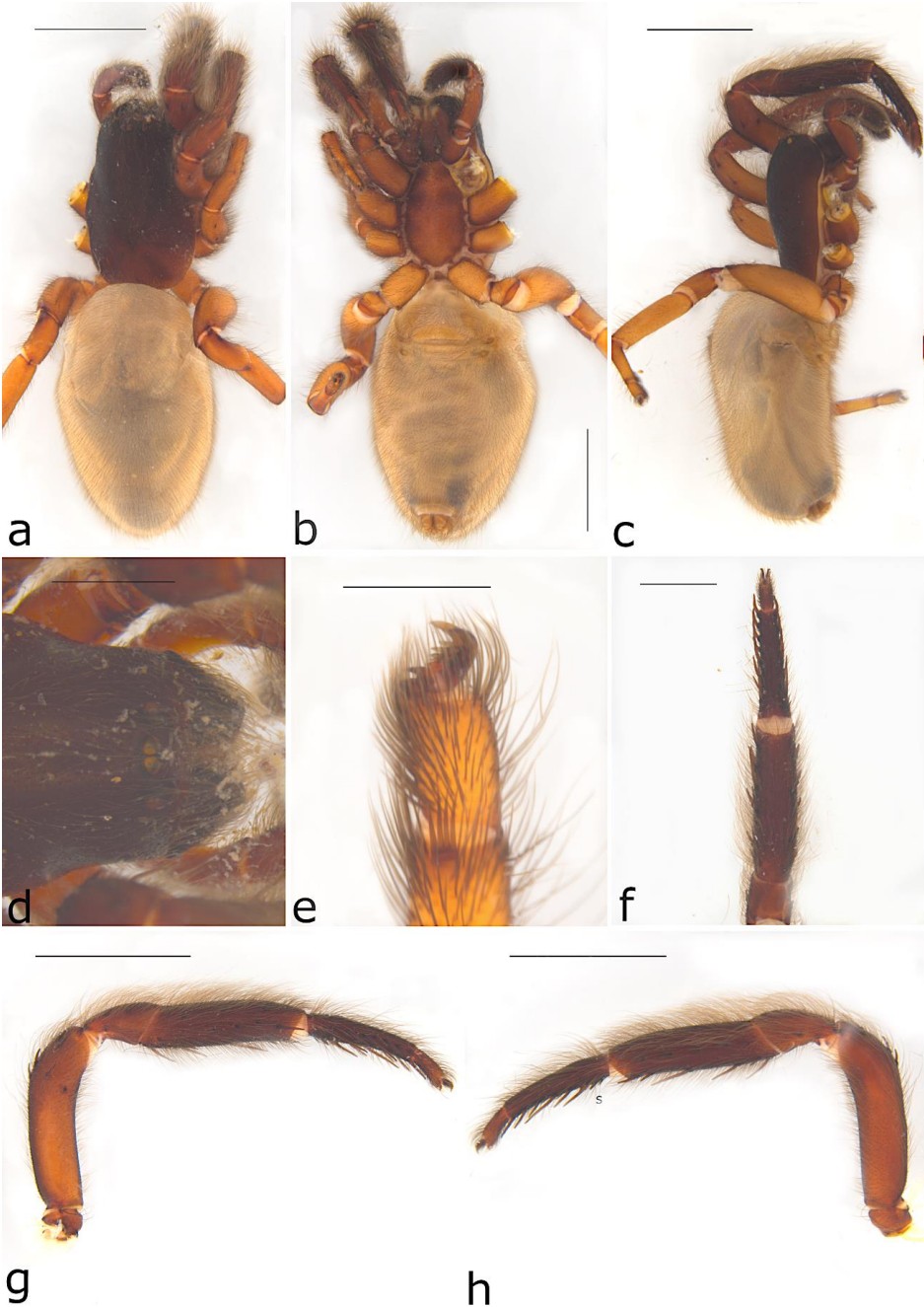

**Figure 48.** *Ariadna propria* sp. nov. ♀paratype (SAMA NN31205) from Melrose (SA): (**a**) habitus, dorsal view; (**b**) same, ventral view; (**c**) same, lateral view; (**d**) eyes, dorsal view; (**e**) left metatarsus IV, preening comb, retrolateral view; (**f**) left leg I, ventral view; (**g**) same, prolateral view; (**h**) same, retrolateral view; cluster of macrosetae indicated by arrow. Scale bars (**a–c**) = 2 mm, (**d**) = 1 mm, (**e**) = 0.5 mm, (**f**) = 1 mm and (**g,h**) = 2 mm.

*Abdomen:* 6.6 long. Dorsally light slate-grey; ventrally cream-grey, yellow-cream around book lungs and epigynal plate; with a covering of fine red-brown setae (Figure 48a–c).

*Legs:* Leg length ratio I > II > IV > III. Leg I total: 10.7: femur 3.5, patella 1.5, tibia 2.9, metatarsus 2.1, tarsus 0.7. Leg II 10.3: 3.2, 1.6, 2.7, 2.1, 0.7. Leg III 7.6: 2.5, 1.2, 1.7, 1.5, 0.7. Leg IV 8.2: 2.6, 1.4, 1.8, 1.7, 0.7. Leg I, II dark red-brown, legs III, IV orange-brown. Femur I strongly bowed in dorsal view. Tarsi short, stout, cylindrical. All leg segments covered in dense, gold-coloured setae. Figure 48f–h. Macrosetae: Leg I: femur dp2ap, d1-1-2ap, dr1ap; patella p1-1, r1; tibia p1-1-1-1, vp1-1-1-1-1ap, vr1-1-1-1-1-1, r1(short)-1(short)-1(short)-1(short)-1(short); metatarsus vr-basal cluster of short macrosetae v2-2-2-2-2-2-2-2-2. Leg II: femur dp1-1ap, d1ap, dr1ap; patella p1-1, r1; tibia p1-1-1, vp1-1-1-1ap; vr1(long)-1(long)-1(long)-1(short)-1(long)-1(short)ap, r1(short)-1(short)-1(short)-1(short)-1(short); metatarsus vr-basal cluster of short macrosetae v2-2-2-2-2-2-2-2-2 (Figure 48h). Leg IV: retrolateral distal preening comb with four macrosetae (Figure 48e). STC I and II with seven teeth, ITC with a small, indistinct tooth.

*Pedipalp:* Densely covered in dense black setae, with numerous macrosetae prolaterally.

*Genitalia:* (Based on paratype; SAMA NN30536a) The anterior receptaculum is bilobed, both lobes around the same size and more or less straight (Figure 49a,b).

**Variation.** Carapace lengths of females examined (*n* = 9) ranged from 4.2 to 4.7 (mean 4.4); colouration was consistent among specimens examined. For variations in leg macrosetae, see Table S3.

**Distribution.** This species was recorded from Melrose in the Southern Flinders Ranges, South Australia (Figure 39).

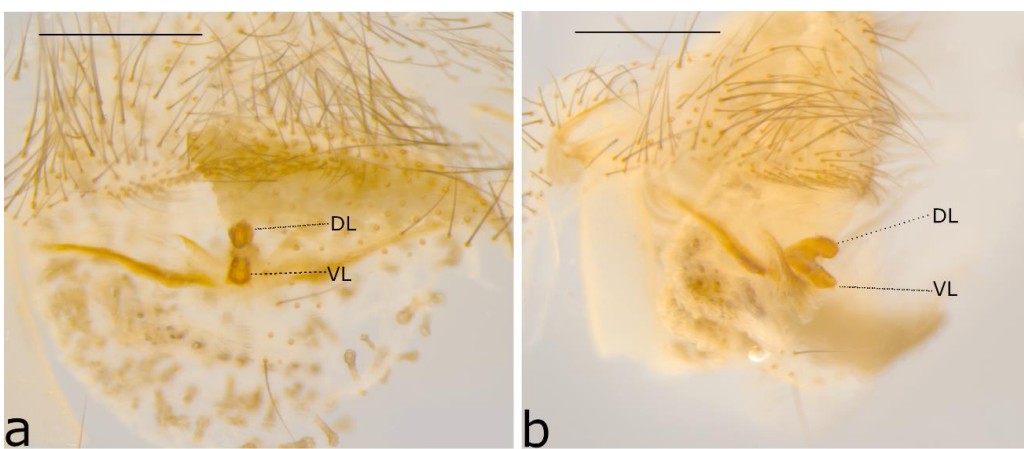

**Figure 49.** *Ariadna propria* sp. nov. ♀paratype (SAMA NN30536a) from Melrose (SA): anterior receptaculum (**a**) ventral view; (**b**) same, lateral view. Scale bar = 0.5 mm. DL = dorsal lobe, VL = ventral lobe.

### *Ariadna rutila* sp. nov.

Figure 50a–h, Figures 51a–c and 52

urn:lsid:zoobank.org:act:847B1C25-03BA-440F-8A4E-FB3CE62E1227

**Type material.** *Holotype* ♂ AUSTRALIA: *South Australia*, Muckera, 30.03° S; 130.02° E, pitfall trap, September 1984, coll. B. Guerin (SAMA NN30477).

**Etymology.** The specific epithet is taken from the Latin for 'golden red' and references the colour of the carapace of this species.

**Diagnosis.** Of the species with two well-pronounced apophyses situated in the medial section of metatarsus I, *A. rutila* sp. nov. can be differentiated from *A. curvata* sp. ov. by the shape of the retrolateral apophysis, which is distinctly raised in *A. curvata* sp. nov. but is low and well raised from the metatarsus in *A. rutila* sp. nov. (Figure 50f vs. Figure 31f). It can be separated from *A. bellatula* sp. nov. by the embolus, the mid portion of which is distinctly thickened at around half the embolus length in *A. bellatula* sp. nov. from

where it tapers to a sinuous tip but is a smooth curve in *A. rutila* sp. nov. (Figure 51a,b vs. Figure 30a,b).

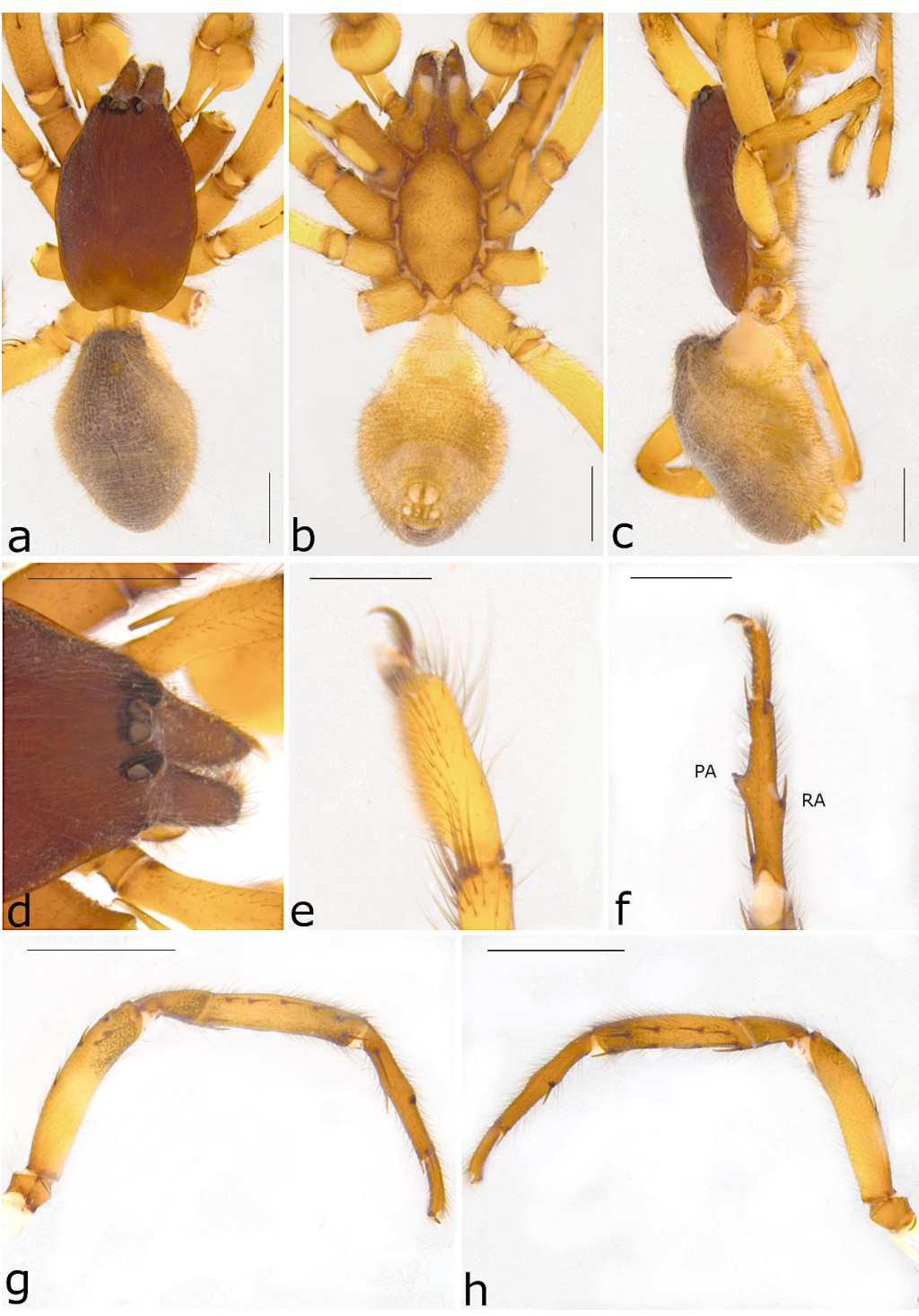

**Figure 50.** *Ariadna rutila* sp. nov. ♂holotype (SAMA NN30477) from Muckera (SA): (**a**) habitus, dorsal view; (**b**) same, ventral view; (**c**) same, lateral view; (**d**) eyes, dorsal view; (**e**) left metatarsus IV, preening comb, retrolateral view; (**f**) left leg I, ventral view; (**g**) same, prolateral view; (**h**) same, retrolateral view. Scale bars (**a**–**d**) = 1 mm, (**e**) = 0.5 mm, (**f**) = 1 mm and (**g**,**h**) = 2 mm.

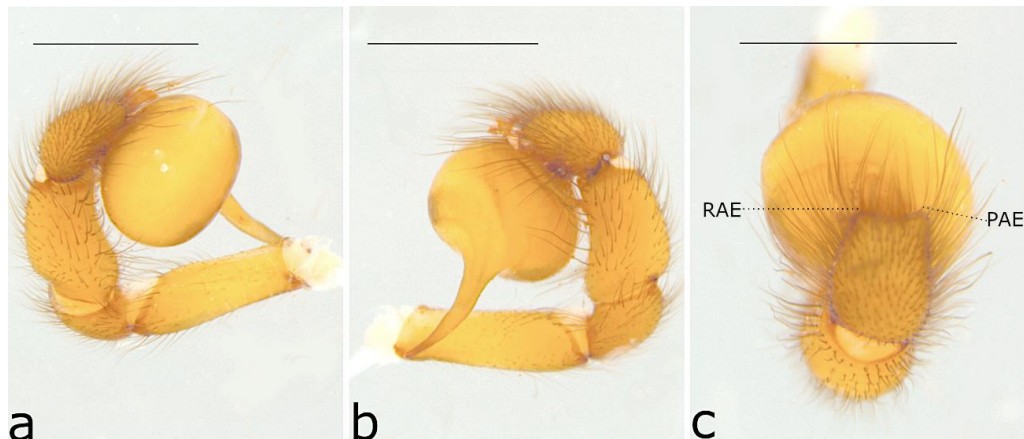

**Figure 51.** *Ariadna rutila* sp. nov. ♂ holotype (SAMA NN30477) from Muckera (SA): (**a**) left pedipalp, prolateral view; (**b**) same, retrolateral view and (**c**) same, cymbium. Scale bar = 1 mm.

**Description.** ♂ (based on holotype; SAMA NN30477). Total length 6.1.

*Cephalothorax:* 3.1 long, 2.1 wide, 1.5 high. Carapace golden red-brown. Sternum pale gold-orange, with darker orange at lateral edges and in pre-coxal triangles. Maxillae pale orange, white apically; labium pale orange, chelicerae dark golden-brown. Figure 50a–c. Carapace rounded rectangular and narrowing anteriorly; edges broadly undulating; fovea indented pit (Figure 50a). Labium about three-quarters length of maxillae. Sternum elongated oval with distinct pre-coxal triangles and with rounded, indistinct intercoxal extensions; with scattered, long, dark setae (Figure 50b). Flattened in lateral view, highest midway between fovea and eye group (Figure 50c). Eye group 0.6 wide, occupying 0.5 of the width of the carapace anteriorly; posterior eye row slightly recurved (Figure 50d).

*Abdomen*: 3.0 long. Dorsally mid violet-grey, paler grey laterally; ventrally pale cream-brown; with a covering of golden-brown setae (Figure 50a–c).

*Legs:* Leg length ratio I > II > IV > III. Leg I total 8.7: femur 2.6, patella 1.0, tibia 2.2, metatarsus 2.0, tarsus 0.9. Leg II 7.9: 2.4, 1.1, 2.1, 1.6, 0.7. Leg III 5.9: 2.1, 0.9, 1.2, 1.1, 0.6. Leg IV 7.8:2.5, 1.0, 1.9,1.6, 0.8. Legs yellow-brown, with scattered dark grey mottled patches on legs I–III. Femur I bowed in dorsal view. Metatarsus I straight, with a prolateral apophysis bearing a short, robust apophysis and a small retrolateral apophysis, bearing a long macroseta; prolateral apophysis at 0.54 of metatarsus length and retrolateral apophysis at 0.45 (measured proximally to distally) (Figure 50f–h). Tarsus I bent prolaterally and moderately bowed ventrally in lateral view, broader at the apex than at the base; tarsi II–IV ventrally incrassate and with scopulate setae; tarsi I with no scopulate setae. Macrosetae: Leg I Femur d1-1-1ap, dp2ap, dr1ap; tibia dp1-1-1-1, vp1-1ap, vr2-1-1-1ap, r1-1-1-1; metatarsus p1 (strong, elongate), p-apophysis-1 (robust, short), vp1ap, r-apophysis-1 (long), vr1ap. Leg II femur d1-1-1-1-1ap, dp1-1ap, dr1ap; tibia p1-1-1, vp1ap, vr2-1-1-1ap, r1-1-1-1-1; metatarsus vp1-1-1ap, vr1-1-1ap, r1-1-1-1. Leg IV femur d1-1-1-1-1-1; metatarsus vp2ap. Retrolateral distal preening comb with four long macrosetae, equal length (Figure 50e). STCs with around nine teeth, ITC short, robust; with a short tooth.

*Pedipalp:* Tibia 0.7 long, 0.5 wide; cymbium 0.6 long, 0.4 wide. Tibia short, curved ventrally, cymbium squared. Bulb large, globular, embolus originating retrolaterally, gently curved, mid portion broad until half the length of the embolus, from where it tapers slightly to the tip; tip broad (Figure 51a–c).

**Distribution:** This species is only known from the type locality, in the arid far north of South Australia (Figure 52).

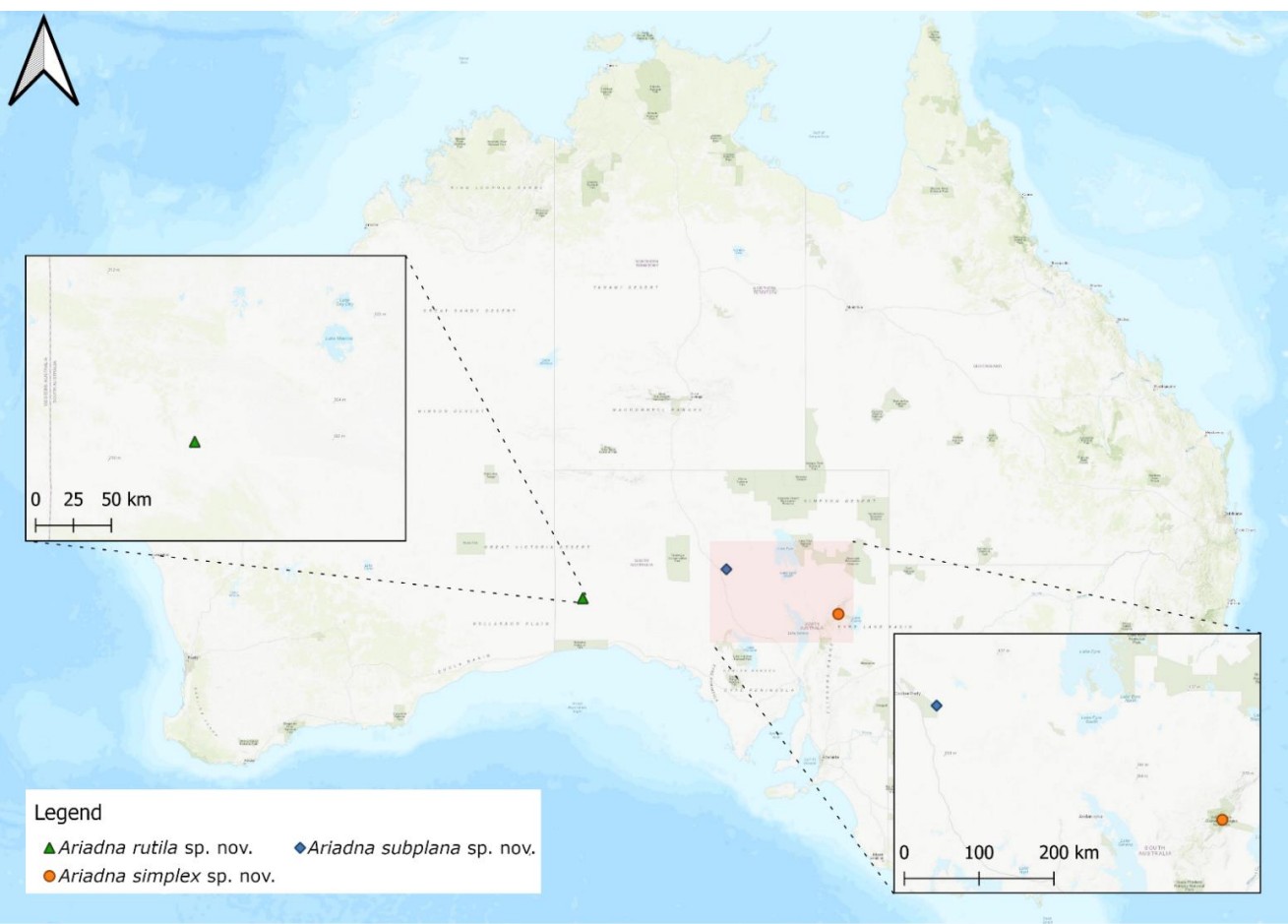

**Figure 52.** Map showing the currently known distribution of *A. rutila* sp. nov., *A. simplex* sp. nov., *A. subplana* sp. nov.

### *Ariadna simplex* **sp. nov.**

Figures 52, 53a–h and 54a–c

urn:lsid:zoobank.org:act:FAAF599F-303F-4C6C-9266-EAAE1688B959

**Type material.** *Holotype* ♂ AUSTRALIA: *South Australia*, 2 km E Weetootla Well, Balcanoona Creek, Gammon Ranges National Park, 30.52° S, 139.18° E, on trunks and branches, 7 May 1989, coll. D. Hirst (SAMA NN30482).

**Etymology.** The specific epithet is a Latin adjective meaning 'simple' and refers to the simplified first metatarsus of this species, which has no apophyses and few macrosetae.

**Diagnosis.** *Ariadna simplex* sp. nov. can be differentiated from most other species of *Ariadna* with no apophyses on metatarsus I by the shape of the pedipalp cymbium, which is squared and blunt in *A. simplex* sp. nov., lacking apical prolateral and/or retrolateral extensions (Figure 53f). It can be differentiated from *A. tria* sp. nov., which also has a blunt cymbium apically by the relative length and shape of the pedipalp tibia, which is elongate, the length being 1.8 times the width in *A. tria* sp. nov., but in *A. simplex* sp. nov., the length is 1.2 times the width (Figure 54c vs. Figure 21c).

**Description.** ♂ (based on holotype; SAMA NN30482). Total length 4.4.

*Cephalothorax:* 2.2 long, 1.4 wide, 1.2 high. Carapace light orange-brown, darker orange colouration demarking the cephalic area. Sternum pale orange-cream, with darker orange pre-coxal triangular extensions. Maxillae orange-brown, paler apically, labium darker orange-brown, chelicerae orange-brown(Figure 53a–c). Carapace squared oval, narrowing only slightly anteriorly blunt posteriorly. With rows of short setae, running longitudinally from fovea to eye group; fovea a shallow indented pit (Figure 53a). Labium about three quarters length of maxillae. Sternum elongated oval with pre-coxal triangles

and with smaller, rounded intercoxal extensions, with scattered, brown setae (Figure 53b). In lateral view, carapace flattened; chelicerae semi-porrect (Figure 53c). Eye group 0.5 wide, occupying 0.5 of the width of the carapace anteriorly; posterior eye row slightly recurved. (Figure 53d).

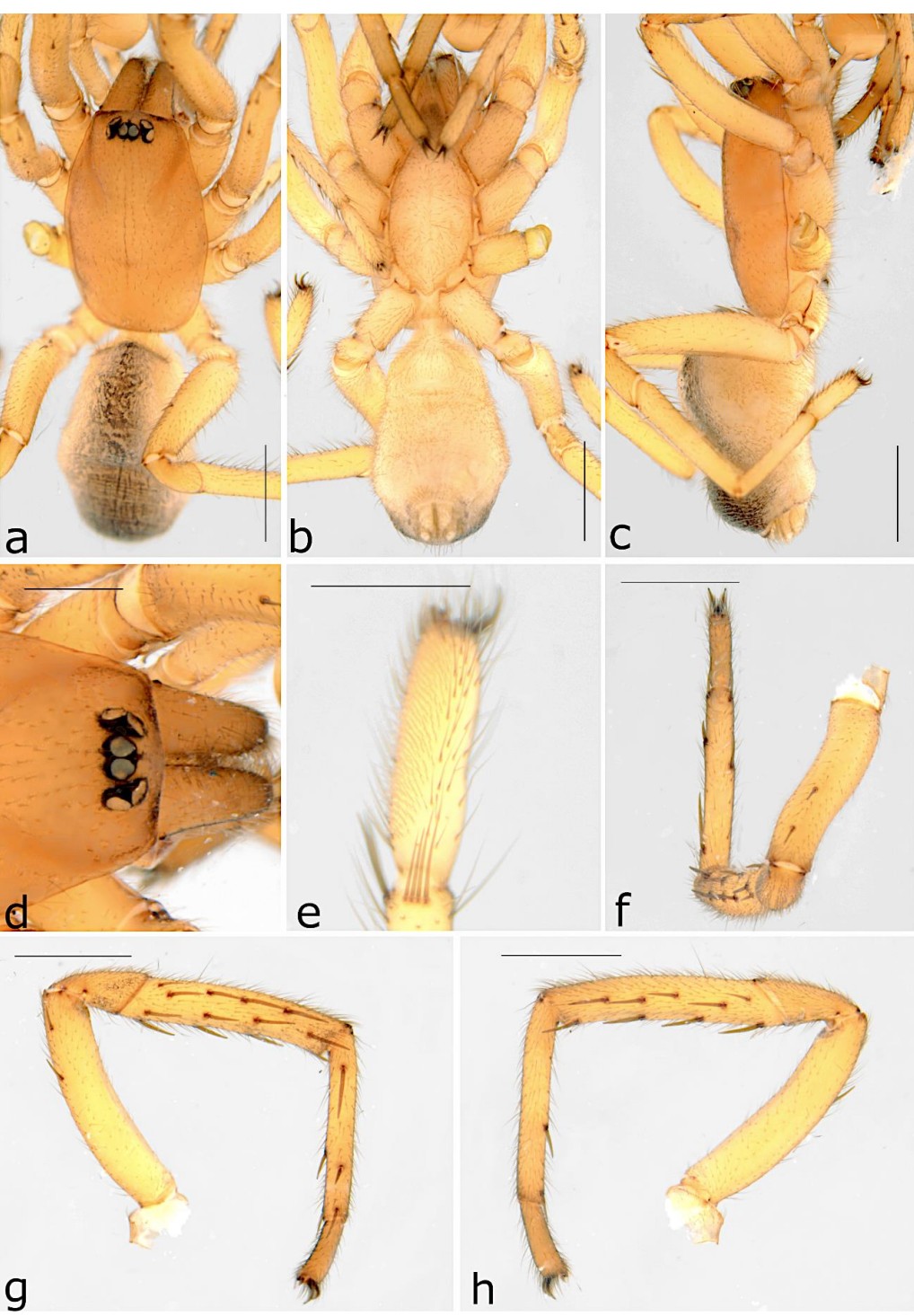

**Figure 53.** *Ariadna simplex* sp. nov. ♂holotype (SAMA NN30482) from the Gammon Ranges (SA): (**a**) habitus, dorsal view; (**b**) same, ventral view; (**c**) same, lateral view; (**d**) eyes, dorsal view; (**e**) left metatarsus IV, preening comb, retrolateral view; (**f**) left leg I, ventral view; (**g**) same, prolateral view; (**h**) same, retrolateral view. Scale bars (**a–c**) = 1 mm, d, (**e**) = 0.5 mm and (**f–h**) = 1 mm.

*Abdomen:* 2.2 long. Dorsally dark slate-grey, laterally cream and ventrally pale cream; with a covering of fine dark brown setae (Figure 53a–c).

*Legs:* Leg length ratio II > I > IV > III. Leg I total 6.6: femur 1.9, patella 0.8, tibia 1.6, metatarsus 1.6, tarsus 0.7. Leg II 6.8: 2.0, 0.8, 1.9, 1.4, 0.7. Leg III 4.9: 1.6, 0.6, 1.1, 0.9, 0.7. Leg IV 5.8: 1.9, 0.7, 1.5, 1.1, 0.6. Legs orange-brown, leg patella and tibia darker orange-brown. Femur I slightly bowed in dorsal view. Metatarsus I straight in ventral view, with no apophyses (Figure 52f–h). Tarsi I slightly curved ventrally in lateral view, slightly broader at apex than at base and with spare scopulate setae apically. Tarsi II, III straight in lateral view, with sparse scopulate setae, tarsi IV inflated ventrally, with scopulate setae along length Macrosetae: Leg I: femur d1-1; tibia p1-1-1-1, vp1-1-1-1ap, vr1-1-1-1, r1-1-1-1-1ap; metatarsus p1 (long)-1-1ap, vr1-1ap. Leg II: femur d1-1-2ap, tibia p1-1-1-1, vp1-1-1-1ap, vr1-1-1-1ap, vr1-1-1-1; metatarsus vp1-1-1ap, vr1-1-1ap, r1-1. Leg IV: femur d1-1; retrolateral distal preening comb with four macrosetae, one longer and three shorter (Figure 53e). STC I, II with around 14 teeth, ITC with small tooth.

*Pedipalp:* Tibia 0.5 long, 0.4 wide; cymbium 0.5 long, 0.4 wide. Pedipalp tibia short, stout. Cymbium squared, with straight edge apically, lacking prolateral or retrolateral apical extensions. The bulb is globular. The embolus originates retrolaterally from the bulb, the mid portion is broad until around half the length of the embolus, from where it tapers apically to tip; tip hooked and sinuous apically (Figure 54a–c).

**Distribution.** This species is only known from the type locality near Weetootla Well, Balcanoona Creek, Vulkathunha-Gammon Ranges National Park, in the Northern Flinders Ranges region of South Australia (Figure 52).

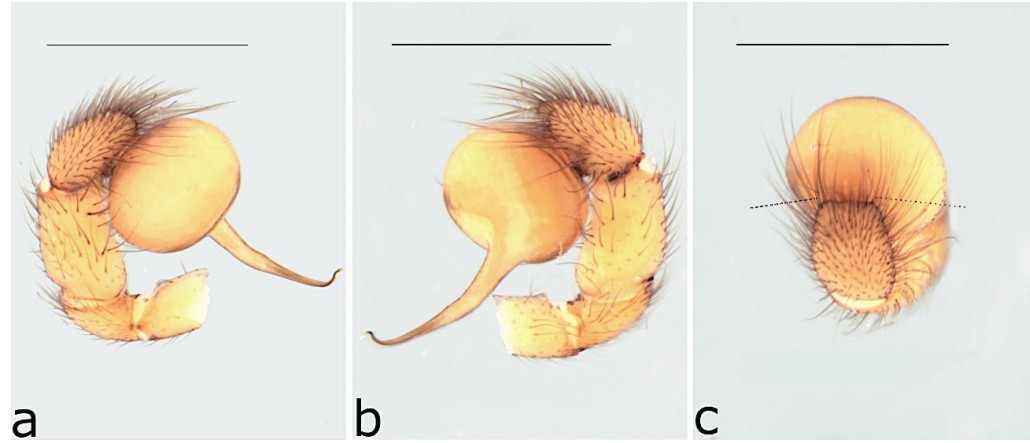

**Figure 54.** *Ariadna simplex* sp. nov. ♂ holotype (SAMA NN30482) from the Gammon Ranges (SA): (**a**) left pedipalp, prolateral view; (**b**) same, retrolateral view; (**c**) same, cymbium. Scale bar = 1 mm.

### *Ariadna subplana* **sp. nov.**

Figures 52, 55a–h and 56a–c

urn:lsid:zoobank.org:act:8B305581-9682-4132-8B40-FAD1FF49BF20

**Type material.** *Holotype* ♂ AUSTRALIA: *South Australia*, 14.5 km WNW Backadinna Hill, 29.13° S, 135.16° E, 7 October 1995, Department of Environment Water and Natural Resources, Stony Desert Biological Survey (SAMA NN30479).

**Etymology.** The specific epithet is taken from the Latin preposition 'sub' meaning 'somewhat' and the Latin adjective 'plana' meaning 'flat' and refers to the small, low apophyses on metatarsus I of this species.

**Diagnosis.** *Ariadna subplana* sp. nov. can be differentiated from other species with the prolateral apophysis being situated in the apical third of metatarsus I, by the prolateral apophysis being a low, broad pyramidal mound, little raised from the metatarsus, whereas in *A. arenacea* sp. nov., *A. diucrura* sp. nov., *A. flavescens* sp. nov., *A. inflata* sp. nov., *A. pollex* sp. nov., *A. ungua* sp. nov. and *A. valida* sp. nov. the prolateral apophysis is well-defined (Figure 55f vs. Figure 26f, Figure 35f, Figure 37f, Figure 40f, Figure 44f, Figure 65f and Figure 69f).

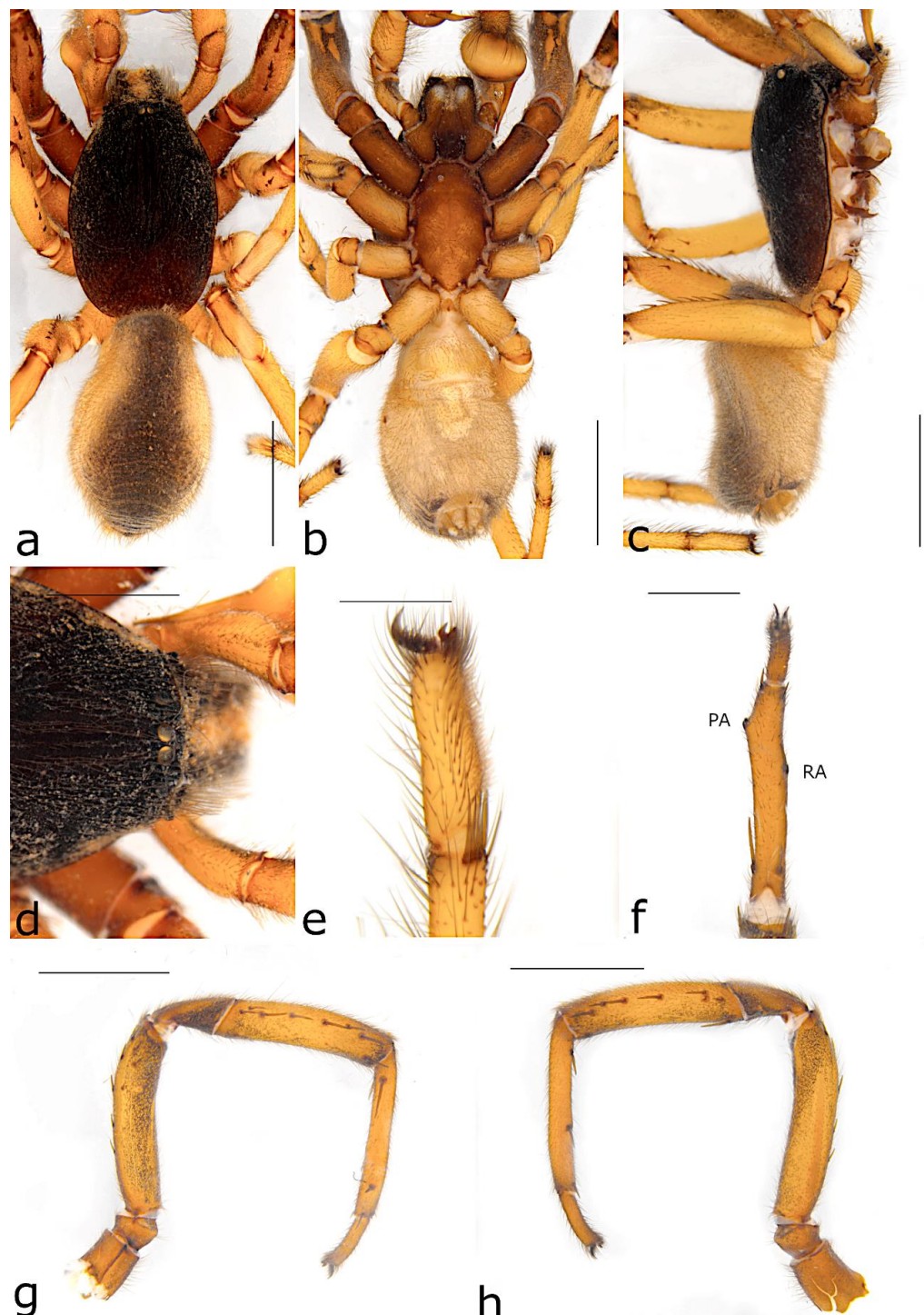

**Figure 55.** *Ariadna subplana* sp. nov. ♂ holotype (SAMA NN30479) from Stony Desert (SA): (**a**) habitus, dorsal view; (**b**) same, ventral view; (**c**) same, lateral view; (**d**) eyes, dorsal view; (**e**) left metatarsus IV, preening comb, retrolateral view; (**f**) left leg I, ventral view; (**g**) same, prolateral view; (**h**) same, retrolateral view. Scale bars (**a**–**c**) = 2 mm, (**d**) = 1 mm, (**e**) = 0.5 mm, (**f**) = 1 mm and (**g**,**h**) = 2 mm.

**Description.** ♂ (based on holotype; SAMA NN30479). Total length 7.2.

*Cephalothorax:* 3.5 long, 2.4 wide, 1.9 high. Carapace dark black, red-brown, black anteriorly. Sternum golden-orange, with darker orange at lateral edges and in pre-coxal triangles and intercoxal extensions. Maxillae dark orange-brown, white apically; labium orange-brown, chelicerae dark red-brown (Figure 55a–c). Carapace rounded oval and narrowing anteriorly; fovea a shallow indented pit (Figure 55a). Labium about four fifths

length of maxillae. Sternum elongated oval with pre-coxal triangles and with distinct triangular intercoxal extensions; with scattered, long, dark setae (Figure 55b). Flattened in lateral view, highest midway between fovea and posterior eye row (Figure 55c). Eye group 0.8 wide, occupying 0.6 of the width of the carapace anteriorly; posterior eye row slightly recurved (Figure 55d).

*Abdomen:* 3.7 long. Dorsally dark violet-grey medially, paler cream laterally; ventrally uniform pale cream-grey; with scattered long and shorter red-brown setae (Figure 55a–c).

*Legs:* Leg length ratio II > IV > I > III. Leg I total 8.5: femur 2.5, patella 1.0, tibia 2.2, metatarsus 2.1, tarsus 0.7. Leg II 9.9: 3.0, 1.0, 2.4, 2.5, 1.0. Leg III 7: 2.3, 0.8, 1.4, 1.6, 0.9. Leg IV 9.1: 2.9, 1.1, 2.3, 1.9, 0.9. Legs yellow-orange with dark grey mottled patches along prolateral edge of legs I and II. Femur I slightly bowed in dorsal view. All legs robust, stout. Metatarsus I sinuous, with a somewhat flattened but distinct prolateral apophyses with a stout macroseta at 0.81 of the metatarsus length (measured proximally to distally); retrolateral apophysis flattened and indistinct, situated at 0.59 of the length of the metatarsus, with a large macroseta on the raised base (Figure 55f–h). Tarsus I moderately bowed ventrally in lateral view, broader at the apex than at the base; tarsi II–IV with scopulate setae; tarsi I with no scopulate setae. Macrosetae: Leg I Femur dp3ap, d1-1-1-1-1ap, dr1ap; tibia p2-1-1-1-1-1, vp1-1ap, vr1-1ap, r1-1-1-1-1-1ap; metatarsus p1, p-apophysis, p1ap, r1-1-1ap. Leg II femur dp2ap, d1-1-1-1ap, dr1ap; tibia p1-1-1, vp1ap, vr1-1-1-1ap, r1-1-1-1-1-2; metatarsus vp1-1-1ap, vr1-1-1ap, r1-1-1. Leg IV femur d1-1-1-1-1-1-1-1-1-1; tibia p1, v1; metatarsus vr2ap. Retrolateral distal preening comb with four long macrosetae, equal length (Figure 55e). STCs with around 11 teeth, ITC short, robust; with a short tooth.

*Pedipalp:* Tibia 0.8 long, 0.6 wide; cymbium 0.7 long, 0.5 wide. Tibia short, rectangular and curved ventrally, cymbium squared. Bulb large globular, embolus originating retrolaterally, mid portion broad to around three-quarters the length of the embolus, where there is a brief prolateral expansion, from which it tapers apically to tip; tip hooked at apex and slightly sinuous. (Figure 56a–c).

**Distribution.** This species is only known from the type locality in the Stony Desert near Backadinna Hill in the NW of South Australia (Figure 52).

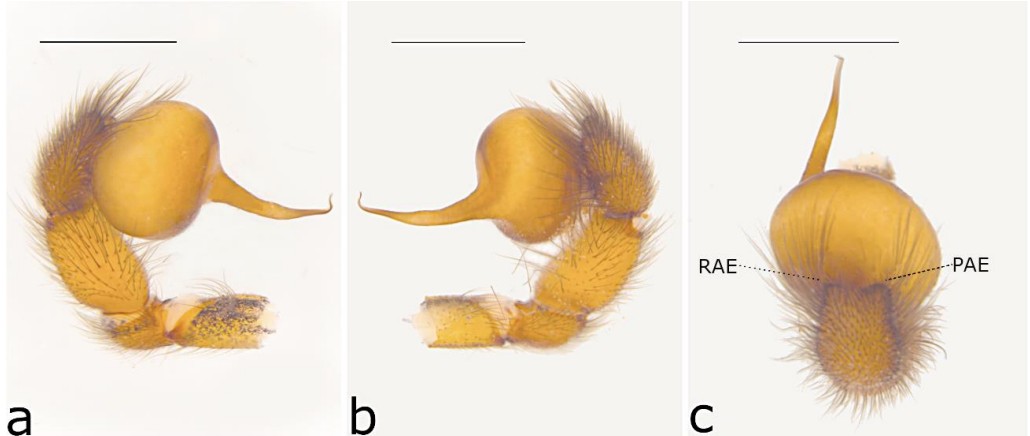

**Figure 56.** *Ariadna subplana* sp. nov. ♂holotype (SAMA NN30479) from Stony Desert (SA): (**a**) left pedipalp, prolateral view; (**b**) same, retrolateral view; (**c**) same, cymbium. Scale bar = 1 mm.

### *Ariadna tangara* Marsh, Baehr, Glatz & Framenau, 2018

Figure 57a–h, Figure 58a–c, Figure 59a–h, Figure 60a,b and Figure 61
urn:lsid:zoobank.org:act:D3FE46A5-3843-4796-8F39-AE463BE46BCD

**Type material.** *Holotype* ♂ AUSTRALIA: *South Australia*, Kangaroo Island, American River, Tangara Drive, 35.78732° S, 137.76703° E, in tube web under bark of *Eucalyptus diversifolia*, under bark, 12 June 2017, coll. J. Marsh (SAMA NN29863).

*Paratypes:* 1♀same data as holotype (SAMA NN29864); 1♂American River Reserve, Tangara Drive, American River, 35.78805° S, 137.76959° E, in web under bark of *Melaleuca*

*halmaturorum*, coll. 5 December 2017 and raised to maturity in captivity, coll. J. Marsh (LIB ZMH-A0003053); 1♀Tangara Drive, American River, 35.80548° S, 137.79443° E, in crevice of bark of old apple tree, in tube web with spiderlings, coll. J. Marsh (LIB ZMH-A0003054).

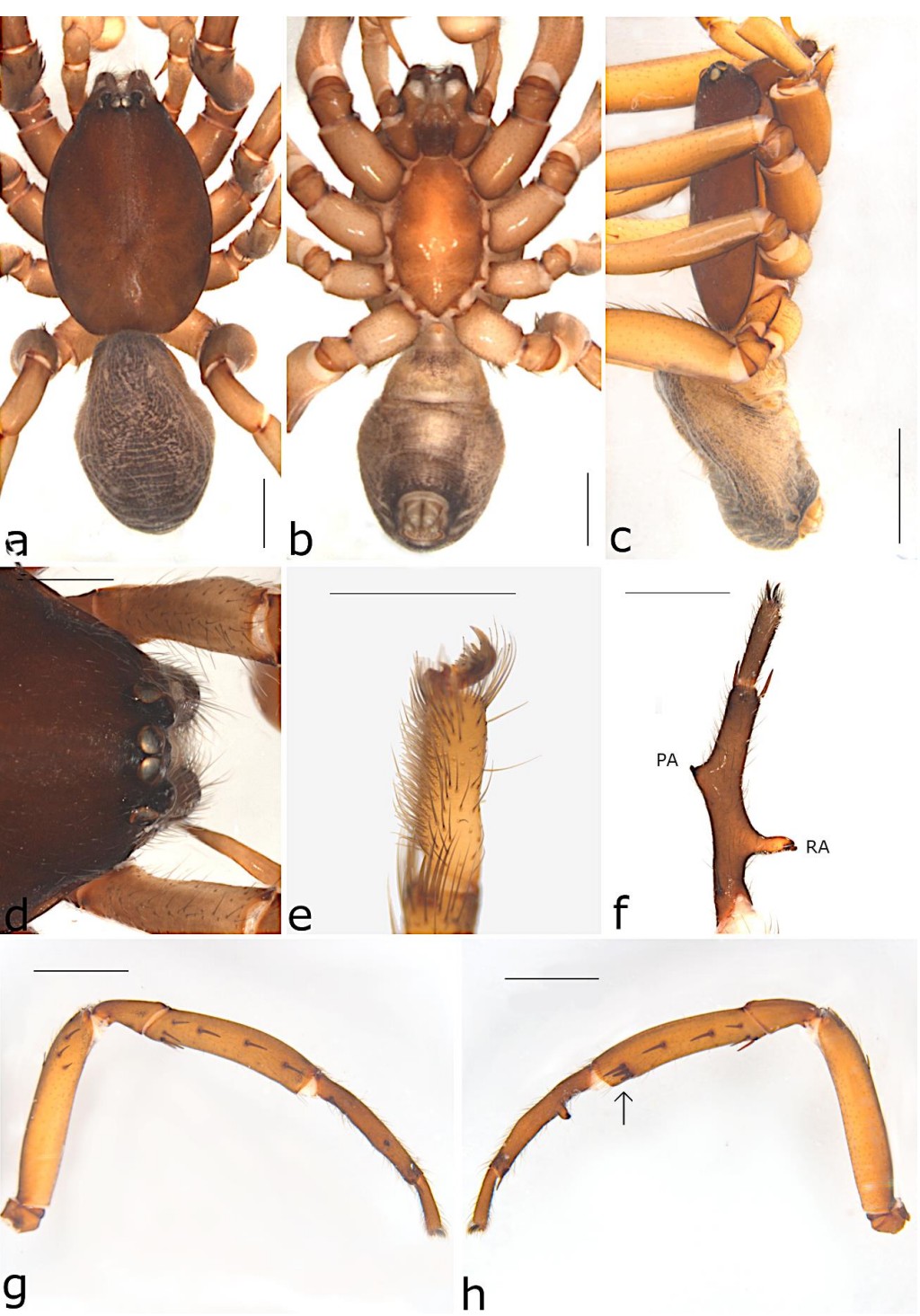

**Figure 57.** *Ariadna tangara* Marsh, Baehr, Glatz & Framenau, 2018. ♂holotype (SAMA NN29863) from Kangaroo Island (SA): (**a**) habitus, dorsal view; (**b**) same, ventral view; (**c**) same, lateral view; (**d**) eyes, dorsal view; (**e**) left metatarsus IV, preening comb, retrolateral view; (**f**) left leg I, ventral view; (**g**) same, prolateral view; (**h**) same, retrolateral view, arrow indicating group of macrosetae on raised base. Scale bars (**a–h**) = 1 mm.

**Other material examined.** AUSTRALIA: *South Australia*, 1♀Tangara Drive, American River, 35.78718° S, 137.76699° E, 8 July 2017, coll. J. Marsh (SAMA NN29896); 1♀same data, except −35.80548° S, 137.79443° E, 25 January 2018 (SAMA NN29901); 1♀Pelican Lagoon walking Trail, American River, 35.79649° S, 137.75096° E, 2 November 2017, coll. J. Marsh (SAMA NN29898); 2♀Antechamber Bay, Chapman River, 35.78521° S, 138.06673° E, 9 September 2017, coll. J. Marsh (SAMA NN29884, NN29885); 1♀same data, except 35.78048° S, 137.59085° E, 21 August 2017 (SAMA NN29894); 1♀same data, except 10 December 2017 (SAMA NN29900); 1♂same data, except coll. 24 January 2018, matured in captivity, May 2018 (SAMA NN29904); 1♀Loverings Road, MacGillivray, 35.89415° S, 137.56953° E, 10 January 2018, coll. J. Marsh (SAMA NN29902); 1♀Three Chain Road, MacGillivray, 35.91067° S, 137.55049° E, 7 August 2017, coll. J. Marsh (SAMA NN29887); 2♀same data, except 30 January 2018 (SAMA NN29903); 1♀Flinders Chase National Park, Kangaroo Island, 35.94° S, 136.78° E, June 2018, coll. J. Marsh (JRM xxx); 1♂Coromandel Valley, Mount Lofty Ranges, 35.03° S, 138.63° E, 17 August 1996, coll. L.N Nicolson (SAMA NN30551). 1♂, 3♀Wilpena Pound, Wilpena Creek, 31.34° S, 139.05° E, 24 April 1987, coll. D. Hirst (SAMA NN30554).

*Victoria*, 1♂Abbottsford, 37.80° S, 145.00° E, 27 April 1990, coll. R. Vella (K-14584); 1♂, 1♀Carnegie, −37.89° S, 145.05° E, 20 July 1947 (K-14590); 1♂Sandringham, 37.95° S, 145.00° E, 20 August 1981, coll. Mrs Janson (K-14576); 1♂, 1♀Newstead, 37.0873° S, 144.08017° E, in the cracks of bark of *Eucalyptus* sp. tree, in dry creek bed, coll. J. Marsh, 19 January 2019 (SAMA NN31204).

*Tasmania*, 1♂1♀King Meadows, Tasmania, 41.4673° S, 147.1542° E, 24 May 2019, coll. J. Douglas (QVMAG QVM:2017:13:0229).

**Diagnosis.** *Ariadna tangara* can be differentiated from all other described species of *Ariadna* in Australia by the presence of a group of three large macrosetae on a shared, raised base, apically on the retrolateral surface of tibia I of males (indicated by arrow in Figure 57h). Females can be distinguished from other species by the genitalia, with the ventral lobe being more or less straight and around 1.25 times the length of the dorsal lobe (Figure 60a,b) and without an apical ventral fold as seen in *A. woinarskii* sp. nov. (Figure 76a,b). Of the closest Tasmanian species, it can be differentiated from *Ariadna major* Hickman, 1929 by the number of macrosetae in metatarsus IV, of which there are seven in *A. major*, but four in *A. tangara* (Figure 59e; JRM pers obs.).

**Remarks.** *Ariadna tangara* was described in detail recently [4].

**Distribution:** This species has the largest recorded distribution of any known species of Ariadna in Australia, ranging from southern South Australia, Victoria and Tasmania (Figure 61).

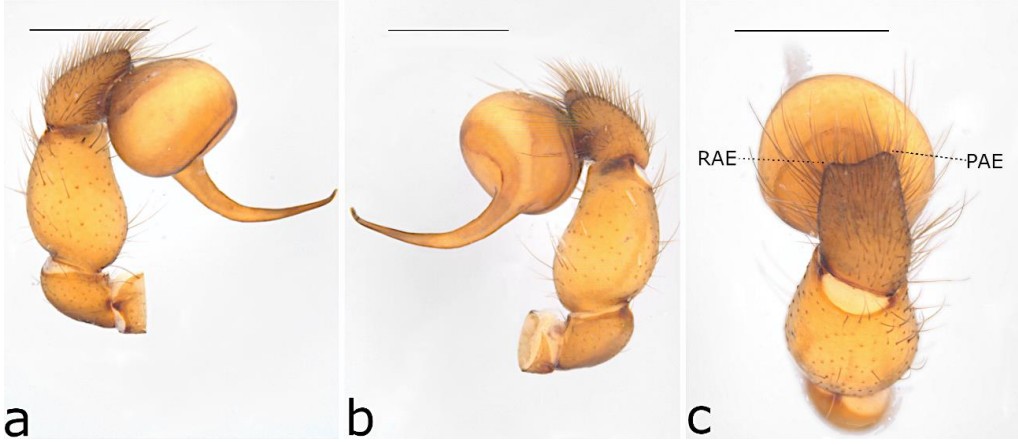

**Figure 58.** *Ariadna tangara* Marsh, Baehr, Glatz & Framenau, 2018. ♂holotype (SAMA NN29863) from Kangaroo Island (SA): (**a**) left pedipalp, prolateral view; (**b**) same, retrolateral view; (**c**) same, cymbium. Scale bar = 1 mm.

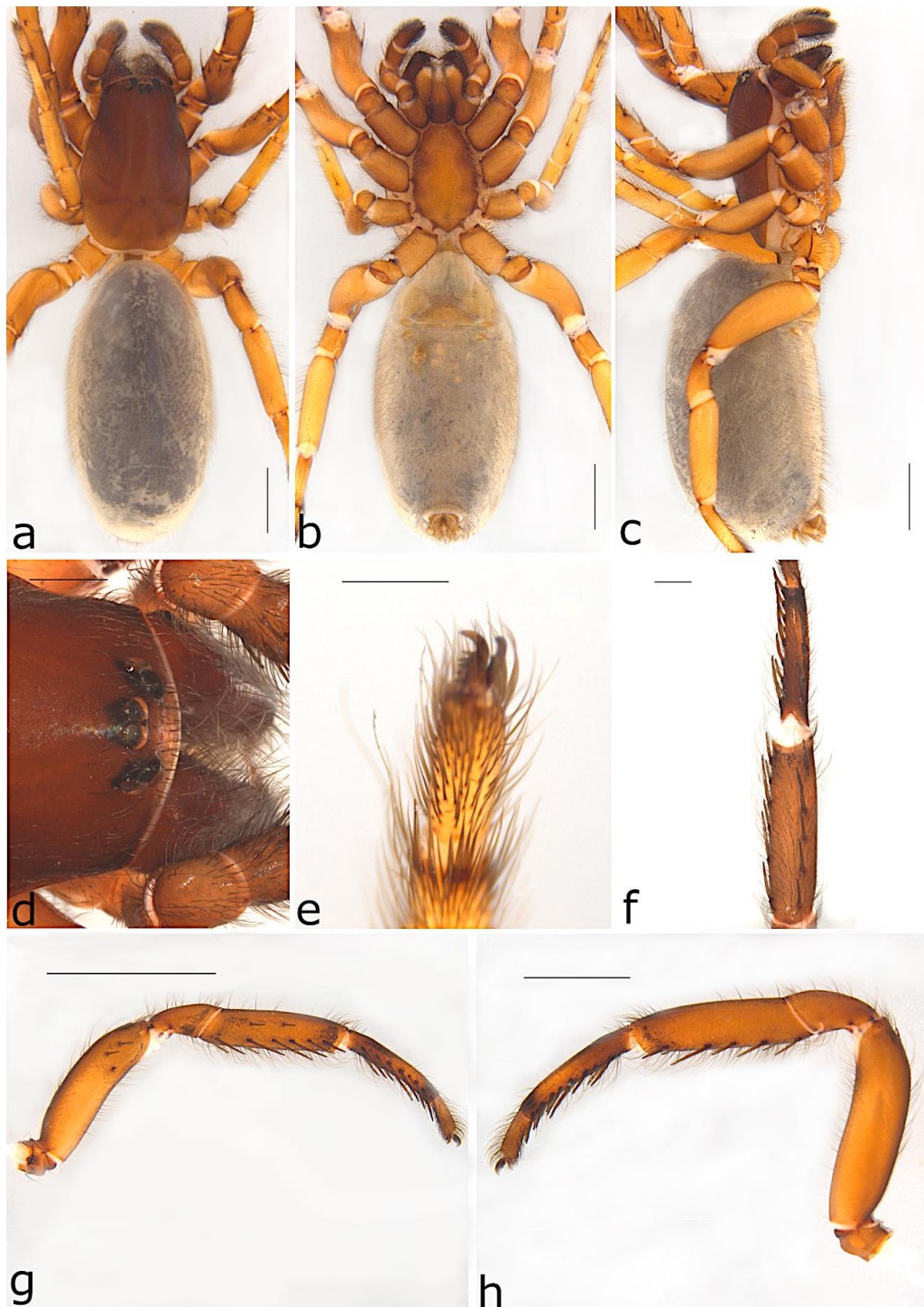

**Figure 59.** *Ariadna tangara* Marsh, Baehr, Glatz & Framenau, 2018. ♀holotype (SAMA NN29864) from Kangaroo Island (SA): (**a**) habitus, dorsal view; (**b**) same, ventral view; (**c**) same, lateral view; (**d**) eyes, dorsal view; (**e**) left metatarsus IV, preening comb, retrolateral view; (**f**) left leg I, ventral view; (**g**) same, prolateral view; (**h**) same, retrolateral view. Scale bars (**a**–**c**) = 1 mm, (**d**–**f**) = 0.5 mm, (**g**) = 2 mm and (**h**) = 1 mm.

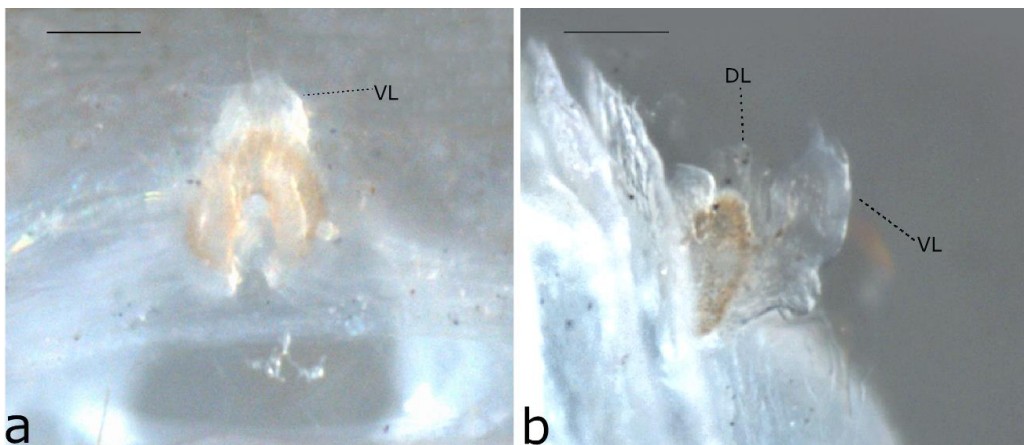

**Figure 60.** *Ariadna tangara* Marsh, Baehr, Glatz & Framenau, 2018. ♀holotype (SAMA NN29864) from Kangaroo Island (SA): anterior receptaculum (**a**) ventral view; (**b**) same, lateral view. Scale bar = 0.2 mm. DL = dorsal lobe, VL = ventral lobe.

**Figure 61.** Map showing the currently known distribution *A. tangara*, illustrating the distribution of the Victorian clade and the South Australian and Tasmanian clade.

### *Ariadna una* sp. nov.

Figure 62a–h, Figures 63a–c and 64

urn:lsid:zoobank.org:act:489F872F-50DC-4D20-AE8C-FCDAFC7AF3C3

**Type material.** *Holotype* ♂ AUSTRALIA: *South Australia:* 23 km WSW Illintjitja, 26.27722° S, 130.17278° E, 9–13 May 1993, pitfall trap, Pitjantjatjara Lands Survey (SAMA NN10578).

**Etymology.** The specific epithet is a Latin adverb meaning 'together with', 'at the same time' and refers to the apparently sympatric distribution of this species with *Ariadna valida* sp. nov.

**Diagnosis.** *Ariadna una* sp. nov. can be differentiated from *A. propria* sp. nov., which also has the retrolateral apophysis situated in the first quarter of the metatarsus, by the shape of the cymbium apically, which has a has a rounded PAE in *A. una* sp. nov. but is straight and squared apically in *A. propria* sp. nov. (Figure 63c vs. Figure 47c). *Ariadna una* sp. nov. is clearly separated from *A. valida* sp. nov., which it is apparently sympatric by the relative position of the retrolateral apophysis on metatarsus I, which is located in the apical third of the metatarsus in *A. una* sp. nov., whilst the prolateral apophysis is located at 0.4 of the metatarsus length, whilst in *A. valida* sp. nov., the retrolateral apophysis is located at 0.5 of the metatarsus length and the prolateral apophysis at 0.6 (Figure 62f vs. Figure 69f). Molecular analyses revealed *A. una* sp. nov. to be closest to *A. bellatula* sp. nov. These species differed morphologically and so were treated as different species. Differences between *A. bellatula* sp. nov. and *A. una* sp. nov. were recorded in the relative position of the apophyses on metatarsus I, with *A. una* sp. nov. prolateral apophysis at 0.44, retrolateral apophysis 0.27 of the metatarsus length; *A. bellatula* sp. nov. the prolateral apophysis was at 0.54, retrolateral apophysis at 0.44 of the metatarsus length the relative shape of the apophyses, which are subequal in *A. una* sp. nov., but the retrolateral apophysis is smaller than the prolateral apophysis in *A. bellatula* sp. nov. (Figure 62f vs. Figure 29f), and by the pedipalp cymbium, which has an enlarged PAE compared to the RAE in *A. una* sp. nov., whereas the PAE and RAE are subequal in *A. bellatula* sp. nov. (Figure 63c vs. Figure 30c).

**Description.** ♂ (based on holotype; SAMA NN10578). Total length 7.1.

*Cephalothorax:* 3.3 long, 2.2 wide, 1.7 high. Carapace golden-red-brown, darker anteriorly. Sternum pale golden-orange, with darker orange at lateral edges and in pre-coxal triangles and intercoxal extensions. Maxillae dark orange-brown, white apically; labium orange-brown, chelicerae dark golden-brown (Figure 62a–c). Carapace rounded oval and narrowing anteriorly; edges gently undulating; fovea a shallow indentation (Figure 62a). Labium about four fifths length of maxillae. Sternum elongated oval with distinct pre-coxal triangles and with rounded, indistinct intercoxal extensions; with scattered, long, dark setae (Figure 62b). Carapace flattened in lateral view, highest just posterior to eye group (Figure 62c). Eye group 0.7 wide, occupying 0.5 of the width of the carapace anteriorly; posterior eye row slightly recurved (Figure 62d).

*Abdomen*: 3.8 long. Dorsally dark violet-grey, paler cream laterally; ventrally pale cream-brown; with sparse red-brown setae (Figure 62a–c).

*Legs:* Leg length ratio I > II > IV > III. Leg I 9.8: femur 2.9, patella 1.1, tibia 2.6, metatarsus 2.3, tarsus 0.9. Leg II 8.9: 2.5, 1.0, 2.3, 2.2, 0.9. Leg III 6.8: 2.2, 0.8, 1.4, 1.5, 0.9. Leg IV 8: 2.4, 0.9, 2.2, 1.5, 1.0. Legs orange-brown, with dark grey mottled patches along prolateral edges of legs I–III. Femur I bowed in dorsal view. Legs robust, stout. Metatarsus I sinuous, with a prolateral and a retrolateral apophysis, both bearing a short, stout macroseta; prolateral apophysis at 0.45 of metatarsus length and retrolateral apophysis at 0.27 (measured proximally to distally); both apophyses are pronounced and pyramidal in shape (Figure 62f–h). Tarsus I bent prolaterally and moderately bowed ventrally in lateral view, broader at apex than at base; tarsi II–IV ventrally incrassate and with scopulate setae; tarsi I with no scopulate setae. Macrosetae: Leg I Femur dp2ap, dr1ap; tibia dp1-1-1-1, vp1-1-1-1ap,vr1-1-1-1ap, r1-1-1-1-1-1; metatarsus p1 (strong, elongate), p-apophysis stout macroseta, vp1ap, r-apophysis,vr1ap. Leg II femur dp1ap, d1, d1ap, dr1ap; tibia p1-1-1, vp1-1ap,vr2-1-1-1-1ap, r1-1-1-1-1-1; metatarsus vp1-1-1ap,vr1-1-1ap, v1-1,vr1-1-1-1-1ap. Leg IV femur d1-1; metatarsusvr1-2ap. Retrolateral distal preening comb with four long macrosetae, equal length (Figure 62e). STCs with around 11 teeth, ITC short, robust, with a short tooth.

*Pedipalp:* Tibia 0.8 long, 0.6 wide; cymbium 0.8 long, 0.5 wide. Tibia bulbous proximally and curved ventrally, cymbium squared. Bulb globular, embolus originating retrolaterally, mid portion only slightly curved, broad until around three quarters of the embolus length,

from which it tapers apically to the tip; tip hooked at the apex and strongly sinuous (Figure 63a–c).

**Distribution.** This species is only known from Illintjitja, in the Pitjantjatjara Lands, in the arid NW of South Australia, where it is apparently sympatric with *A. valida* sp. nov. (Figure 64).

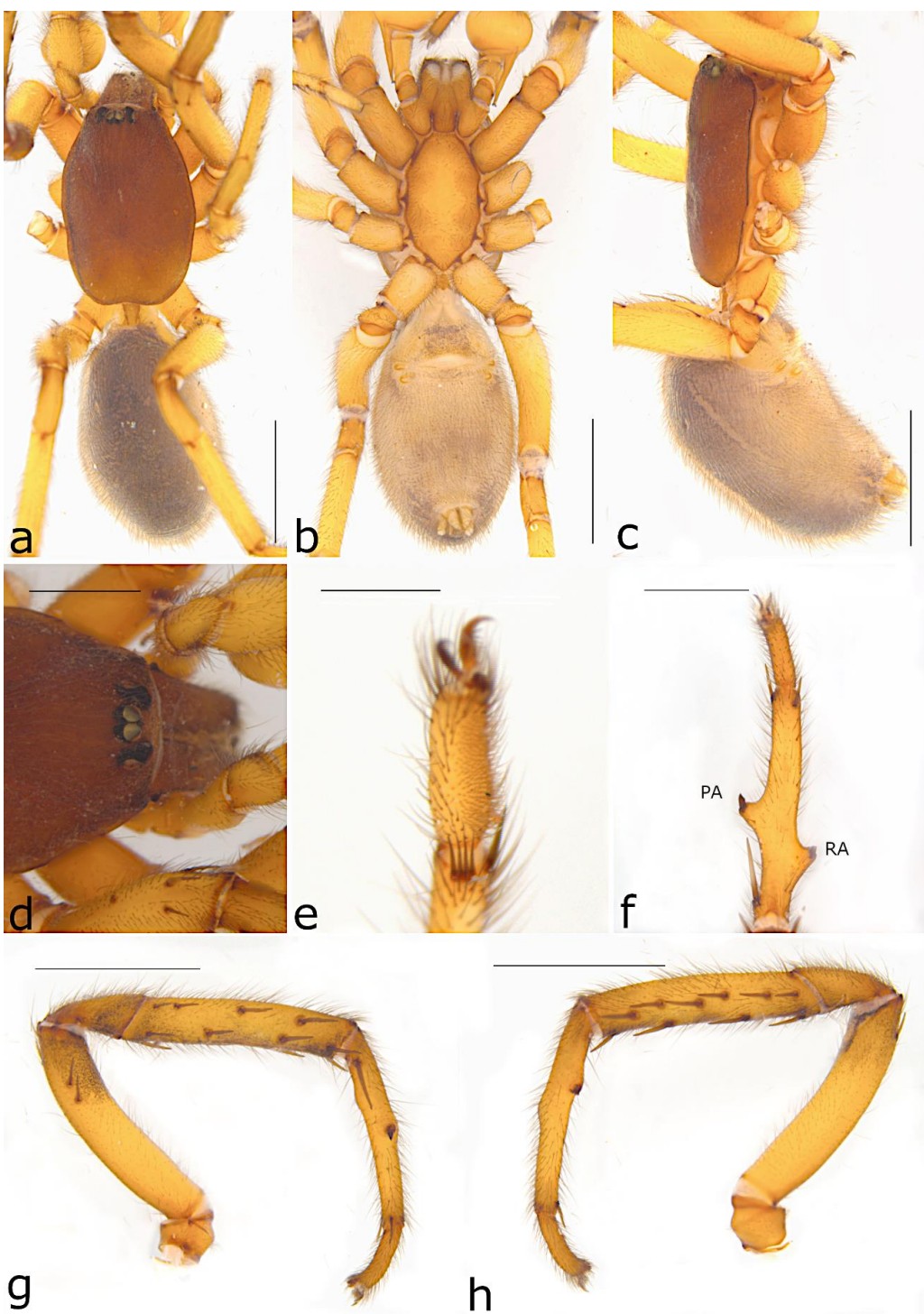

**Figure 62.** *Ariadna una* sp. nov. ♂holotype (SAMA NN10578) from near Illintjitja (SA): (**a**) habitus, dorsal view; (**b**) same, ventral view; (**c**) same, lateral view; (**d**) eyes, dorsal view; (**e**) left metatarsus IV, preening comb, retrolateral view; (**f**) left leg I, ventral view; (**g**) same, prolateral view; (**h**) same, retrolateral view. Scale bars (**a**–**c**) = 2 mm, (**d**) = 1 mm, (**e**) = 0.5 mm, (**f**) = 1 mm and (**g**,**h**) = 2 mm.

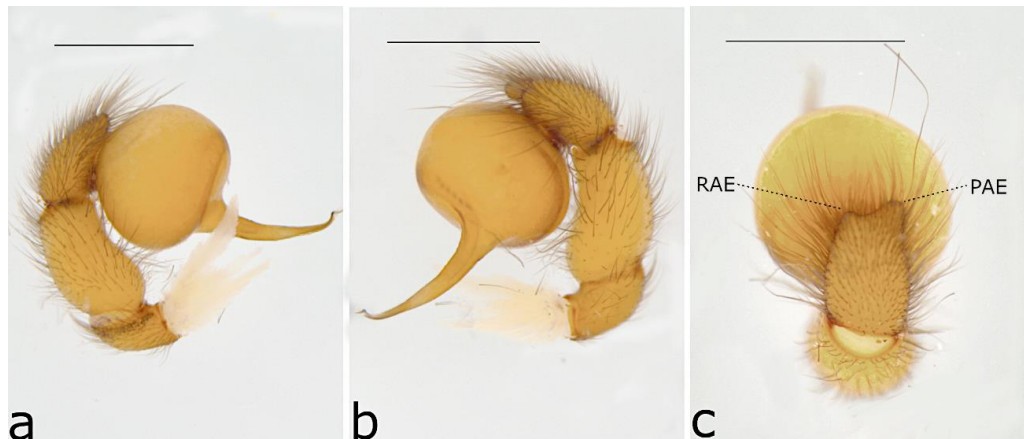

**Figure 63.** *Ariadna una* sp. nov. ♂holotype (SAMA NN10578) from near Illintjitja (SA): (**a**) left pedipalp, prolateral view; (**b**) same, retrolateral view; (**c**) same, cymbium. Scale bar = 1 mm.

**Legend**

● *Ariadna una* sp. nov.  ▲ *Ariadna valida* sp. nov.
■ *Ariadna ungua* sp. nov.  ◆ *Ariadna woinarskii* sp. nov.

**Figure 64.** Map showing the currently known distribution of *A. una* sp. nov., *A. ungua* sp. nov., *A. valida* sp. nov. and *A. woinarskii* sp. nov.

### *Ariadna ungua* sp. nov.

Figure 64, Figure 65a–h, Figure 66a–c, Figures 67a–h and 68a,b

urn:lsid:zoobank.org:act:57DA6D52-94B0-4D0E-B3D3-3EE99EFE65E7

**Type material.** *Holotype* ♂ AUSTRALIA: *South Australia*, Lake Dam (near Lake Frome), 30.91° S, 139.82° E, pitfall trap, 9 May 1989, coll. D. Hirst (SAMA NN30481).

*Paratypes:* 1♀Same data as holotype (SAMA NN31203); 1♀same data as holotype (SAMA NN30473).

**Etymology.** The specific epithet is the Latin noun for claw and refers to the elongate STC of legs I, II and IV of this species.

**Diagnosis.** Of the species with two well-defined apophyses situated in the apical half of metatarsus I, *A. ungua* sp. nov. can be differentiated from *A. pollex* by the shape of the pedipalp tibia, which is slender, 1.5 the length times the width in *A. ungua*, but is bulbous and stout in *A. pollex*, being 1.3 the length times the width (Figure 66a,b vs. Figure 45a,b). Females of *A. ungua* sp. nov. can be differentiated from other similar described South Australian species by a combination of a grey abdomen, lacking dorsal abdominal transverse striations, the absence of a cluster of short ventroretrolateral macrosetae on metatarsus I (Figure 67h vs. Figure 48h) and by the ventral lobe of the anterior receptaculum, which, in ventral view, is elongate, straight and bulbous apically (Figure 68a,b).

**Description.** ♂(based on holotype; SAMA NN30481). Total length 6.7.

*Cephalothorax:* 3.3 long, 2.2 wide, 1.8 high. Carapace golden orange-brown, darker red-brown anteriorly and darker on lateral edges, with scattered setae. With faint, mottled medial longitudinal darker striations, extending out from fovea and around eye mound. Sternum pale orange-cream, with darker orange at lateral edges and faintly in pre-coxal triangles and intercoxal extensions. Maxillae orange-cream, white apically, labium orange-brown, chelicerae red-brown (Figure 65a–h). Carapace elongate, rounded rectangular, and narrowing anteriorly, fovea a shallow indented pit (Figure 65a). Labium about four fifths length of maxillae. Sternum elongated oval with pre-coxal triangles and with smaller, rounded intercoxal extensions, with scattered, long, dark setae, row of three elongate setae along lateral edges of sternum (Figure 65b). In lateral view, carapace highest behind posterior eye row (Figure 65c). Eye group 0.8 wide, occupying 0.6 of the width of the carapace anteriorly; posterior eye row slightly recurved (Figure 65d).

*Abdomen:* 3.4 long. Dorsally grey-cream, uniform in colour; ventrally uniform pale grey-cream. With scattered red-brown setae (Figure 65a–c).

*Legs:* Leg length ratio IV > I > II > III. Leg I 9.5: femur 2.8, patella 1.1, tibia 2.4, metatarsus 2.3, tarsus 0.9. Leg II 9.4: 3.0, 1.2, 2.5, 1.8, 0.9. Leg III 7.3: 2.4, 1.0, 1.8, 1.2, 0.9. Leg IV 9.6: 3.0, 1.0, 2.6, 1.9, 1.1. Legs yellow-cream, darker towards apex. Femur I bowed in dorsal view. Metatarsus I sinuous, with two opposing apophyses; measuring proximally to distally, prolateral apophysis arising at 0.69 length of metatarsus, and retrolateral apophysis arising at 0.55 of length. Prolateral apophysis distinct bearing short, stout macroseta; retrolateral apophysis distinct, bearing long macroseta (Figure 65f–h). Tarsus I moderately bowed ventrally in lateral view, broader at apex than at base; tarsi II–IV ventrally incrassate. Tarsi II, III with sparse scopulate setae apically, tarsi IV with denser scopulate macrosetae along length; tarsi I with no scopulate setae. Macrosetae: Leg I Femur dp2ap, d1-1; tibia p1-1-1-1, vp1-1-1-1ap, vr2-1-1-1ap, r1-1-1; metatarsus apophysis (blunt)- p1ap, r-apophysis, r1ap. Leg II femur dp1ap, d1-1-2; tibia p1-1, vp1ap, vr1-1-1-1-1ap, r1-1-1-1-1; metatarsus vp1-1-1ap, vr1-1-1ap, r1-1-1. Leg IV femur d1; metatarsus vr2ap. Retrolateral distal preening comb with four macrosetae, equal length (Figure 65e). STCs with around 16 teeth, ITC with elongate, thin tooth; claw of legs I, II and IV elongate.

*Pedipalp:* Tibia 0.8 long, 0.5 wide; cymbium 0.7 long, 0.5 wide. Tibia rectangular and curved ventrally, cymbium squared, with a small but well-defined PAE and a small, rounded and ill-defined RAE. Bulb large globular, embolus originating retrolaterally, broad until around three-quarters of its length, where there is a brief prolateral expansion, from which it tapers apically to the tip, the tip sinuous and strongly hooked at the apex (Figure 66a–c).

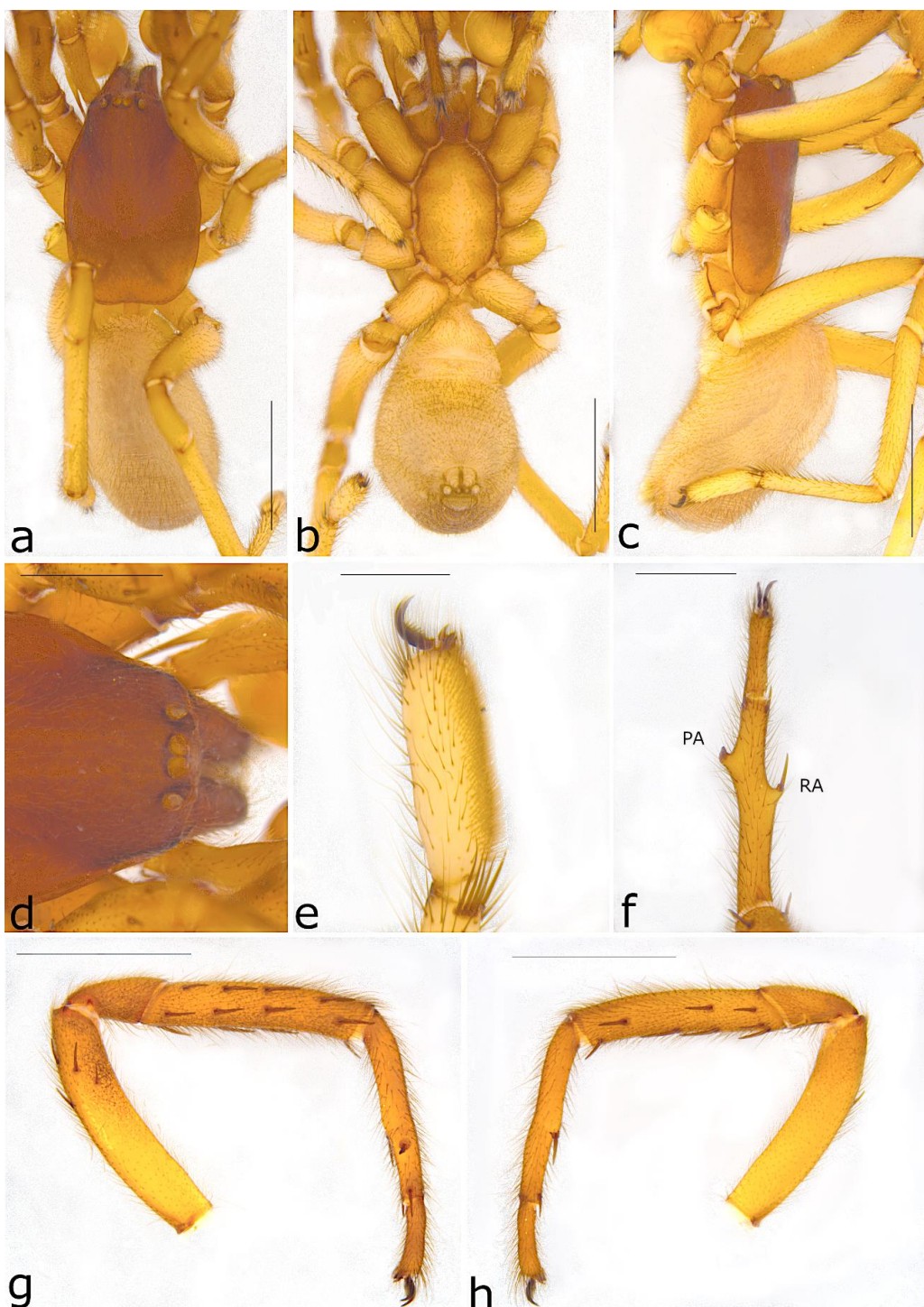

**Figure 65.** *Ariadna ungua* sp. nov. ♂ holotype (SAMA NN30481) from Lake Dam, near Lake Frome/Munda (SA): (**a**) habitus, dorsal view; (**b**) same, ventral view; (**c**) same, lateral view; (**d**) eyes, dorsal view; (**e**) left metatarsus IV, preening comb, retrolateral view; (**f**) left leg I, ventral view; (**g**) same, prolateral view; (**h**) same, retrolateral view. Scale bars (**a**–**c**) = 2 mm, (**d**) = 1 mm, (**e**) = 0.5 mm, (**f**) = 1 mm and (**g**,**h**) = 2 mm.

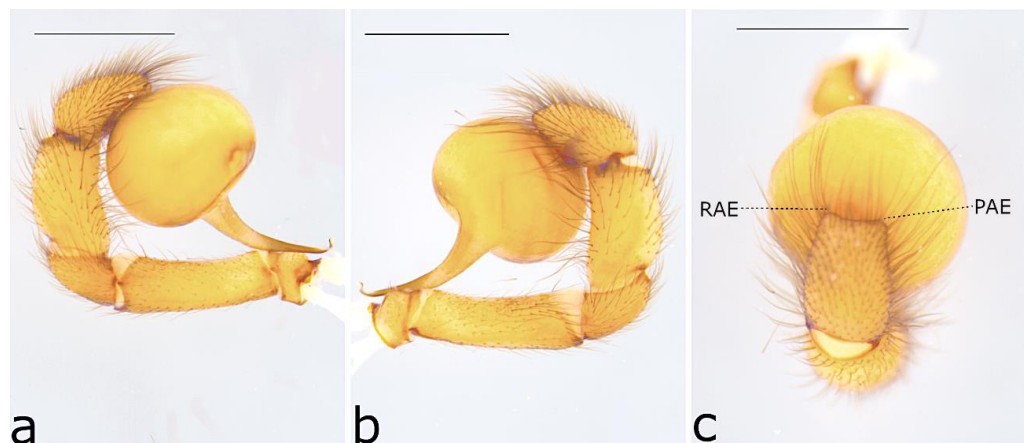

**Figure 66.** *Ariadna ungua* sp. nov. ♂holotype (SAMA NN30481) from Lake Dam, near Lake Frome/Munda (SA): (**a**) left pedipalp, prolateral view; (**b**) same, retrolateral view; (**c**) same, cymbium. Scale bar = 1 mm.

**Description.** ♀(based on paratype; SAMA NN31203). Total length 9.3.

*Cephalothorax*: 3.5 long, 2.2 wide, 2.1 high. Carapace golden orange-brown posteriorly, darker red-brown anteriorly, darker line demarcating the cephalic area. Sternum uniform yellow-brown, darker in colour at lateral edges and in pre-coxal triangles and intercoxal extensions. Maxillae orange-brown, white apically, labium dark orange-brown, chelicerae dark red-brown (Figure 67a–h). Carapace elongate, rectangular, narrowing anteriorly forming a neck; fovea not evident (Figure 67a). Labium about three quarters length of maxillae. Sternum elongated oval with pre-coxal triangles and with broad, rounded intercoxal extensions and with scattered, long, dark setae (Figure 67b). Carapace gently domed in lateral view, highest midway between fovea and eye group (Figure 67c). Eye group 0.8 wide, occupying 0.6 of the width of the carapace anteriorly; posterior eye row slightly recurved (Figure 67d).

*Abdomen*: 5.8 long. Dorsally violet-grey in a broad median strip, pale cream laterally and ventrally uniform cream. Elongate oval and with scattered red-brown setae (Figure 67a–c).

*Legs:* Leg length ratio I > II = IV > III. Leg I total 7.8: femur 2.5, patella 1.1, tibia 2.0, metatarsus 1.5, tarsus 0.7. Leg II 7.4: 2.4, 1.1, 1.9, 1.4, 0.6. Leg III 5.8: 2.0, 0.9, 1.2, 1.0, 0.7. Leg IV 7.4: 2.5, 1.1, 1.8, 1.3, 0.7. Legs I, II darker yellow-brown, with extensive dark mottled patches. Leg III light yellow-brown, with less extensive mottled colouration and leg IV more or less uniform pale yellow-brown. Femur I bowed in dorsal view. Figure 67f–h. Macrosetae: Leg I Femur dp2ap; tibia dp1, vp1-1-1-1, vr1-1-1-1ap; metatarsus v2-2-2-2-2-1-2. Leg II femur dp1ap; tibia p1-1-1ap, vr1-1-1-1ap; metatarsus vp2-2-2-2-2-2-2ap. Leg IV retrolateral distal preening comb with 4 elongate macrosetae, equal length (Figure 67e). STC I, II, with around 9 teeth, ITC with small tooth. STC and ITC of legs I, II, IV elongate.

*Pedipalp*: With numerous thick macrosetae prolaterally.

*Genitalia*: (Based on paratype; SAMA NN30473) Anterior receptaculum straight in ventral view, laterally bilobed, with the lateral lobe smaller than the dorsal lobe (Figure 68a,b).

**Variation.** Carapace lengths of females examined (*n* = 2) were 3.3 and 3.5; colouration was consistent for both specimens. For variations in leg macrosetae, see Table S3.

**Distribution.** This species is only known from near Lake Frome/Munda, located to the east of the Northern Flinders Ranges region in South Australia (Figure 64).

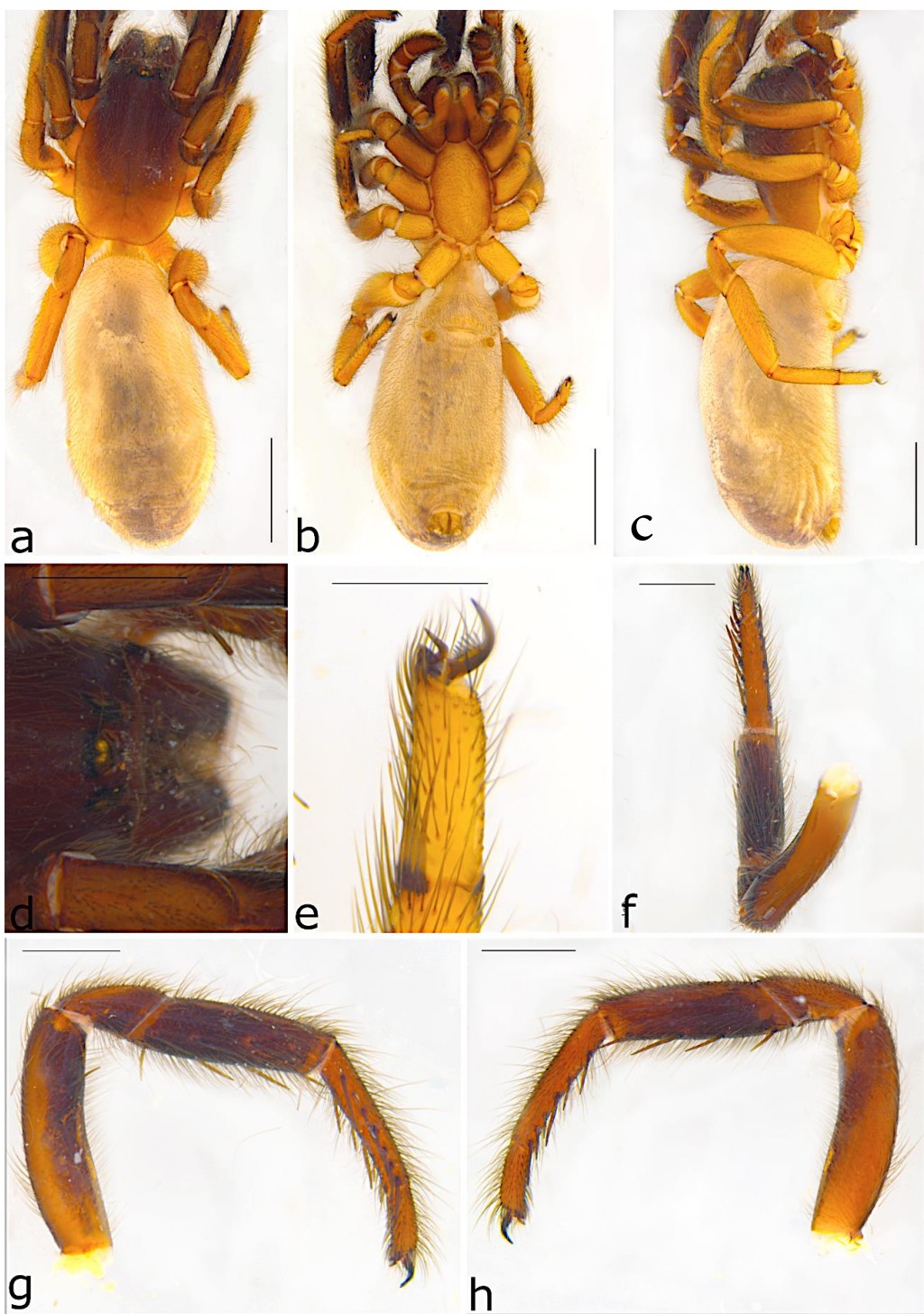

**Figure 67.** *Ariadna ungua* **sp. nov.** ♀paratype (SAMA NN31203) from Lake Dam, near Lake Frome/Munda (SA): (**a**) habitus, dorsal view; (**b**) same, ventral view; (**c**) same, lateral view; (**d**) eyes, dorsal view; (**e**) left metatarsus IV, preening comb, retrolateral view; (**f**) left leg I, ventral view; (**g**) same, prolateral view; (**h**) same, retrolateral view. Scale bars (**a**–**c**) = 2 mm, (**d**) = 1 mm, (**e**) = 0.5 mm and (**f**–**h**) = 1 mm.

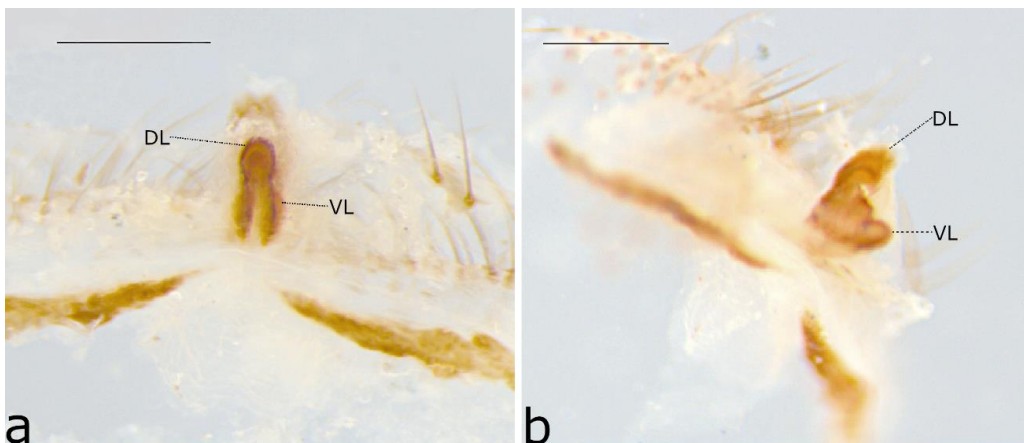

**Figure 68.** *Ariadna ungua* sp. nov. ♀paratype (SAMA NN30473) from Lake Dam, near Lake Frome/Munda (SA): anterior receptaculum (**a**) ventral view; (**b**) same, lateral view. Scale bar = 0.2 mm. DL = dorsal lobe, VL = ventral lobe.

### *Ariadna valida* sp. nov.

Figure 64, Figure 69a–h, Figure 70a–c, Figures 71a–h and 72a,b

urn:lsid:zoobank.org:act:041C894D-C181-4D7D-9359-F2A01B45C2AE

Type material. *Holotype* ♂AUSTRALIA: *South Australia:* 10 km S Illintjitja, 26.26° S, 130.39° E, pitfall trap, 4–8 May 1993, Pitjantjatjara Lands survey (SAMA NN10579).

*Paratype* ♀AUSTRALIA: *South Australia:* 25 km E Vokes Hill Corner, Mamungari Conservation Park, park boundary, 28.57° S, 130.69° E, 14 April 1994, coll. D. Hirst (SAMA NN30489).

**Other material examined.** 1♀AUSTRALIA: *South Australia:* Serpentine Lakes, Mamungari Conservation Park, 28.55° S, 129.04° E, 17 April 1994, coll. D. Hirst (SAMA NN30461).

**Etymology.** The specific epithet valida (meaning 'powerful') is a Latin singular feminine adjective in the nominative case and refers to the large size and robust build of this species.

**Diagnosis.** Of the species with two prominent apophyses on metatarsus I and with the prolateral apophysis located in the apical third of metatarsus I, *A. valida* sp. nov. can be differentiated by the retrolateral apophysis being distinctly raised but smaller than the prolateral apophysis, contrasting that of *A. flavescens* sp. nov., *A. arenacea* sp. nov. and *A. inflata* sp. nov., in which the retrolateral apophysis is indistinct and only slightly raised from the metatarsus (Figure 69f, vs. Figures 26f, 37f and 40f); and from *A. pollex* sp. nov. and *A. ungua* sp. nov., in which both apophyses are subequal in size and strongly raised (Figure 69 vs. Figures 44f and 65f). *Ariadna valida* sp. nov. can be differentiated from *A. diucrura* sp. nov. by the shape of the cymbium, which has a distinct and angular PAE and RAE, separated by a distinct v-shaped notch in *A. valida* sp. nov. but has an apically rounded PAE and indistinct RAE in *A. diucrura* sp. nov. (Figure 70c vs. Figure 36c). *Ariadna valida* sp. nov. is clearly separated from *A. una* sp. nov., which it is apparently occurs in sympatry, by the relative position of the retrolateral apophysis on metatarsus I, which is located in the apical third of the metatarsus in *A. una* sp. nov., whilst the prolateral apophysis is located at 0.4 of the metatarsus length, whilst, in *A. valida* sp. nov., the retrolateral apophysis is located at 0.5 of the metatarsus length and the prolateral apophysis at 0.6 (Figure 69f vs. Figure 62f) and by molecular data, with a divergence in the 12S gene of 12.7%.

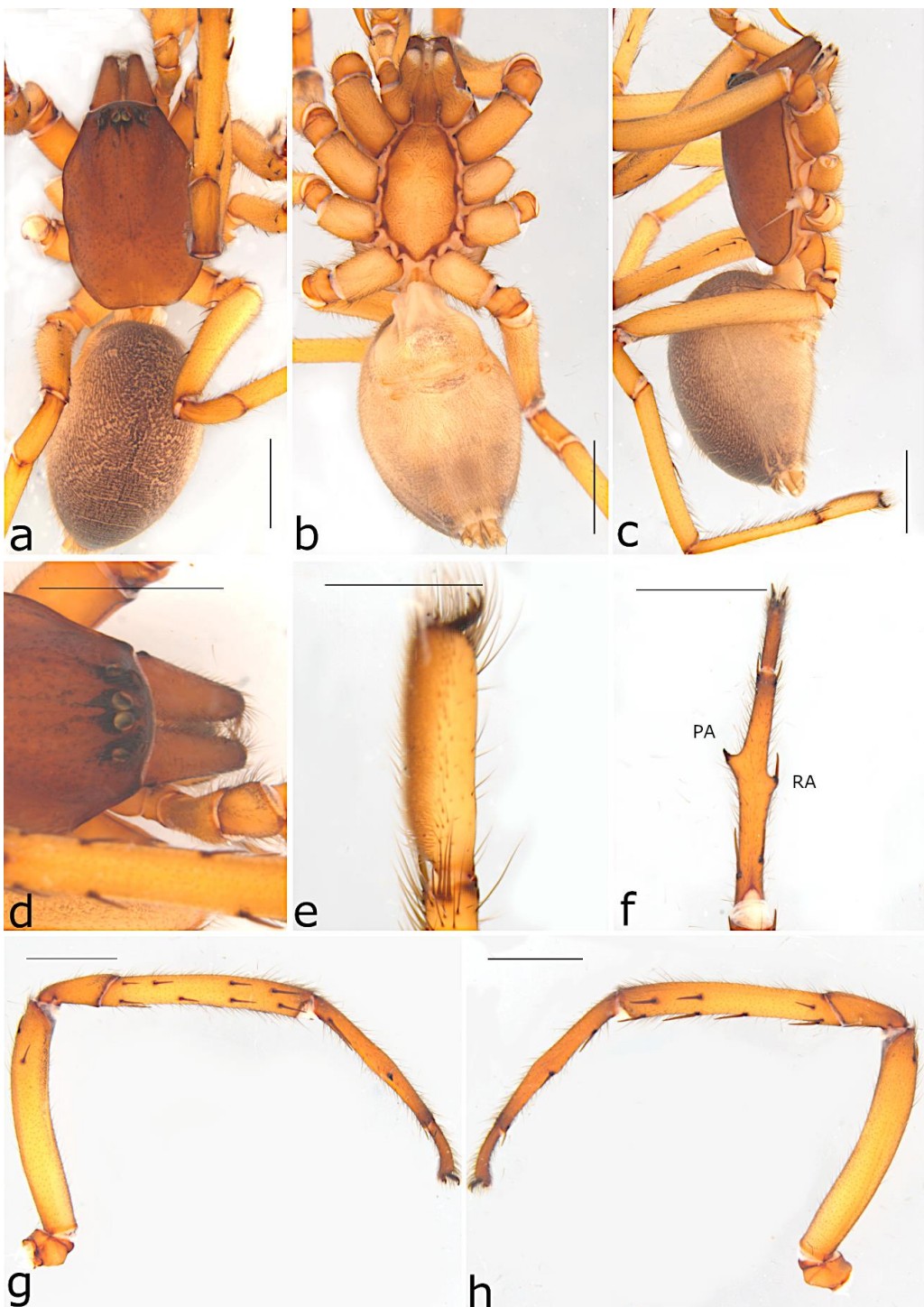

**Figure 69.** *Ariadna valida* sp. nov. ♂ holotype (SAMA NN10579) from near Illintjitja (SA): (**a**) habitus, dorsal view; (**b**) same, ventral view; (**c**) same, lateral view; (**d**) eyes, dorsal view; (**e**) left metatarsus IV, preening comb, retrolateral view; (**f**) left leg I, ventral view; (**g**) same, prolateral view; (**h**) same, retrolateral view. Scale bars (**a**–**d**) = 2 mm, (**e**) = 1 mm and (**f**–**h**) = 2 mm.

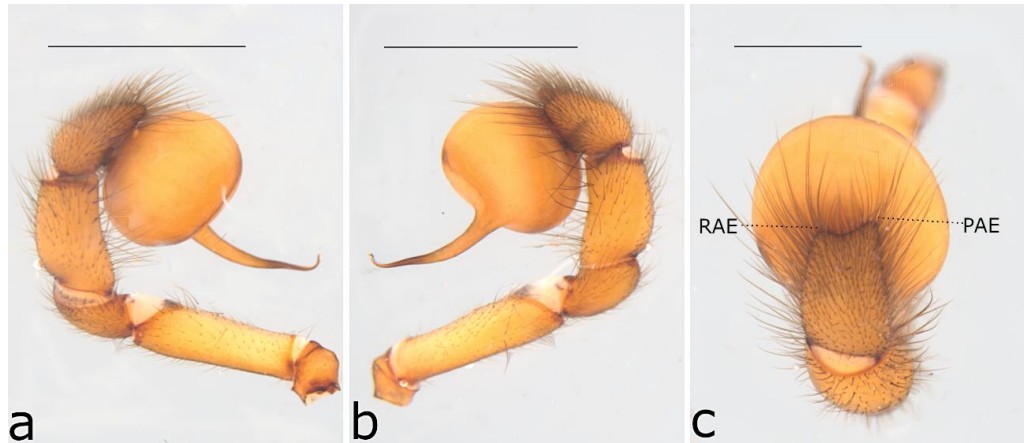

**Figure 70.** *Ariadna valida* sp. nov. ♂holotype (SAMA NN10579) from near Illintjitja (SA): (**a**) right pedipalp (mirrored), prolateral view; (**b**) same, retrolateral view; (**c**) same, cymbium. Scale bar = (**a**,**b**) = 2 mm and (**c**) = 1 mm.

**Description.** ♂(based on holotype; SAMA NN10579). Total length 9.8.

*Cephalothorax:* 4.6 long, 3.0 wide, 1.6 high. Carapace red-brown, with darker mottled colouring and dark setae arranged in longitudinal rows from fovea to eye group. Sternum golden red-brown, dark red-brown in pre-coxal triangles. Maxillae golden red-brown, white apically, labium darker red-brown, lighter apically. Chelicerae red-brown (Figure 69a–c). Carapace elongate oval, narrowing anteriorly; lateral edges strongly undulating; fovea a shallow indented pit (Figure 69a). Labium elongate about four fifths length of maxillae. Sternum elongate oval, with scattered setae and with pre-coxal triangles and faint intercoxal extensions (Figure 69b). Carapace gently raised towards the anterior in lateral view, highest just prior to the eye group. Chelicerae semi-porrect (Figure 69c). Eye group 1.1 wide, occupying 0.6 of the width of the carapace anteriorly; posterior eye row slightly recurved (Figure 69d).

*Abdomen*: 5.2 long. Dorsally slate-grey, with fine, pale cream transverse striations, ventrally pale cream-grey; with sparse brown setae (Figure 69a–c).

*Legs:* Leg length ratio I = II > IV > III. Leg I total 16.3: femur 5.2, patella 1.7, tibia 4.4, metatarsus 3.7, tarsus 1.3. Leg II 16.3: 5.3, 1.8, 4.1, 3.8, 1.3. Leg III 12.6: 4.1, 1.4, 2.9, 2.9, 1.3. Leg IV 15.4: 5.0, 1.6, 4.0, 3.3, 1.5. Legs golden orange-brown. Leg I metatarsus and tarsus darker orange-brown. Femur I bowed in dorsal view. Metatarsus I straight, with a prolateral and a retrolateral apophysis, prolateral apophysis at 0.61 of metatarsus length and retrolateral apophysis at 0.55 (measured proximally to distally). Both apophyses distinct, prolateral bearing a short, stout macroseta, retrolateral apophysis an elongate macroseta (Figure 69f–h). Tarsus I bowed ventrally in lateral view and was broader at the apex than at the base; tarsi II–IV were straight. Tarsi II, III, IV with sparse scopulate setae apically, tarsi IV inflated ventrally. Coxae I elongate. Macrosetae: Leg I Femur dp1, d1, dr1; tibia dp1-1-1-1, vp1-1-1-1ap, vr2-1-1-1ap, r1-1-1; metatarsus p1(long)-apophysis-1ap, r1-apophysis-1ap. Leg II femur dp1ap, d1-1, dr1ap; tibia dp1-1-1, p1-1ap, vr1-1-1ap, r1-1; metatarsus p1-1, vp1-1-1ap, vr1-1-1ap. Leg IV femur dp1, d1-1-1-1; tibia v1; metatarsus v2ap. Retrolateral distal preening comb with four long macrosetae, three longer, one shorter (Figure 69e). STCs with around 14 teeth, ITC with an elongated, stout well-defined tooth.

*Pedipalp:* Tibia 1.1 long, 0.7 wide; cymbium 1.1 long, 0.7 wide. Tibia short, broad, gently curved ventrally and expanded dorsally. Cymbium with enlarged triangular PAE, smaller RAE. Bulb large, oval, globular, embolus originating retrolaterally, mid portion broad until around three quarters of the length of the embolus, from where it tapers apically to tip; tip strongly hooked at apex (Figure 70a–c).

**Description.** ♀(based on paratype; SAMA NN30489). Total length 11.0.

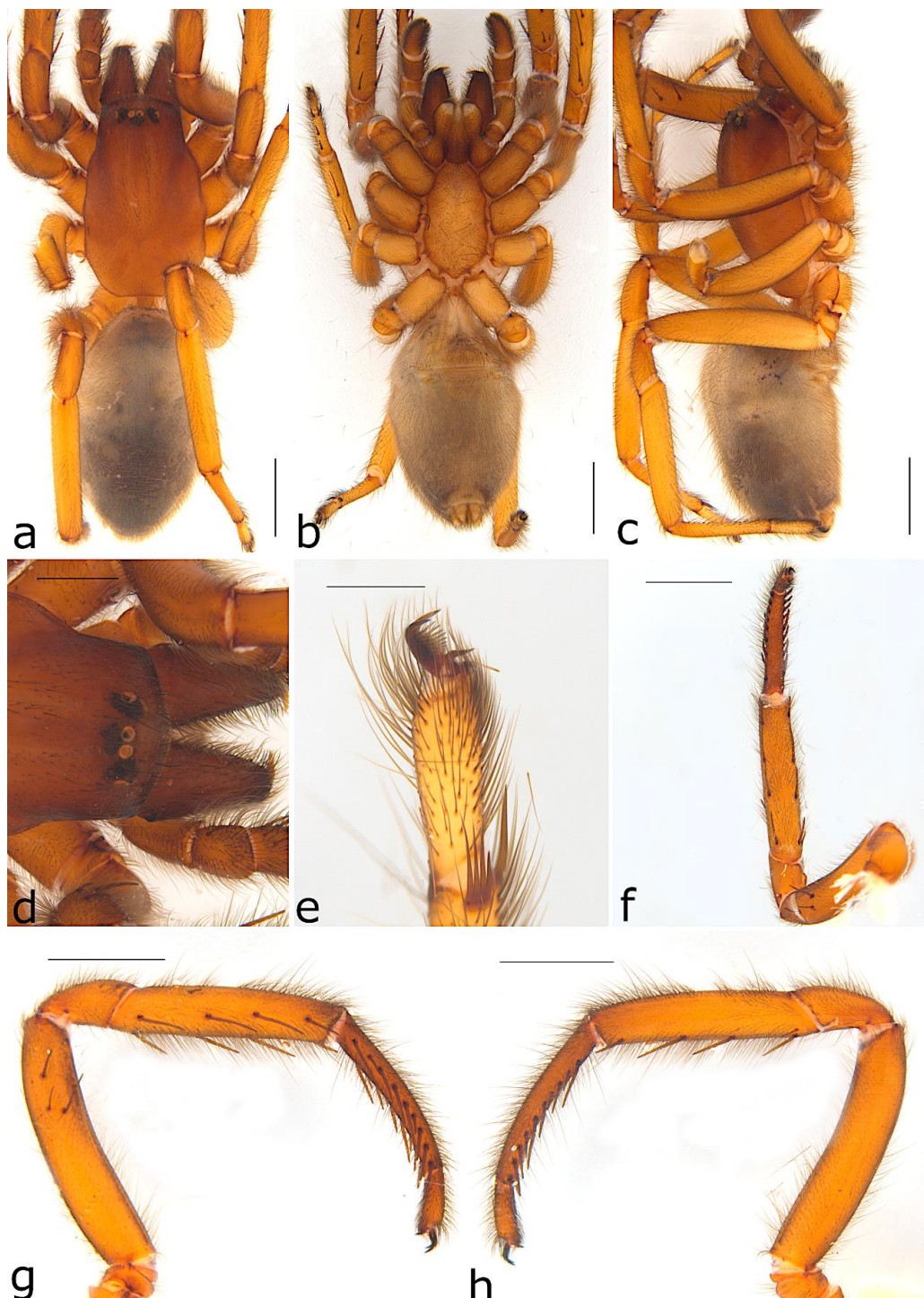

**Figure 71.** *Ariadna valida* sp. nov. ♀paratype (SAMA NN30489) from Mamungari (SA): (**a**) habitus, dorsal view; (**b**) same, ventral view; (**c**) same, lateral view; (**d**) eyes, dorsal view; (**e**) left metatarsus IV, preening comb, retrolateral view; (**f**) left leg I, ventral view; (**g**) same, prolateral view; (**h**) same, retrolateral view. Scale bars (**a**–**d**,**f**–**h**) = 1 mm and (**e**) = 0.5 mm.

*Cephalothorax:* 5.1 long, 3.1 wide, 3.1 high. Carapace golden orange-brown. Sternum pale cream-brown, with fine, faint dark medial strip anteriorly, which continues on to labium. Maxillae and labium orange-brown, paler apically; chelicerae darker red-brown. Figure 71a–c. Carapace elongated oblong, with undulating lateral edges and forming a broad, squared neck anteriorly. Fovea an indistinct shallow indented pit (Figure 71a). Labium around three quarters the length of the maxillae. Sternum elongate oval, forming

and broad neck and convex apically, rounded posteriorly with sparse, scattered dark setae and distinct pre-coxal triangles (Figure 71b). Carapace domed in lateral view, highest midway between fovea and eye group (Figure 71c). Eye group 1.1 wide, occupying 0.6 of the width of the carapace anteriorly; posterior eye row slightly recurved (Figure 71d).

*Abdomen:* 5.9 long. Dorsally uniform violet-grey; ventrally cream-grey; covering of fine dark brown setae (Figure 71a–c).

*Legs:* Leg length ratio IV > I = II > III. Leg I total 12.4: femur 4.1, patella 1.5, tibia 3.2, metatarsus 2.7, tarsus 0.9. Leg II 12.4: femur 3.9, patella 1.5, tibia 3.2, metatarsus 2.7, tarsus 1.1. Leg III 9.7: 3.3, 1.2, 2.2, 2.0, 1.0. Leg IV 12.5: 4.2, 1.4, 3.3, 2.4, 1.2. Legs pale orange-brown, legs I and II darker orange-brown. Femur I strongly bowed in dorsal view. Tarsi short, stout. Tarsi I, II broader at apex than at base. All leg segments covered in sparse, black setae. Figure 71f–h. Macrosetae: Leg I: femur dp3ap; tibia p1, vp1-1-1-1ap, vr2-1-1-1; metatarsus v2-2-2-2-2-2-2-2-2-1. Leg II: femur dp1ap, dr1ap; tibia p1, vp1-1-1-1ap; vr1-1-1-1ap; metatarsus v2-2-2-2-2-2-2-2-2-2-1. Leg IV: retrolateral distal preening comb with four macrosetae, three longer, one shorter (Figure 71e). STC I, II with nine teeth, ITC with small, stout tooth.

*Pedipalp:* Densely covered in dense black setae, with numerous macrosetae prolaterally; single macrosetae prolaterally on pedipalp patella.

*Genitalia:* The anterior receptaculum is bilobed, ventral lobe gently sinuous in ventral view. Dorsal lobe around the same length as ventral lobe (Figure 72a,b)

**Variation.** Carapace lengths of females examined (*n* = 2) were 5.1 and 5.3; colouration was consistent for both specimens. For variation in leg macrosetae see Table S3.

**Remarks.** Males and females of this species were matched using molecular evidence, with a mean intraspecific divergence in the 12S gene of 4.8%, locational data and morphological data, namely the number of macrosetae in the preening comb of leg IV, pattern of macrosetae on the tibia of leg I and the large size of males and females.

**Distribution.** Known from Illintjitja in the Pitjantjatjara Lands, where it is apparently sympatric with *A. una* sp. nov., and from Vokes Hill Corner, Mamungari Conservation Park, and Serpentine Lakes, Mamungari Conservation Park, in mid-west South Australia (Figure 64).

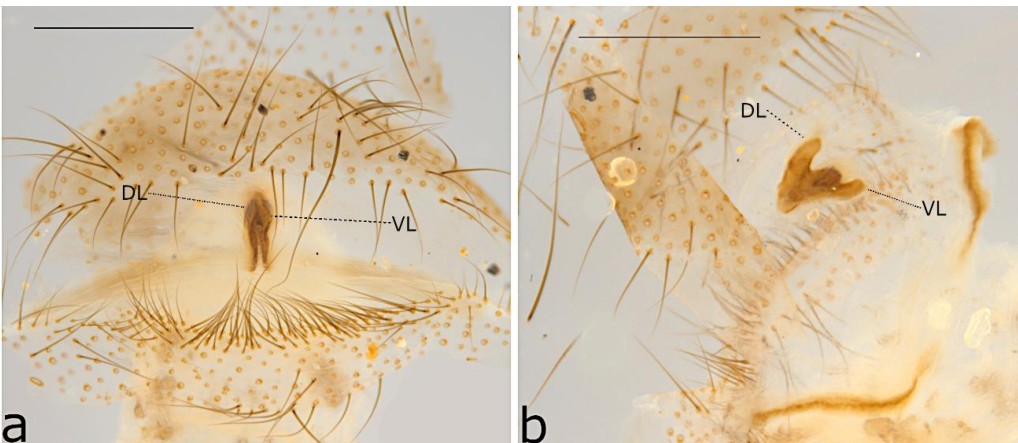

**Figure 72.** *Ariadna valida* sp. nov. ♀paratype (SAMA NN30489) from Mamungari (SA) (SA): anterior receptaculum (**a**) ventral view; (**b**) same, lateral view. Scale bar (**a**) = 0.5 mm and (**b**) = 0.2 mm. DL = dorsal lobe, VL = ventral lobe.

## *Ariadna woinarskii* **sp. nov.**

Figure 64, Figure 73a–h, Figure 74a–c, Figures 75a–h and 76a,b
urn:lsid:zoobank.org:act:B0A58C9B-DF35-43CA-A7BC-F9410B058256

**Type material.** *Holotype* ♂ AUSTRALIA: *South Australia*, Kangaroo Island, Lathami Conservation Park, 35.62736° S; 137.23563° E, in tube webs in cracks in rock wall; adjacent to creek, bottom of a gully, 7 July 2020, coll. J. Marsh (SAMA NN31200).

*Paratype* ♀same data as holotype (SAMA NN31201).

**Other material examined.** 1♂West Island, SW of Victor Harbour, 35.60797° S, 138.59189° E, April 1989, coll. S. Lewer (SAMA NN30570); 1♀, juv, West Island, 35.60797° S, 138.59189° E, under granite, 11 February 1989, coll. S. Lewer (SAMA NN30572). 1♀Kangaroo Island, Lathami Conservation Park, 35.62811° S, 137.23930° E, in tube web in crevice of vertical rock wall adjacent to small creek line, in steep gully, 7 July 2020, coll. J. Marsh (SAMA NN31202). 1♀Lincoln National Park, parking area, near Flinders Monument, 34.80° S, 135.94° E, under bark, 27 March 1987, coll. D.C. Lee and D. Hirst (SAMA NN30547). 5 1♀, 4 juvenile, Head of Bight, littoral zone, rocky faces and cliffs, 31.46° S, 131.12° E, 28 September 1988, coll. D. Hirst (SAMA NN30550).

**Etymology.** The type locality of this species was impacted by high severity fire in the 2019–2020 black summer wildfires on Kangaroo Island. This species is named in honour of Prof. John C.Z. Woinarski, in recognition of his invaluable role in promoting the conservation of fire-impacted invertebrates in Australia.

**Diagnosis.** *Ariadna woinarskii* sp. nov. can be clearly differentiated from sympatric species *A. tangara* by the lack of apophyses on metatarsus I in *A. woinarskii* (Figure 73f vs. Figure 57f), and from *A. clavata* by the lack of dorsal transverse abdominal striations in *A. woinarskii* sp. nov. (Figure 73a vs. Figure 4a). It can be differentiated from the morphologically closest species, *A. insula* sp. nov. by the macrosetae of metatarsus I, which has pro and retrolateral macrosetae in addition to ventral retrolateral and ventral prolateral *A. insula* sp. nov., whereas in *A. woinarskii* sp. nov. there are only ventral retrolateral and ventral prolateral macrosetae and by the sinuous metatarsus I of *A. woinarskii* sp. nov. (Figure 73f–h vs. Figure 42f–h). Females of *A. woinarskii* sp. nov. can be differentiated from other described *Ariadna* by a combination of the uniformly grey abdomen (Figure 75a) and the lobes of the anterior receptaculum, on ventral view which are elongate and sinuous, with a distinct ventral facing apical fold on the dorsal lobe (Figure 76a,b).

**Description.** ♂(based on holotype; SAMA NN31200). Total length 5.1.

*Cephalothorax:* 2.5 long, 1.6 wide, 1.1 high. Carapace mid-brown, darker on lateral edges, irregular mottled medial longitudinal darker striations, extending anterior to fovea and around eye mound, darker patches at base of setae; sternum orange-brown with dark grey mottled colouring, darker orange at lateral edges and in pre-coxal triangles and inter-coxal extensions, maxillae cream, paler in colour apically, labium olive-cream, chelicerae mid brown(Figure 73a–c). Carapace oval, narrowing anteriorly. With scattered sparse, long dark setae, forming longitudinal rows medially between fovea and posterior eye row. Surface of carapace with fine, scale-like markings; fovea a shallow indented pit (Figure 73a). Labium about four fifths length of maxillae; chelicerae with basal transverse ridge. Sternum elongated oval with pre-coxal triangles and with smaller, rounded intercoxal extensions, with scattered, long, black setae; ventrally inflated at joins with coxae (Figure 73b). Carapace flattened in lateral view, highest just behind posterior eye row (Figure 73c). Posterior eye row straight, eyes large, eye group occupying about half of carapace width (Figure 73d).

*Abdomen:* 2.6 long. Dorsally violet-grey, with faint light irregular striations, ventrally dark grey, with broad, irregular, longitudinal cream patch. With covering of fine brown setae (Figure 73a–c).

*Legs:* Leg length ratio I > II > IV > III. Leg I total 8.1: femur 2.4, patella 0.8, tibia 2.2, metatarsus 2.0, tarsus 0.7. Leg II 7.7: 2.3, 0.9, 1.9, 1.9, 0.7. Leg III 5.4: 1.7, 0.6, 1.2, 1.3, 0.6. Leg IV 6.5: 2.1, 0.8, 1.7, 1.3, 0.6. Legs cream, darker grey-brown towards apex. Femur I slightly bowed in dorsal view, metatarsus I gently sinuous when viewed ventrally (Figure 73f–h). Tarsi short, tarsi I curved ventrally; tarsi and metatarsi II, and tarsi III with distal ventral scopulate setae. Tarsi I slightly bowed ventrally in lateral view, broader at apex than at base; tarsi II–IV straight in lateral view. Macrosetae: Leg I: femur d2ap, d1-1-1ap, dr1ap; tibia p1-1-1-1, vp1-1-1ap, vr1-1-1-1-1ap, r1-1-1-1; metatarsus vp1-1-1ap, vr1-1-1ap, Leg II: femur d1-1/0, d2ap, dp2ap; tibia p1-1-1, vp1ap, vr1-1-1-1ap, r1-1-1; metatarsus p1-1-1ap, v1-1, r-1-1-1ap. Leg IV: femur d1-1-1. Retrolateral distal preening comb with 4 macrosetae (Figure 73e). STC I, II with 12 teeth, ITC with small tooth.

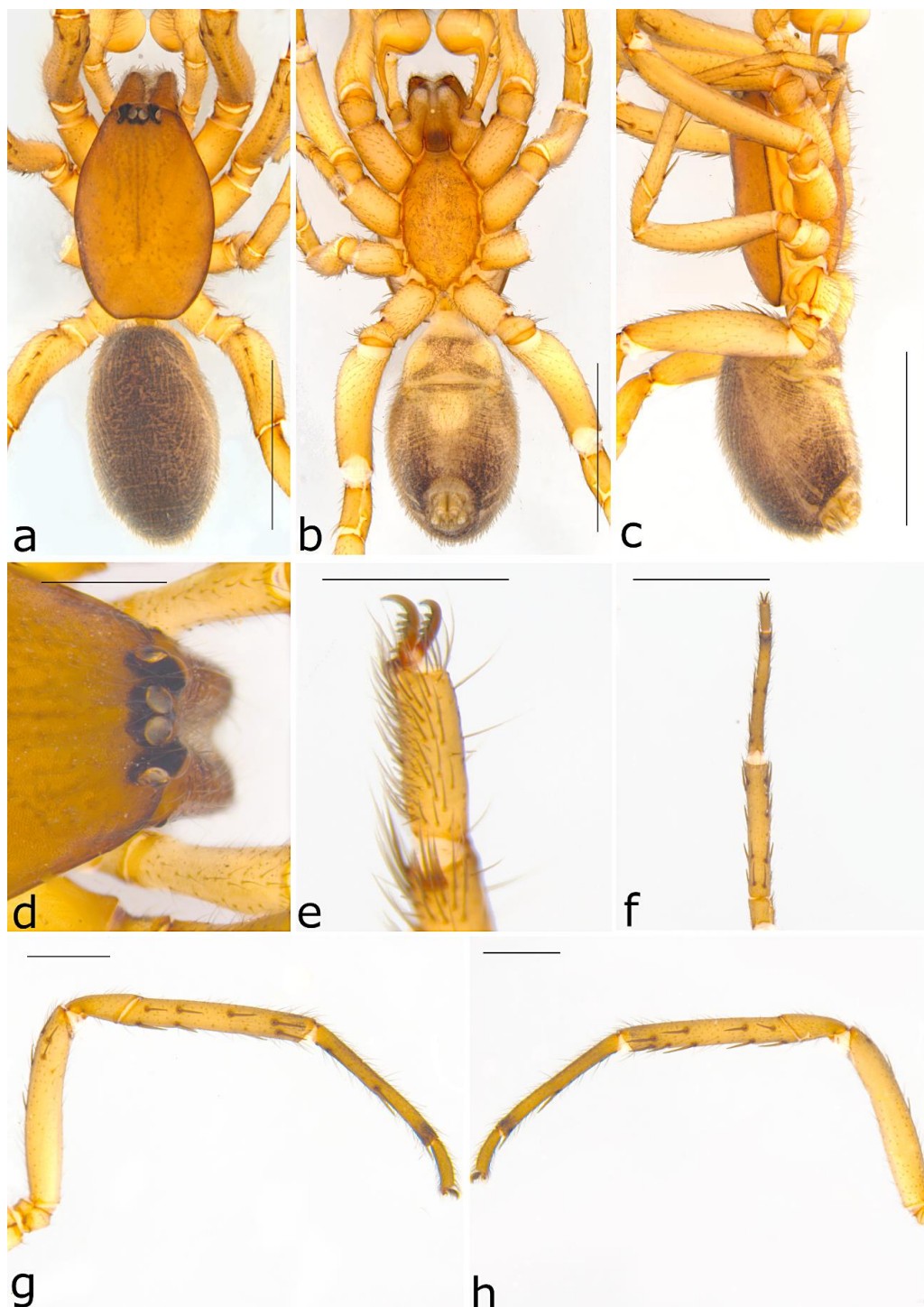

**Figure 73.** *Ariadna woinarskii* sp. nov. ♂holotype (SAMA NN31200) from Kangaroo Island (SA): (**a**) habitus, dorsal view; (**b**) same, ventral view; (**c**) same, lateral view; (**d**) eyes, dorsal view; (**e**) left metatarsus IV, preening comb, retrolateral view; (**f**) left leg I, ventral view; (**g**) same, prolateral view; (**h**) same, retrolateral view. Scale bars (**a**–**c**) = 2 mm, (**d**,**e**) = 0.5 mm and f–h = 1 mm.

*Pedipalp:* Tibia 0.5 long, 0.3 wide; cymbium 0.5 long, 0.3 wide. Pedipalp tibia rectangular, with ventral indentation. Cymbium rounded rectangular and the bulb globular, with apical PAE and RAE. The embolus originates retrolaterally from the bulb, the mid portion is broad until around three quarters of the length of the embolus, where there is a brief prolateral expansion, from which it tapers apically to the tip; embolus tip is sinuous (Figure 74a–c).

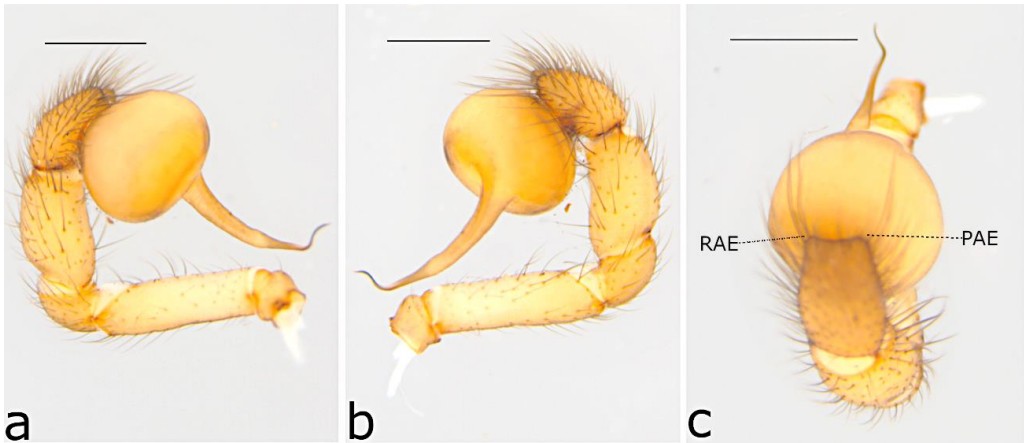

**Figure 74.** *Ariadna woinarskii* sp. nov. ♂holotype (SAMA NN31200) from Kangaroo Island (SA): (**a**) left pedipalp, prolateral view; (**b**) same, retrolateral view; (**c**) same, cymbium. Scale bar = 0.5 mm.

**Description.** ♀(based on paratype; SAMA NN31201). Total length 6.9.

*Cephalothorax*: 3.1 long, 1.9 wide, 1.7 high. Carapace orange-brown, darker anteriorly, sternum golden-brown, paler medially, with irregular slightly darker intercoxae patches, labium dark red-brown, paler apically, maxillae orange-brown, white apically. Chelicerae orange-brown(Figure 75a–c). Carapace oval, with broad, square 'neck' anteriorly, anterior edge convex, laterally edges gently undulating; fovea indistinct shallow pit (Figure 75a). Labium narrowed anteriorly, about $\frac{3}{4}$ length of maxillae. Chelicerae broad, robust, with dense, long, brown setae. Sternum elongated oval and covered with long, fine macrosetae, lateral edges with large, defined pre-coxal triangles, which are larger posteriorly, and with small, rounded intercoxal extensions (Figure 75b). From lateral view carapace raised to fovea and then levelled until it lowers just posterior to eyes (Figure 75c). Posterior eye row slightly recurved, eyes 0.7 wide and occupying 0.5 of carapace width (Figure 75d).

*Abdomen*: 5.2 long. Dorsally pale slate-grey, ventrally pale cream anteriorly with two darker grey broad median stripes posteriorly. With short, dense setae (Figure 75a–c).

*Legs:* Leg length ratio I > II > IV > III. Leg I total 7.4: femur 2.2, patella 0.9, tibia 2.0, metatarsus 1.7, tarsus 0.6. Leg II 6.8: 1.9, 1.0, 1.7, 1.5, 0.7. Leg III 5: 1.6, 0.7, 1.1, 1.1, 0.5. Leg IV 6.6: 2.0, 1.0, 1.7, 1.3, 0.6. Legs golden-brown, legs I, II darker golden-brown. Femur I bowed in dorsal view. Figure 75f–h. Macrosetae: Leg I femur: dp3ap, d1, dr1; tibia p1-1-1, vp1-1-1-1-1-1-1ap, vr1-1-1-1-1-1-1ap, r1-1; metatarsus v2-2-2-2-2-2-2. Leg II femur dp1-1ap, d1-1, dr1ap; tibia p1-1-1, vp1-1-1ap, vr1-1-1-1-1-1-1-1; metatarsus v2-2-2-2-2-2-2-2. Leg IV none. Legs with covering of setae, denser on leg I. Retrolateral distal preening comb with 4 macrosetae, 3 short, 1 long (Figure 75e). STC I, II with six teeth, ITC with a small, stout tooth.

*Pedipalp:* Densely covered in dense black setae, with numerous macrosetae prolaterally.

*Genitalia:* (Based on female SAMA NN31202) The anterior receptaculum is bilobed, both the ventral and dorsal lobes are elongate and sinuous, the ventral lobe being strongly sinuous when viewed ventrally; and with a distinct ventral facing fold apically (Figure 76a,b).

**Variation.** Males (*n* = 2), carapace length ranged from 2.5 to 2.7 (mean 2.6); females (*n* = 8), carapace length ranged from 3.1 to 4.5 (mean 3.9). Colour varied little between specimens.

**Remarks.** Males and females of this species were matched using molecular evidence, with a mean intraspecific divergence in the 12S gene of 2.7%, locational data and morphological data, namely the number of macrosetae in the preening comb of leg IV, and the structure of the anterior receptaculum of females.

**Distribution.** This species' distribution extends from Northern Kangaroo Island, through to southwestern South Australia (Figure 64).

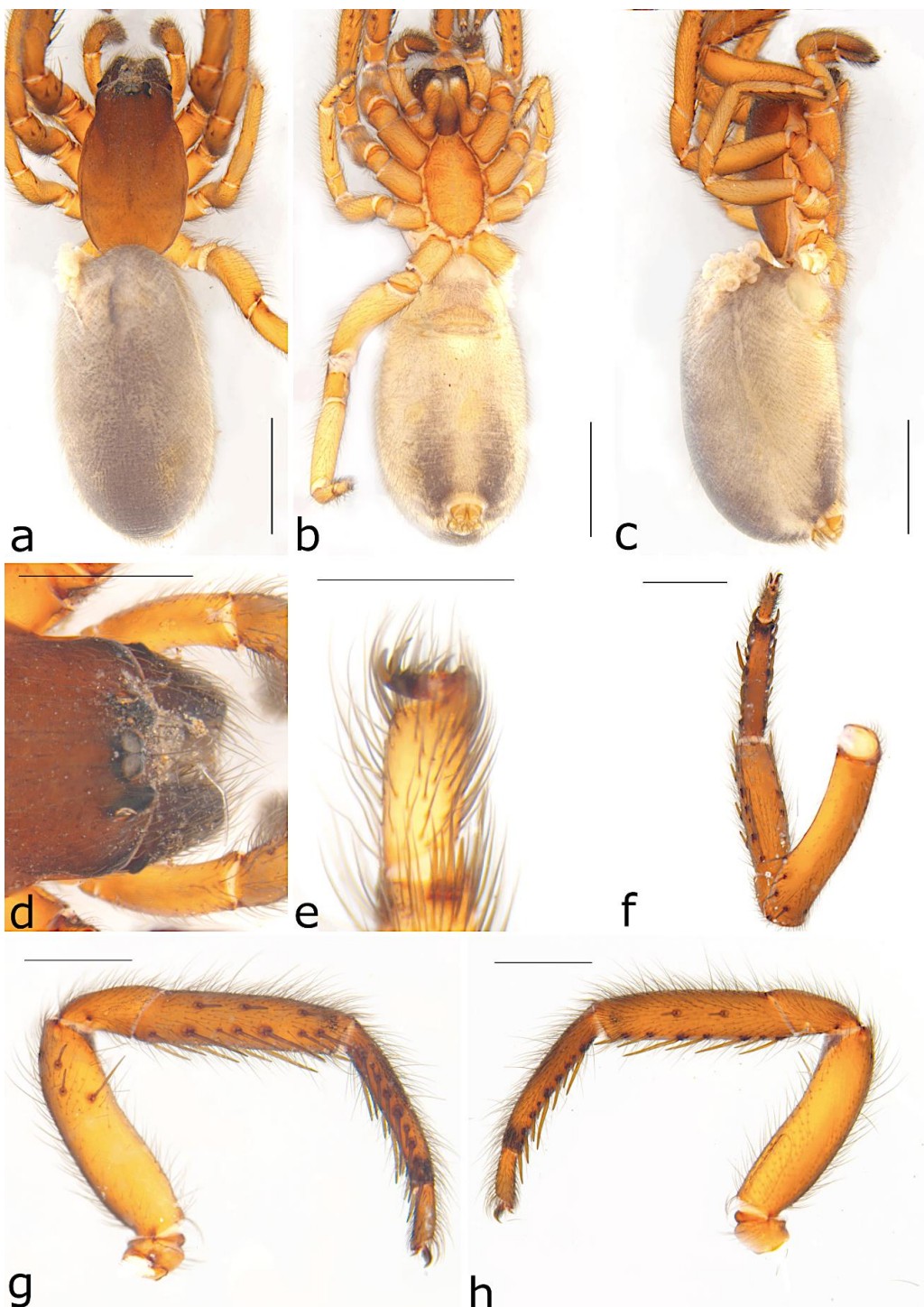

**Figure 75.** *Ariadna woinarskii* sp. nov. ♀paratype (SAMA NN31201) from Kangaroo Island (SA): (**a**) habitus, dorsal view; (**b**) same, ventral view; (**c**) same, lateral view; (**d**) eyes, dorsal view; (**e**) left metatarsus IV, preening comb, retrolateral view; (**f**) left leg I, ventral view; (**g**) same, prolateral view; (**h**) same, retrolateral view. Scale bars (**a**–**d**,**f**–**h**) = 1 mm and (**e**) = 0.5 mm.

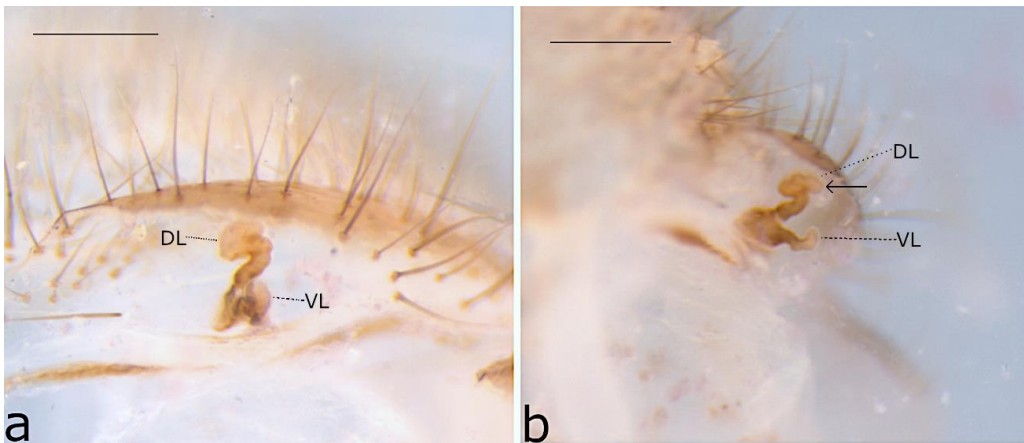

**Figure 76.** *Ariadna woinarskii* sp. nov. ♀(SAMA NN31203) from Kangaroo Island (SA): anterior receptaculum (**a**) ventral view; (**b**) same, lateral view, ventral facing apical fold on DL indicated by arrow. Scale bar (**a**) = 0.2 mm and (**b**) = 0.25 mm. DL = dorsal lobe, VL = ventral lobe.

## 5. Discussion

This review of South Australian and Victorian *Ariadna* species doubles the number of described *Ariadna* in Australia, revealing a remarkable diversity centred in the arid regions of South Australia. Southern Australia's arid zone is known for its highly diverse lineages of mygalomorph spiders. Phylogenetic and biogeographic studies on these groups have revealed complex patterns of diversification and derived xeric adaptions, occurring with the aridification of the continent (for example [33,34]).

Of the 23 species described in this study, 17 are known only from single collection localities. Limited collecting effort is likely to have influenced this apparent localized distribution pattern. However, the high turnover of species, particularly in northern South Australia, coupled with the reported limited dispersal abilities of Segestriidae, indicates that natural range restriction is present in the group, a conclusion supported by other studies on Segestriidae (for example [6,15,35]). Species with high levels of range restriction, i.e., short-range endemics, are likely to be of elevated conservation concern [36]. Many of the species considered in this study warrant further surveys to better determine levels of endemism and to assess their conservation status.

Five of the species treated in this study, *A. clavata*, *A. tangara*, *A. una* sp. nov., *A. valida* sp. nov. and *A. woinarskii* sp. nov., occur in sympatry with at least one other species, supporting findings from other studies of sympatry in Segestriidae in Australia and world-wide (for example [4,6]). This propensity for sympatry, in addition to the morphologically conservative morphology of females, means that, for most species, descriptions based only on female morphology and matching males to females based on morphology is problematic. In this study, we therefore adopted a conservative approach and did not describe species where only the female was known and where the molecular analyses did not distinguish from, or match the specimen to, nearby males. Typically, in *Ariadna*, mature males are only present for a restricted time-period, whereas mature females can be collected most of the year [4,6], thus resulting in widespread collecting biases. In both the South Australian Museum and Museum Victoria collections, there was a greater number and diversity of mature females compared to males. It is therefore likely that the species described in this review represent only a small portion of true segestriid diversity in these states and dedicated surveys for male Segestriidae, or large-scale molecular analysis of females, are needed to better determine the true diversity.

The use of spider genitalia in species delineation is well-supported [37,38]. For groups, such as Segestriidae, which belong to the Synspermiata clade [39] and have highly simplified genitalia, its usefulness in species delineation, especially when combined with somatic characters is documented (for example [3,4,9,15]), and confirmed by this study. Our analyses showed that, in males, important genitalic characters were the presence of,

and the shape of, PAE and RAE on the pedipalp cymbium; the shape of the embolus and the relative length and shape of the pedipalp tibia. In females, the structure, size and shape of the anterior receptaculum were important.

Analysis of inter- and intraspecies variations of somatic characters revealed abdominal colouration of males and females, and the presence of, relative size of and relative position of apophyses on metatarsus I of males to be reliable characters for delineation. A previous study demonstrated that the apophyses of metatarsus I of males in *A. tangara* are used to hold the female in a 'mating grasp' whilst copulating; the apophyses effectively locking on to the female's cephalothorax and preventing her movement [4]. The role of these apophyses in copulation suggests they are under sexual selection pressures, and the usefulness of the presence and structures of apophyses or mating spurs on leg I for species diagnosis, has been shown for a range of spider groups, including Segestriidae (for example [3,6,40]).

Examination of the macrosetae patterns showed frequent variation in macrosetae both between individuals of the same species and between opposing legs of the same individual in males and females. This was true for most species and for most leg segments, including the ventroprolateral and ventroretrolateral macrosetae of the tibia and metatarsus of females. This has consequences for taxonomy of Segestriidae in Australia given most of the historical species descriptions were based only on females, and the absolute number of macrosetae on the ventral surface of tibiae and metatarsi I was frequently used diagnostically (for example [41,42]). Whilst our findings indicate the absolute number of macrosetae is generally not a reliable character, we found that the overall pattern of macrosetae in most leg segments of leg I was, for example, the presence or absence of prolateral or retrolateral macrosetae on tibia I and the overall pattern of macrosetae on metatarsus I of males, even if the absolute number of macrosetae varied. This is in agreement with findings from other studies for example, (for example [4,6,15]). Exceptions to this rule were found in the number of macrosetae in the retrolateral preening comb on metatarsus IV, which did not show any intraspecific variation in the specimens examined, and in the number of prolateral and retrolateral macrosetae on the patella of *A. propria* sp. nov.

Morphological and molecular results were concordant for most species groupings. Species in the *clavata* group formed a strongly differentiated clade, supporting the results of the morphological analyses. Intraspecies divergence in the mitochondrial 12S sequence data ranged from a mean of 2.7% for *A. woinarskii* sp. nov. to a mean of 4.8% for *A. valida* sp. nov. *Ariadna tangara* was split into two distinct clades, with a divergence of 7.8%, a South Australian and Tasmanian clade and a Victorian clade. This is the most widely distributed species of *Ariadna* known from Australia and given the high level of genetic divergence, the species requires more detailed analyses, including a larger number of fresh specimens to better explore the population structure and investigate the possibilities of a species complex. *Ariadna tangara* frequently occurs in tube webs within crevices of fallen trees and logs [4]. The movement of firewood is a known human-mediated dispersal mechanism of wood inhabiting insects [43–45] and its role in the dispersal of wood inhabiting segestriid species is one that requires further research.

The suggested COI divergence limits for some mygalomorph spider taxa are 9.5% [46], and while the use of 12S data as a marker for the species level study has been shown to be effective for a number of spider groups (for example [47–50]), the divergence limits for spiders are not as well-understood as for COI but expected to be lower (for example [51]).

Molecular analyses revealed interspecific divergences for *A. subplana* sp. nov. and *A. una* sp. nov. (8.6%) and *A. subplana* sp. nov. and *A. bellatula* sp. nov. (8.2%) to be low. However, the morphological analyses for these groups showed interspecific differences in the shapes of the cymbium and embolus of the male pedipalp and in the size and relative position of the apophyses of metatarsus I. These findings illustrate the importance of an integrative approach to taxonomy, utilising morphological and molecular data, and the potential for barcoding data used in isolation to fail to recognise a species, thus supporting the findings of [52].

**Supplementary Materials:** The following supporting information can be downloaded at https://www.mdpi.com/article/10.3390/taxonomy2040028/s1: Table S1: Pair-wise patristic distances comparison for mtDNA COI haplotypes from 45 *Ariadna* specimens (21 species) and 11 outgroups, as shown in Figure 1. Details for each are in Table 1. Table S2: Pair-wise patristic distances comparison for mtDNA 12S haplotypes from 27 *Ariadna* specimens (17 species) and two outgroups, as shown in Figure 2. Details for each are in Table 1. Table S3: Position of apophyses on metatarsus I of males; analysis of variation in the pattern and number of macrosetae for males and females of species where more than one specimen is known.

**Author Contributions:** Conceptualization, J.R.M. and V.W.F.; methodology, J.R.M.; molecular sequencing and analysis, M.I.S. and T.B. and writing, J.R.M., M.I.S. and V.W.F. All authors have read and agreed to the published version of the manuscript.

**Funding:** This research was funded by Department of Agriculture, Water and Environment, Australia and the Australian Biological Resources Study (ABRS) National Taxonomy Research Grant Program (NTRGP), grant number 4-EHOJ1AU.

**Institutional Review Board Statement:** Not applicable.

**Informed Consent Statement:** Not applicable.

**Data Availability Statement:** Not applicable.

**Acknowledgments:** Many thanks to Matthew Shaw (SAMA) and Simon Hinkley (MV) for the loan of the Segestriidae material examined in this study. Thanks to Mark Harvey (WAM), Matthew Shaw (SAMA), Robert Raven (QM), Nadine Dupérré (ZMH), Elise-Anne Leguin (MNHN) and John Douglas (QVMAG) for access to the type material of Australian Segestriidae and for taking images of the types when we were not able to examine them in person. Thanks to J. Beccaloni (NHM) and S. Hinkley (MV) for assisting with the search for the type of *A. burchelli*.

**Conflicts of Interest:** The authors declare no conflict of interest. The funders had no role in the design of the study; in the collection, analyses or interpretation of the data; in the writing of the manuscript or in the decision to publish the results.

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
