# Peer review of "The Tube-Web Spiders of the Genus Ariadna (Araneae: Segestriidae) from South Australia and Victoriaâ€"

_2673-6500, doi:10.3390/taxonomy2040028_

Round 1
Reviewer 1 Report
This is a generally great piece of work, as I have seen previously from these authors.
A few minor issues need to be addressed.
Davies, V. T. (1985). Araneomorphae (in part). Zoological Catalogue of Australia 3: 49-125. deals with Australian Araidna and should be cited.
Table 2 seems to repeat what is in the descriptions; it can be deleted.
Section 2.1. First paragraph, largely concordant seems to mask the similarity; concord usally refers to the agreement between people and not not adequately reflect the rigour of the statement.
Some apologia needs to be given for using CO-1 as a phylogenetic surrogate. Why no nuclear data?
Italics are lost in the key.
The most important section "4. Taxonomy" is lost in another paragraph.
Defense needs to be given for the confusing use of the term "macrosetae". No previous reason has been given for the confusion with spines, as in Araidna spinosa. Spines are articulated and operate under hydraulic pressure; setae, macro or otherwise cannot be elevated and the setae in the preening combs bear no relation to those spines on the legs..
I do applaud the use of the tern maxillae in spiders; the American tendency to use the term endite is incorrect. Endites are fixed processes off the maxillae or Crustacea, including insects, and always matched by exites on the outer face of the maxillae.
Figures. They are good but "soft" and as you see by the attached on which I have simply use an unsharp mask on the screen capture, a more pleasing sharper image will result from running such a mask on the originals.
Acceptable with minor changes; no additional review needed,

Reviewer 2 Report
The current paper reveals the rich species diversity of the genus Ariadna in South Australia and Victoria. From only 3 known Ariadna species in these regions to up to 26 species including 23 new species, it is amazing. From the content of the paper itself, undoubtedly it is very meaningful. But Of all 25 species in the paper, only 7 species are described or redescribed based on both male and female sexes. It isn’t good that too much species by single sex are reported. So, it is encouraged that authors add several sentences to state or explain why the species by only male sex in present paper isn’t conspecific with the known species by only female sex in Australia). In addition, I also have some other suggestions.
1. The ms isn’t short and it is better if authors can make the structure and description more concise.
I suggest authors to delete “Colour in ethanol”, divide the content of this part and then integrate them into the other parts, for example “Cephalothorax”, “Abdomen” and so on.
For some species, if there is no any other examined material except types for some species, “other material examined” can completely deleted.
The description related with leg measurement occupy too much space. In fact, about this, there is a very common model in most references on spiders. I have given my suggestion to authors in PDF version of ms.
2. Need to further uniform and standardized format as possible as can according to requirements of “Taxonomy”.
Though I don’t know the specific requirements of “Taxonomy”, I still find many formatting problems are present in the present ms. For example, it is unstandardized that the title of table appears below the table in Table 2, and that all species names appeared in the maps aren’t in Italics. And other more problems see PDF version of ms.
3. About figures, in whole, it is good that authors have provided photos of both habitus and genitalia. However, I think that it will be better if authors can improve figures in the following aspects:
Some scale bars are too long. Shorter scale bars are suggested, especially in Fig. 6a-c, e.
In addition, scale bars are better placed in the blank place. The places of scale bars are not good in Figs 6a-d, 20a-e…
It seems that some photos can’t clearly show the related structures. I suggest that authors provide photos with higher quality. If because of limits of instruments, authors can’t get clearer photos, authors can consider to supply some drawing to show the detailed of some important structures. For example, female genitalia and preening comb.
In addition, some important structures are needed to label in the photos using their abbreviations, for example, PE, AE and so on.
4. Several references need to be added: Coyle 1984; Knot et al. 2012; Marsh et al. 2017; Simon et al., 1994.
5. Other more suggestions have been mentioned by me in pdf version of ms. But because there are too many species, for the same suggestions or comments, I haven’t mentioned them again and again.
